# BNEM: A Boltzmann Sampler Based on Bootstrapped Noised Energy Matching

## Abstract

Generating independent samples from a Boltzmann distribution is a highly relevant problem in scientific research, *e.g.* in molecular dynamics, where one has initial access to the underlying energy function but not to samples from the Boltzmann distribution. We address this problem by learning the energies of the convolution of the Boltzmann distribution with Gaussian noise. These energies are then used to generate independent samples through a denoising diffusion approach. The resulting method, Noised Energy Matching (NEM), has lower variance and only slightly higher cost than previous related works. We also improve NEM through a novel bootstrapping technique called Bootstrap NEM (BNEM) that further reduces variance while only slightly increasing bias. Experiments on a collection of problems demonstrate that NEM can outperform previous methods while being more robust and that BNEM further improves on NEM.

## 1 Introduction

A fundamental problem in probabilistic modeling and physical systems simulation is to sample from a target Boltzmann distribution $\mu_{\text{target}} \propto \exp(-\mathcal{E}(x))$ specified by an energy function $\mathcal{E}(x)$. A prominent example is protein folding, which can be formalized as sampling from a Boltzmann distribution (Śledź & Caflisch, 2018) with energies determined by inter-atomic forces (Case et al., 2021). Having access to efficient methods for solving the sampling problem could significantly speed up drug discovery (Zheng et al., 2024) and material design (Komanduri et al., 2000).

However, existing methods for sampling from Boltzmann densities have problems scaling to high dimensions and/or are very time-consuming. As an alternative, Akhound-Sadegh et al. (2024) proposed Iterated Denoising Energy Matching (iDEM), a neural sampler based on denoising diffusion models which is not only computationally tractable but also guarantees good coverage of all modes. iDEM uses a bi-level training scheme that iteratively generates samples from the learned sampler and then does score matching using only the target energy and its gradient. Nevertheless, iDEM requires a large number of samples for its Monte Carlo (MC) score estimate to have low variance and a large number of integration steps even when sampling from simple distributions. Also, its effectiveness highly depends on the choice of noise schedule and score clipping. These disadvantages demand careful hyperparameter tuning and raise issues when working with complicated energies.

To further push the boundary of diffusion-based neural samplers, we propose Noised Energy Matching (NEM), which learns a series of noised energy functions instead of the corresponding score functions. Despite a need to differentiate the energy network when simulating the diffusion sampler, NEM targets less noisy objectives as compared with iDEM. Additionally, using an energy-based parametrization enables NEM to use bootstrapping techniques for more efficient training and Metropolis-Hastings corrections for more accurate simulation. By applying the bootstrapping technique, we propose a variant of NEM called Bootstrap NEM (BNEM). BNEM estimates high noise-level energies by bootstrapping from current energy estimates at slightly lower noise levels. BNEM increases bias but reduces variance in its training target.

We conduct experiments on a 2-dimensional 40 Gaussian Mixture Model (GMM), a 4-particle double-welling potential (DW-4), a 13-particle Lennard-Jones potential (LJ-13) and a 55-particle Lennard-Jones potential (LJ-55). We empirically find that our methods lead to state-of-the-art performance on these tasks. Additionally, we found that targeting energies instead of scores is more

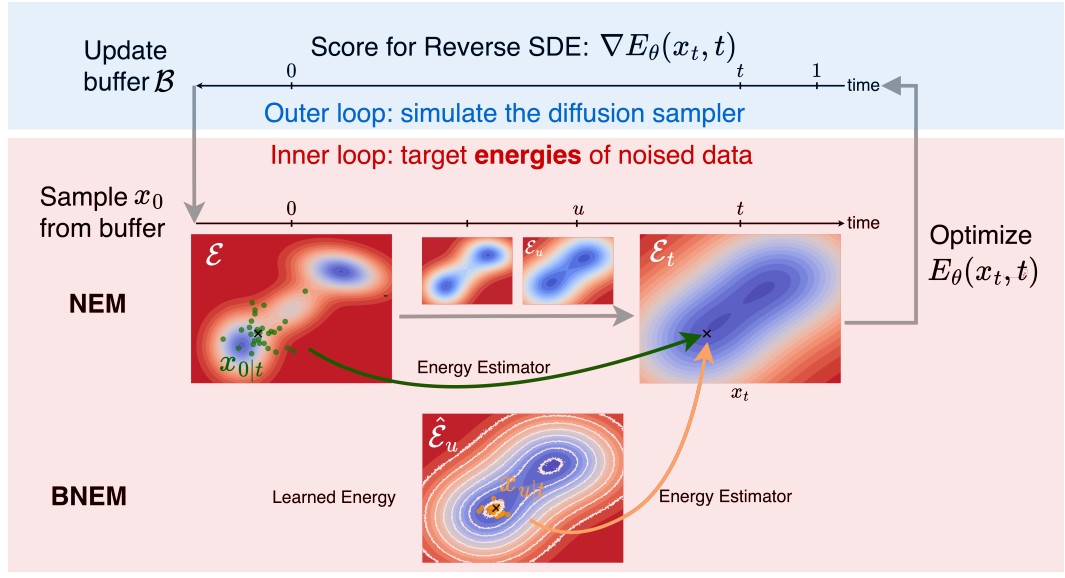

Figure 1: Both EnDEM and BEnDEM parameterize a time-dependent energy network $E_\theta(x_t, t)$ to target the energies of noised data. EnDEM targets an MC energy estimator computed by system energies; BEnDEM targets a Bootstrap energy estimator computed by learned energies at a slightly lower noise level. Contours are the ground truth energies at different noise levels; ● represents samples used for computing the MC energy estimator, ● represents samples used for computing the Bootstrap energy estimator, and the white contour line represents the learned energy at time $u$.

robust, requiring fewer Monte Carlo samples during training and fewer integration steps during sampling. This compensates for the need to differentiate through energy networks to obtain scores.

Our contributions are as follows:

- We introduce NEM in section 3.3, including its methodology and theoretical analysis on training target variance and bias, which showcases the advantage of targeting noised energies rather than noised scores.

- We introduce BNEM in section 3.4, where we also theoretically show the Variance-Bias trade-off implied by the bootstrapping energy estimation.

- We present experiment results on four different tasks in section 5, showcasing the advantage of BNEM and NEM compared with DEM. We also conduct ablation studies, which show that NEM is more robust than DEM regarding the number of samples used for training and the number of integration steps used for sampling.

## 2 PRELIMINARY

We consider learning a generative model for sampling from the Boltzmann distribution

$$\mu_{\text{target}} = \frac{\exp(-\mathcal{E}(x))}{Z}, \quad \text{where} \quad Z = \int \exp(-\mathcal{E}(x))dx, \tag{1}$$

$\mathcal{E}$ is the energy function and $Z$ is the intractable partition function. Generating accurate samples from this type of distribution is highly challenging. The recent success of Diffusion Models provides a promising way to solve this issue.

**Diffusion Models** (Sohl-Dickstein et al., 2015; Ho et al., 2020; Song et al., 2020) learn a generative process that starts from a known and tractable base distribution, *a.k.a.* denoising process, which is the inverse of a tractable noising process that starts from the target distribution. Formally, given samples from the target distribution, $x_0 \sim \mu_{\text{target}}$, the noising process is an SDE towards a known

base distribution $p_1$:

$$dx_t = f(x_t, t)dt + g(t)dw_t, \quad \text{where} \quad t \in [0, 1], \tag{2}$$

$f(x_t, t)$ is called drift coefficient, $g(t)$ is the diffusion coefficient and $w_t$ is standard Brownian Motion. Diffusion Models work by approximately solving the following inverse SDE:

$$dx_t = [f(x_t, t) - g^2(t)\nabla \log p_t(x_t)]dt + g(t)d\tilde{w}_t \tag{3}$$

where $\tilde{w}_t$ is again standard Brownian Motion. In the example of the Variance Exploding (VE) noising process, $f(x_t, t) \equiv 0$ and the perturbation kernel of the noising process is given by $q_{t|0}(x_t|x_0) = \mathcal{N}(x_t; x_0, \sigma_t^2)$, where $\sigma_t^2 := \int g^2(s)ds$. Then the learning objective of DMs is obtained by using Tweedie's formula (Efron, 2011):

$$\mathcal{L}_{DM} = \mathbb{E}_{x_0 \sim p_0, t \sim [0,1], x_t \sim q_{t|0}} \left[ \left\| \frac{x_0 - x_t}{\sigma_t^2} - s_\theta(x_t, t) \right\|^2 \right] \tag{4}$$

which allows us to approximate the marginal scores $\nabla \log p_t(x_t)$ with a score network $s_\theta(x_t, t)$ that is parameterized by $\theta$ and that targets the conditional scores $\nabla \log p_{t|0}(x_t|x_0) = \nabla \log \mathcal{N}(x_t; x_0, \sigma_t^2 I)$.

## 3 METHODS

In this section, we first provide an overview before presenting the formalization and training paradigm of simulation-free energy matching. We then discuss the theoretical advantages of our energy matching over score matching. Finally, we describe how bootstrapping is employed to gain improvement.

### 3.1 OVERVIEW OF NEM FRAMEWORK

This work intends to train a diffusion-based neural sampler that enables diffusion sampling to draw samples from $\mu_{\text{target}}(x) = \frac{\exp(-\mathcal{E}(x))}{Z}$, where we only have access to the energy function $\mathcal{E}$ without any known data from the target distribution.

As in Figure 1, our methods apply the iterative training paradigm, where the inner loop updates the buffer for training the neural sampler, and the outer loop uses the neural sampler to collect new pseudo data to update the buffer. In the inner loop, our model is trained on varied forms of Monte Carlo estimation of the energy.

### 3.2 DENOISING DIFFUSION-BASED BOLTZMANN SAMPLER

We consider training an energy-based diffusion sampler corresponding to a variance exploding (VE) noising process defined by $dx_t = g(t)dw_t$, where $t \in [0, 1]$, $g(t)$ is a function of time and $w_t$ is Brownian motion. The reverse SDE with Brownian motion $\bar{w}_t$ is $dx_t = -g^2(t)\nabla \log p_t(x_t)d_t + g(t)d\bar{w}_t$, where $p_t$ is the marginal of the diffusion process starting at $p_0 := \mu_{\text{target}}$.

Given the energy $\mathcal{E}(x)$ and the perturbation kernel $q_t(x_t|x_0) = \mathcal{N}(x_t; x_0, \sigma_t^2)$, where $\exp(-\mathcal{E}(x)) \propto p_0(x)$ and $\sigma_t^2 := \int_s^t g^2(s)ds$, one can obtain the marginal noised density $p_t$ as

$$p_t(x_t) \propto \int \exp(-\mathcal{E}(x_0))\mathcal{N}(x_t; x_0, \sigma_t^2 I)dx_0 = \mathbb{E}_{\mathcal{N}(x; x_t, \sigma_t^2 I)}[\exp(-\mathcal{E}(x))]. \tag{5}$$

Going a step further, the RHS of Eq. 5 defines a Boltzmann distribution over the noise-perturbed distribution $p_t$. The noised energy is defined as the negative logarithm of this unnormalized density

$$\mathcal{E}_t(x) := -\log \mathbb{E}_{\mathcal{N}(x; x_t, \sigma_t^2 I)}[\exp(-\mathcal{E}(x))], \quad \text{where} \quad \exp(-\mathcal{E}_t(x_t)) \propto p_t(x_t). \tag{6}$$

**Training on MC estimated targets** We can approximate the gradient of $\log p_t$ by fitting a score network $s_\theta(x_t, t)$ to the gradient of Monte Carlo (MC) estimates of Eq. 6, leading to iDEM. The MC score estimator $S_K$ and the training objective can be written as

$$S_K(x_t, t) := \nabla \log \frac{1}{K} \sum_{i=1}^{K} \exp(-\mathcal{E}(x_{0|t}^{(i)})), \quad x_{0|t}^{(i)} \sim \mathcal{N}(x; x_t, \sigma_t^2 I), \tag{7}$$

$$\mathcal{L}_{\text{DEM}}(x_t, t) := \|S_K(x_t, t) - s_\theta(x_t, t)\|^2. \tag{8}$$

Alternatively, we can fit an energy network $E_\theta(x_t, t)$ to MC estimates of Eq. 6. The gradient of this energy network w.r.t. input $x_t$, *i.e.* $\nabla E_\theta(x_t, t)$, can then be used to estimate the score required for diffusion-based sampling. The MC energy estimator $E_K$ and the training objective can be written as

$$E_K(x_t, t) := -\log \frac{1}{K} \sum_{i=1}^{K} \exp(-\mathcal{E}(x_{0|t}^{(i)})), \quad x_{0|t}^{(i)} \sim \mathcal{N}(x; x_t, \sigma_t^2 I), \tag{9}$$

$$\mathcal{L}_{\text{NEM}}(x_t, t) := \|E_K(x_t, t) - E_\theta(x_t, t)\|^2. \tag{10}$$

To enable diffusion-based sampling, one is required to differentiate the energy network to obtain the marginal scores, *i.e.* $\nabla E_\theta(x_t, t)$, which doubles the computation of evaluating $E_\theta(x_t, t)$. Notice that $S_K(x_t, t) = -\nabla E_K(x_t, t)$, regressing the MC energy estimator $E_K$ doesn't need to compute the gradient of target energy $\mathcal{E}$ during training but it is required to compute the gradient of the energy network $E_\theta$ during sampling. In other words, it moves the need for differentiation from given energy function $\mathcal{E}$ to neural networks $E_\theta$, which can be beneficial for training on complicated energy functions.

**Bi-level Iterative Training Scheme**  To train the diffusion on the estimated targets, we should obtain noising exact samples from the target. Previous works (Akhound-Sadegh et al., 2024; Midgley et al., 2023) used data points generated by a current learned denoising procedure.

We follow their approach and use a bi-level iterative training scheme for noised energy matching. This involves

- An outer loop that simulates the diffusion sampling process to generate more informative samples. These samples are then used to update a replay buffer $\mathcal{B}$.
- A simulation-free inner loop that matches the noised energies (NEM) or scores (DEM) evaluated at noised versions of the samples stored in the replay buffer.

The significance of this iterated training scheme is proven by Akhound-Sadegh et al. (2024). We, therefore, stick to using it for all relevant samplers' training. The iterated training procedure of NEM is illustrated in Algorithm 1, and its training pipeline is visualized in Figure 1.

### 3.3 ENERGY-BASED LEARNING VS SCORE-BASED LEARNING

Both the MC score estimator $S_K$ and the MC energy estimator $E_K$ are biased estimators, where the bias of $S_K$ (Eq. 12) is characterized by Akhound-Sadegh et al. (2024) that it can decrease to 0 when the number of MC samples $K$ increases. We first characterize the bias of $E_K$ in the following Proposition 1, which shows the advantage of NEM in terms of the smaller bias of its regression target.

**Proposition 1** *If* $\exp(-\mathcal{E}(x_{0|t}^{(i)}))$ *is sub-Gaussian, then there exists a constant* $\tilde{c}(x_t)$ *such that with probability* $1 - \delta$ *over* $x_{0|t}^{(i)} \sim \mathcal{N}(x_t, \sigma_t^2)$, *we have*

$$\|E_K(x_t, t) - \mathcal{E}_t(x_t)\| \leq \frac{\tilde{c}(x_t)\sqrt{\log(1/\delta)}}{\sqrt{K}} \tag{11}$$

*with* $c(x_t)/\tilde{c}(x_t) = 2(1 + \|\nabla \mathcal{E}_t(x_t)\|)$, *where*

$$\|S_K(x_t, t) - S_t(x_t)\| \leq \frac{c(x_t)\sqrt{\log(1/\delta)}}{\sqrt{K}}. \tag{12}$$

Proposition 1 shows that the training target of NEM has a smaller error bound (Eq. 11) than the one of DEM (Eq. 12), especially in regions with a steep gradient, *i.e.* large $\|\nabla \mathcal{E}_t(x)\|$. Specifically, the bias of $E_K$ can be characterized through the above error bound, which is provided in the following corollary:

**Corollary 1** *If* $\exp(-\mathcal{E}(x_{0|t}^{(i)}))$ *is sub-Gaussian, then the bias of* $E_K$ *can be approximated as*

$$\text{Bias}[E_K(x_t, t)] := \mathbb{E}[E_K(x_t, t)] - \mathcal{E}_t(x_t) = \frac{v_{0t}(x_t)}{2m_t^2(x_t)K} \tag{13}$$

*where* $m_t(x) = \exp(-\mathcal{E}_t(x_t))$ *and* $v_{0t}(x) = \text{Var}_{\mathcal{N}(x; x_t, \sigma_t^2 I)}[\exp(-\mathcal{E}_t(x))]$.

---

**Algorithm 1** Iterated training for Noised Energy Matching

---

**Require:** Network $E_\theta$, Batch size $b$, Noise schedule $\sigma_t^2$, Base distribution $p_1$, Num. integration steps $L$, Replay buffer $\mathcal{B}$, Max Buffer Size $|\mathcal{B}|$, Num. MC samples $K$
1: **while** Outer-Loop **do**
2:     $\{x_1\}_{i=1}^b \sim p_1(x_1)$
3:     $\{x_0\}_{i=1}^b \leftarrow \text{sde.int}(\{x_1\}_{i=1}^b, -\nabla E_\theta, L)$         $\triangleright$ Simulate the reverse SDE for sampling
4:     $\mathcal{B} = (\mathcal{B} \cup \{x_0\}_{i=1}^b)$                 $\triangleright$ Update Buffer $\mathcal{B}$
5:     **while** Inner-Loop **do**
6:         $x_0 \leftarrow \mathcal{B}.\text{sample}()$            $\triangleright$ Uniform sampling from $\mathcal{B}$
7:         $t \sim \mathcal{U}(0,1), x_t \sim \mathcal{N}(x_0, \sigma_t^2)$
8:         $\mathcal{L}_{\text{NEM}}(x_t, t) = \|E_K(x_t, t) - E_\theta(x_t, t)\|^2$
9:         $\theta \leftarrow \text{Update}(\theta, \nabla_\theta \mathcal{L}_{\text{NEM}})$
10:    **end while**
11: **end while**
**Ensure:** $s_\theta$

---

**Algorithm 2** Inner-loop of Bootstrap Noised Energy Matching training

---

**Require:** Network $E_\theta$, Batch size $b$, Noise schedule $\sigma_t^2$, Replay buffer $\mathcal{B}$, Num. MC samples $K$
1: **while** Inner-Loop **do**
2:     $x_0 \leftarrow \mathcal{B}.\text{sample}()$               $\triangleright$ Uniform sampling from $\mathcal{B}$
3:     $t \sim \mathcal{U}(0,1), x_t \sim \mathcal{N}(x_0, \sigma_t^2)$
4:     $n \leftarrow \arg\{i : t \in [t_i, t_{i+1}]\}$           $\triangleright$ Identify the time split range of $t$
5:     $s \sim \mathcal{U}(t_{n-1}, t_n), x_s \sim \mathcal{N}(x_0, \sigma_s^2)$
6:     $l_s(x_s) \leftarrow \|E_K(x_s, s) - E_\theta(x_s, s)\|^2 / \sigma_s^2$
7:     $l_t(x_t) \leftarrow \|E_K(x_t, t) - E_\theta(x_t, t)\|^2 / \sigma_t^2$
8:     $\alpha \leftarrow \min(1, l_t(x_t)/l_s(x_s))$
9:     with probability $\alpha$,
10:         $\mathcal{L}_{\text{BNEM}}(x_t, t) = \|E_K(x_t, t, s; \text{StopGrad}(\theta)) - E_\theta(x_t, t)\|^2$
11:    Otherwise,                 $\triangleright$ Use MC estimator if the model is not well trained
12:         $\mathcal{L}_{\text{BNEM}}(x_t, t) = \|E_K(x_t, t) - E_\theta(x_t, t)\|^2$
13:    $\theta \leftarrow \text{Update}(\theta, \nabla_\theta \mathcal{L}_{\text{BNEM}})$
14: **end while**
**Ensure:** $E_\theta$

---

The complete proofs of Proposition 1 and Corollary 1 are given in Appendix A. Additionally, we characterize the variances of $S_K$ and $E_K$ as follows.

**Proposition 2** *If $\exp(-\mathcal{E}(x_{0|t}^{(i)}))$ is sub-Gaussian and $\|\nabla \exp(-\mathcal{E}(x_{0|t}^{(i)}))\|$ is bounded, the total variance of the MC score estimator $S_K$ is consistently larger than that of the MC energy estimator $E_K$ in regions associated with low energies, with*

$$\frac{\mathbf{tr}\left(Cov[S_K(x_t, t)]\right)}{Var[E_K(x_t, t)]} = 4(1 + \|\nabla \mathcal{E}_t(x_t)\|)^2. \tag{14}$$

*In regions associated with high energies, $Var[E_K(x_t, t)] < \mathbf{tr}(Cov[S_K(x_t, t)])$ holds when the target energy $\mathcal{E}(x_t)$ is positively related to at least one element of the score $\nabla \mathcal{E}(x_t)$.*

This shows that the MC energy estimator can provide a less noisy training signal than the score one, showcasing the theoretical advantage of NEM compared with DEM. One might wonder if the differentiation of the energy network could amplify errors in NEM. The answer is that, even though the energy errors are amplified, these errors are typically very small, and therefore the errors in the differentiated energy are still smaller than the score errors. Our experiments illustrate this by supporting NEM over DEM. The complete proof for the above results is provided in Appendix B.

### 3.4 IMPROVEMENT WITH BOOTSTRAPPED ENERGY ESTIMATION

Using an energy network that directly models the noisy energy landscape has additional advantages. Intuitively, the variances of $E_K$ and $S_K$ explode at high noise levels as a result of the VE noising

process. However, we can reduce variance of the training target in NEM by using the learned noised energies at just slightly lower noise levels rather than using the target energy at time $t = 0$. Based on this, we propose Bootstrap NEM, or **BNEM**, which uses a novel MC energy estimator at high noise levels that is bootstrapped from the learned energies at slightly lower noise levels. Suppose that $E_\theta(\cdot, s)$ is an energy network that already provides an accurate estimate of the energy at a low noise level $s$, we can then construct a bootstrap energy estimator at a higher noise level $t > s$ by using

$$E_K(x_t, t, s; \theta) := -\log \frac{1}{K} \sum_{i=1}^{K} \exp(-E_\theta(x_{s|t}^{(i)}, s)), \quad x_{s|t}^{(i)} \sim \mathcal{N}\left(x; x_s, (\sigma_t^2 - \sigma_s^2)I\right), \quad (15)$$

The loss used by BNEM is then

$$\mathcal{L}_{\text{BNEM}}(x_t, t|s) := \|E_K\left(x_t, t, s; \text{StopGrad}(\theta)\right) - E_\theta(x_t, t)\|^2, \quad (16)$$

where the gradient of (Eq. 15) with respect to $\theta$ is stopped. Let $E_\theta(x_s, s)$ is learned from the original MC estimator (Eq. 9) and ideally we suppose it learns perfectly where $E_{\tilde{\theta}}(x_s, s) = E_K(x_s, s)$. By plugging it into Eq. 15, one can show that bootstrapping once $(0 \to s \to t)$ is equivalent to target Eq. 9 with $K^2$ MC samples, *i.e.* $E_K(x_t, t, s; \tilde{\theta}) = E_{K^2}(x_t, t)$. Therefore, bootstrapping multiple times through the bootstrapping trajectory is ideally equivalent to use polynomial number of MC samples, leading to large reductions in both variance and bias of the training target. A detailed discussion can be found in Appendix C.

However, as $E_K$ is a random variable, one can only fit its expected value which is biased (see Corollary 1). Therefore, though we can polynomially reduce (w.r.t. number of MC samples $K$) the original variance and bias of $E_K$, we introduce an accumulated bias when plugging $\mathbb{E}[E_K]$ into Eq. 15. Nevertheless, with appropriate bootstrapping hyperparameters, the bias of the bootstrapped estimator is not necessarily larger than $E_K$. Additionally, it benefits from a training target with reduced variance. We characterize both the bias and variance of this bootstrap energy estimator by the following proposition:

**Proposition 3** *Given a bootstrap trajectory $\{s_i\}_{i=0}^{n}$ such that $\sigma_{s_i}^2 - \sigma_{s_{i-1}}^2 \leq \kappa$, where $s_0 = 0$, $s_n = 1$ and $\kappa > 0$ is a small constant. Suppose $E_\theta$ is incrementally optimized from $t \in [0, 1]$ as follows: if $t \in [s_i, s_{i+1}]$, $E_\theta(x_t, t)$ targets an energy estimator bootstrapped from $\forall s \in [s_{i-1}, s_i]$ using Eq. 15. For $\forall 0 \leq i \leq n$ and $\forall s \in [s_{i-1}, s_i]$, the variance of the bootstrap energy estimator is given by*

$$Var[E_K(x_t, t, s; \theta)] = \frac{v_{st}(x_t)}{v_{0t}(x_t)} Var[E_K(x_t, t)] \quad (17)$$

*and the bias of $E_K(x_t, t, s; \theta)$ is given by*

$$\text{Bias}[E_K(x_t, t, s; \theta)] = \frac{v_{0t}(x_t)}{2m_t^2(x_t)K^{i+1}} + \sum_{j=1}^{i} \frac{v_{0s_j}(x_t)}{2m_{s_j}^2(x_t)K^j}. \quad (18)$$

*where $m_z(x_z) = \exp(-\mathcal{E}_z(x_z))$ and $v_{yz}(x_z) = \text{Var}_{\mathcal{N}(x; x_z, (\sigma_z^2 - \sigma_y^2)I)}[\exp(-\mathcal{E}_y(x))]$ for $\forall 0 \leq y < z \leq 1$.*

A detailed discussion and proof are given in Appendix D. Proposition 3 demonstrates that the bootstrap energy estimator, which estimates the noised energies by sampling from a $x_t$-mean Gaussian with smaller variance, can reduce the variance of the training target while this new target can introduce accumulated bias. Proposition 3 shows that, the bias of BNEM consists of two components: (1) the target bias term which is reduced by a factor of $K^i$ compared to NEM, and (2) the sum of intermediate biases, *i.e.* accumulated bias, which are each reduced by factors of $K^j$ where $j \geq 1$. Since $K$ is typically large and $i \geq 1$, the target bias term in BNEM is substantially smaller than in NEM. While BNEM does introduce additional accumulated terms, these terms are also reduced by powers of $K$, resulting in an overall lower bias compared to NEM under similar computational budgets. Therefore, within proper choice of number of MC samples $K$ and bootstrap trajectory $\{s_i\}_{i=1}^{n}$, the bias of BNEM (Eq. 13) can be smaller than that of NEM (Eq. 18) while enjoying a lower-variance training target. This provides theoretical guarantee for improving performance via bootstrapping. In the following paragraph, we define how we select $s$ and $t$ using a novel variance-controlled bootstrap schedule and training scheme.

**Variance-Controlled Bootstrap Schedule**   BNEM aims to trade the bias of the learning target to its variance. To ensure that the variance of the Bootstrap energy estimator at $t$ bootstrapped from $s$ is controlled by a predefined bound $\beta$, *i.e.* $\sigma_t^2 - \sigma_s^2 \leq \beta$, we first split the time range $[0,1]$ with $0 = t_0 < t_1 < ... < t_N = 1$ such that $\sigma_{t_{i+1}}^2 - \sigma_{t_i}^2 \leq \beta/2$; then we uniformly sample $s$ and $t$ from adjacent time splits during training for variance control.

**Training of BNEM**   To train BNEM, it is crucial to account for the fact that bootstrap energy estimation at $t$ can only be accurate when the noised energy at $s$ is well-learned. To favor this, we first use NEM to obtain an initial energy network and then apply training with the bootstrapped estimator. We then aim to use bootstrapping only when the energy network is significantly better at time $s$ than at time $t$. We quantify this by evaluating the NEM losses at times $t$ and $s$ given a clean data point $x_0$. In particular, we could compare $\tilde{l}_s(x_s) = \mathcal{L}_{\text{NEM}}(x_s, s)$ and $\tilde{l}_t(x_t) = \mathcal{L}_{\text{NEM}}(x_t, t)$, where $x_s \sim \mathcal{N}(x_0, \sigma_s^2 I)$ and $x_t \sim \mathcal{N}(x_0, \sigma_t^2 I)$. However, the variance of training targets increases with time, meaning that a direct comparison of these losses is not reliable. To avoid this, we normalize these losses according to the noise schedule variance and compare instead $l_s(x_s) = \mathcal{L}_{\text{NEM}}(x_s, s)/\sigma_s^2$ and $l_t(x_t) = \mathcal{L}_{\text{NEM}}(x_t, t)/\sigma_t^2$. We then adopt a rejection training scheme according to the ratio of these normalized NEM losses:

   (a) given $s$, $t$ and $x_0$, we first noise $x_0$ to $s$ and $t$ respectively, and compute the normalized losses defined above, *i.e.* $l_s(x_s)$ and $l_t(x_t)$;

   (b) these losses indicate how well the energy network fits the noised energies at different times; we then compute $\alpha = \min(1, l_s(x_s)/l_t(x_t))$;

   (c) with probability $\alpha$, we accept targeting an energy estimator at $t$ bootstrapped from $s$ and otherwise, we stick to targeting the original MC energy estimator.

We provide a full description of the inner-loop of BNEM training in Algorithm 2.

## 4   RELATED WORKS

**Boltzmann Generator.** To learn a neural sampler for Boltzmann distribution, unlike data-driven tasks where a sufficient amount of data is available, simply minimizing the reverse Kullback-Leibler (KL) divergence, *i.e.* $\min_\theta \mathbb{D}_{KL}(\mu_{target} \| q_\theta)$, can lead to mode-seeking behavior. Boltzmann Generator(Noé et al., 2019) addresses this problem by minimizing the combination of forward and reverse KL divergence.

**PIS, DDS, and FAB.** Inspired by the rapid development of deep generative models, *e.g.* diffusion models (Song & Ermon, 2019; Ho et al., 2020), pseudo samples could be generated from an arbitrary prior distribution. Then, we can train the neural samplers by matching these sample trajectories, as in Path Integral Sampler (PIS)(Zhang & Chen, 2022) and Denoising Diffusion Sampler (DDS) (Vargas et al., 2023). Midgley et al. (2023) further deploy a replay buffer for the trajectories while proposing an $\alpha$-divergence as the objective to avoid mode-seeking. However, these methods require simulation during training, which still poses challenges for scaling up to higher dimensional tasks.

**iDEM.** To further boost scalability and mode-coverage, iDEM (Akhound-Sadegh et al., 2024) is proposed to target an MC score estimator that estimates scores of noised data, enabling the usage of the efficient diffusion sampling. iDEM is trained with a bi-level scheme: (1) a simulation-free inner-loop that targets time-involved scores of buffer data; (2) an outer-loop that simulates the learned diffusion sampler to generate more informative data in the buffer. It achieves previous state-of-the-art performance on the tasks above.

**iEFM.** To regress $x_t$ directly, one can leverage Flow Matching Lipman et al. (2023). Woo & Ahn (2024) proposes iEFM, a variant of iDEM that targets the MC estimated vector fields in a Flow Matching fashion, which is empirically found to outperform iDEM in GMM and DW-4. In fact, we found that iEFM and iDEM can be linked through Tweedie's formula (Efron, 2011) shown in Appendix H with supplementary experiments provided in Appendix J.8.

Table 1: Neural sampler performance comparison for 4 different energy functions. We measured the performance using data Wasserstein-2 distance (x-$\mathcal{W}_2$), Energy Wasserstein-2 distance ($\mathcal{E}$-$\mathcal{W}_2$), and Total Variation (TV). * indicates divergent training. Each sampler is evaluated with 3 random seeds and we report the mean $\pm$ standard deviation for each metric. † indicates a large standard deviation of metric ($> 20\%$) and we report the best value.

| Energy → | **GMM-40** ($d=2$) | | | **DW-4** ($d=8$) | | | **LJ-13** ($d=39$) | | | **LJ-55** ($d=165$) | | |
|---|---|---|---|---|---|---|---|---|---|---|---|---|
| Sampler ↓ | x-$\mathcal{W}_2$ | $\mathcal{E}$-$\mathcal{W}_2$ | TV | x-$\mathcal{W}_2$ | $\mathcal{E}$-$\mathcal{W}_2$ | TV | x-$\mathcal{W}_2$ | $\mathcal{E}$-$\mathcal{W}_2$ | TV | x-$\mathcal{W}_2$ | $\mathcal{E}$-$\mathcal{W}_2$ | TV |
| DDS | $15.04_{\pm2.97}$ | $305.13_{\pm186.06}$ | $0.96_{\pm0.01}$ | $0.82_{\pm0.21}$ | $558.79_{\pm787.92}$ | $0.38_{\pm0.14}$ | * | * | | | * | |
| PIS | $6.58_{\pm1.68}$ | $79.86_{\pm7.79}$ | $0.95_{\pm0.01}$ | * | * | * | * | | * | * | * | * |
| FAB | $9.08_{\pm1.41}$ | $47.60_{\pm7.24}$ | $0.79_{\pm0.07}$ | $0.62_{\pm0.02}$ | $112.70_{\pm20.33}$ | $0.38_{\pm0.02}$ | * | * | * | * | * | * |
| iDEM | $8.21_{\pm5.43}$ | $60.49_{\pm70.12}$ | $0.82_{\pm0.03}$ | $0.50_{\pm0.03}$ | $2.80_{\pm1.72}$ | $0.16_{\pm0.01}$ | $0.87_{\pm0.00}$ | $6770†$ | $0.06_{\pm0.01}$ | $2.06_{\pm0.04}$ | $17651†$ | $0.16_{\pm0.02}$ |
| NEM (ours) | $5.28_{\pm0.89}$ | $44.56_{\pm39.56}$ | $0.91_{\pm0.02}$ | $\mathbf{0.48_{\pm0.02}}$ | $0.85_{\pm0.52}$ | $\mathbf{0.14_{\pm0.01}}$ | $0.87_{\pm0.01}$ | $5.01_{\pm3.14}$ | $\mathbf{0.03_{\pm0.00}}$ | $1.90_{\pm0.01}$ | $118.58_{\pm106.63}$ | $0.10_{\pm0.02}$ |
| BNEM (ours) | $\mathbf{3.66_{\pm0.30}}$ | $\mathbf{1.87_{\pm1.00}}$ | $\mathbf{0.79_{\pm0.04}}$ | $0.49_{\pm0.02}$ | $\mathbf{0.38_{\pm0.09}}$ | $\mathbf{0.14_{\pm0.01}}$ | $0.86_{\pm0.00}$ | $\mathbf{1.02_{\pm0.69}}$ | $\mathbf{0.03_{\pm0.00}}$ | $\mathbf{1.88_{\pm0.01}}$ | $119.46_{\pm77.92}$ | $\mathbf{0.08_{\pm0.01}}$ |

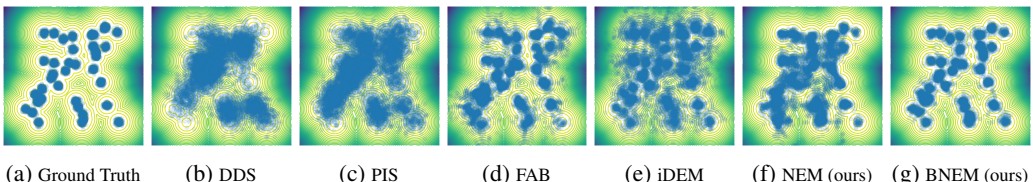

(a) Ground Truth  (b) DDS  (c) PIS  (d) FAB  (e) iDEM  (f) NEM (ours)  (g) BNEM (ours)

Figure 2: Sampled points from samplers applied to GMM-40 potentials, with the ground truth represented by contour lines.

# 5 EXPERIMENTS

We evaluate our methods and baseline models on 4 potentials. A complete description of all energy functions, metrics, and experiment setups is in Appendix I. Supplementary experiments can be found in Appendix J. We provide an anonymous link to our code for implementation reference[1].

**Datasets.** We evaluate all neural samplers on 4 different datasets: a GMM with 40 modes ($d=2$), a 4-particle double-well (DW-4) potential ($d=8$), a 13-particle Lennard-Jones (LJ-13) potential ($d=39$) and a 55-particle Lennard-Jones (LJ-55) potential ($d=165$). For LJ-n potentials, the energy can be extreme when particles are too close to each other, creating problems for estimating noised energies. To overcome this issue, we smooth the Lennard-Jones potential through the cubic spline interpolation, according to Moore et al. (2024).

**Baseline.** We compare NEM and BNEM to following recent works: Denoising Diffusion Sampler (DDS)(Vargas et al., 2023), Path Integral Sampler (PIS)(Zhang & Chen, 2022), Flow Annealed Bootstrap (FAB)(Midgley et al., 2023) and Iterated Denoising Energy Matching (iDEM)(Akhound-Sadegh et al., 2024). Due to the high complexity of DDS and PIS training due to their simulation-based nature, we limit their integration step when sampling to 100. Also, as iDEM (Akhound-Sadegh et al., 2024) shows excellent performance of GMM and DW-4, we consider showing its robustness together with NEM and BNEM by limiting both the integration step and number of MC samples to 100. For complex tasks, *i.e.* LJ-13 and LJ-55, we stick to using 1000 steps for reverse SDE integration and 1000 MC samples in the estimators. For BNEM, we set $\beta = 0.1$ for all tasks. We train all samplers using an NVIDIA-A100 GPU.

**Architecture.** We use the same network architecture (MLP for GMM and EGNN for particle systems, *i.e.* DW-4, LJ-13, and LJ-55) in DDS, PIS, iDEM, NEM and BNEM. To ensure a similar number of parameters for each sampler, if the score network is parameterized by $s_\theta(x,t) = f_\theta(x,t)$, the energy network is set to be $E_\theta(x,t) = \mathbf{1}^\top f_\theta(x,t) + c$ with a learnable scalar $c$. Furthermore, this setting ensures SE(3) invariance for the energy network. However, FAB requires an invertible architecture. For a fair comparison, we replace the neural network architecture in FAB with a continuous flow matching, to ensure a similar number of parameters for each sampler.

**Metrics.** We use data 2-Wasserstein distance (x-$\mathcal{W}_2$), energy 2-Wasserstein distance ($\mathcal{E}-\mathcal{W}_2$), and Total Variation (TV) as metrics. TV is computed from data in GMM; For equivariant systems, *i.e.* DW-4, LJ-13, and LJ-55, TV is based on the interatomic distance. To compute $\mathcal{W}_2$ and TV, we use

---

[1]https://anonymous.4open.science/r/nem-D664/README.md

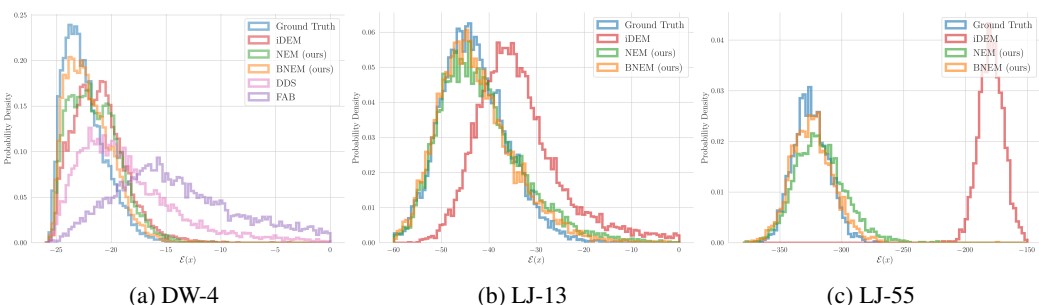

(a) DW-4          (b) LJ-13          (c) LJ-55

Figure 3: Histogram for energies of samples generated by each sampler.

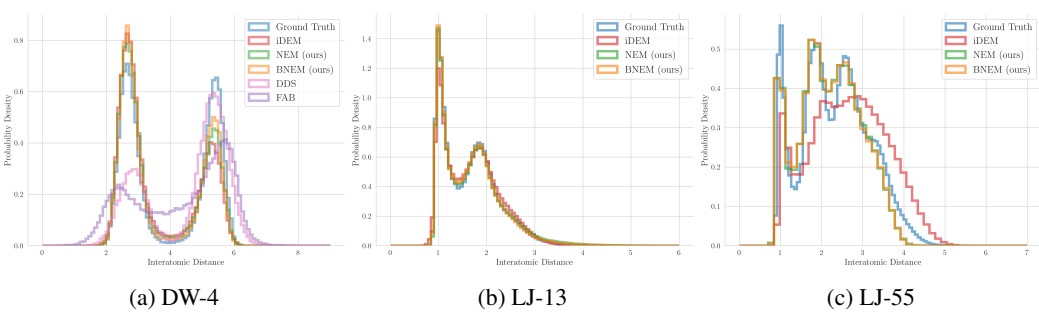

(a) DW-4          (b) LJ-13          (c) LJ-55

Figure 4: Histogram for interatomic distance of samples generated by each sampler.

pre-generated samples as datasets: (a) For GMM, we sample from the ground truth distribution; (b) For DW-4, LJ-13 and LJ-55, we use samples from Klein et al. (2023b).

## 5.1 MAIN RESULTS

We report x-$\mathcal{W}_2$, $\mathcal{E}$-$\mathcal{W}_2$, and TV for all tasks in Table 1. The table demonstrates that by targeting less noisy objectives, NEM outperforms DEM on most metrics, particularly for complex tasks such as LJ-13 and LJ-55. Figure 2 visualizes the generated samples from each sampler in the GMM benchmark. When the compute budget is constrained—by reducing neural network size for DDS, PIS, and FAB, and limiting the number of integration steps and MC samples to 100 for all baselines—none of them achieve high-quality samples with sufficient mode coverage. In particular, iDEM produces samples that are not concentrated around the modes in this setting. Conversely, NEM generates samples with far fewer outliers, focusing more on the modes and achieving the best performance on all metrics. BNEM can further improve on top of NEM, generating data that are most similar to the ground truth ones.

For the equivariant tasks, *i.e.* DW-4, LJ-13, and LJ-55, we compute the energies and interatomic distances for each generated sample. Empirically, we found that smoothing the energy-distance function by a cubic spline interpolation (Moore et al., 2024) to avoid extreme values (i.e. when the particles are too close) is a key step when working with the Lennard Jones potential. Furthermore, this smoothing technique can be applied in a wide range of many-particle systems (Pappu et al., 1998). Therefore, NEM and BNEM can be applied without significant modeling challenges. We also provide an ablation study on applying this energy-smoothing technique to score-based iDEM. The results in Table 4 suggest that it could help to improve the performance of iDEM but NEM still outperforms it.

Figures 3 and 4, show the histograms of the energies and interatomic distances, respectively. It shows that both NEM and BNEM closely match the ground truth densities, outperforming all other baselines. Moreover, for the most complex task, LJ-55, NEM demonstrates greater stability during training, generating more low-energy samples, unlike iDEM, which is more susceptible to instability and variance from different random seeds.

For $\mathcal{E}$-$\mathcal{W}_2$, it is susceptible to outliers with high energy, especially in complex tasks like LJ-13 and LJ-55, where an outlier that corresponds to a pair of particles that are close to each other can result in

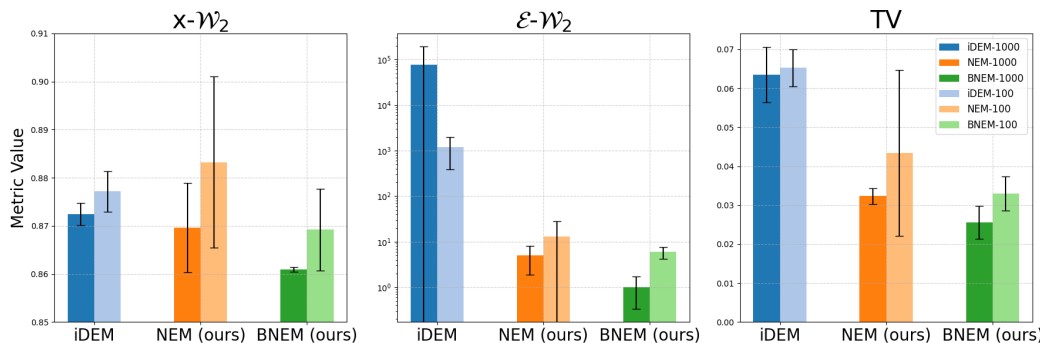

Figure 5: Barplots comparing DEM, NEM, and BNEM evaluations with 1000 vs. 100 integration steps and MC samples on the LJ-13 benchmark.

an extremely large value of this metric. We find that for LJ-n tasks, NEM and BNEM tend to generate samples with low energies and result in low $\mathcal{E}\text{-}\mathcal{W}_2$, while iDEM can produce high energy outliers and therefore corresponds to the extremely high values of this metric. We also notice that for LJ-55, the mean value of $\mathcal{E}\text{-}\mathcal{W}_2$ of BNEM is similar to NEM, while its results have smaller variance indicating a better result. Overall, NEM consistently outperforms iDEM, while BNEM can further improve NEM and achieve state-of-the-art performance in all tasks.

## 5.2 ROBUSTNESS OF DEM, NEM, AND BNEM

Table 1 partially shows the better robustness of NEM and BNEM in terms of fewer integration steps and MC samples, compared with iDEM and other baselines. We further explore the robustness in a more complex benchmark, LJ-13, to demonstrate the advantage of our energy-based models, *i.e.* NEM and BNEM. Figure 5 visualizes the difference of metrics when reducing the integration steps and number of MC samples from 1000 to 100. A complete comparison between iDEM, NEM, and BNEM under different computational budgets is provided in Table 5 (Appendix J). It shows that when limiting the computing budgets, iDEM can degrade significantly in GMM and DW-4 potentials, while NEM is less affected. In LJ-13 potential, both iDEM and NEM can degrade, while NEM-100 is still better than DEM-100 and even better than DEM-1000 in $\mathcal{E}\text{-}\mathcal{W}_2$ and TV. Furthermore, BNEM can achieve better performance. BNEM-100 is less affected and matches the performance of iDEM-1000 in GMM and DW-4, and even outperforms iDEM-1000 in the more complex LJ-13 potential, showcasing its capability.

## 6 CONCLUSION

In this work, we propose NEM and BNEM, neural samplers for Boltzmann distribution and equilibrium systems like many-body systems. NEM uses a novel Monte Carlo energy estimator with reduced bias and variance. BNEM builds on NEM, employing an energy estimator bootstrapped from lower noise-level data, theoretically trading bias for variance. Empirically, BNEM achieves state-of-the-art results on 4 different benchmarks, GMM, DW-4, LJ-13, and LJ-55.

**Limitations and future work.** Even though NEM can outperform DEM with fewer integration steps, the requirement of differentiating the neural network w.r.t. the input poses a memory issue for high-dimensional tasks, and therefore would require further improvement for scalability in terms of memory in the future; Secondly, though BNEM demonstrates its potential to achieve improvement on top of NEM, its training process is yet not stable enough. Therefore, improving its training stability is of interest in the future; Besides, the cubic-spline interpolation technique is applied for the LJ-n potentials, and therefore a more general learning-based technique that can fix the extreme energy issue, such as contrastive learning, is more desired; Furthermore, the well-learned noised energies allow us more possible ways to improve generation quality beyond normal denoising diffusion approach, such as integrating Metropolis-Hastings correction inside the denoising process to generate more accurate samples.

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

APPENDIX

## TABLE OF CONTENTS

## A  PROOF OF PROPOSITION 1

**Proposition 1** *If* $\exp(-\mathcal{E}(x_{0|t}^{(i)}))$ *is sub-Gaussian, then there exists a constant* $\tilde{c}(x_t)$ *such that with probability* $1 - \delta$ *over* $x_{0|t}^{(i)} \sim \mathcal{N}(x_t, \sigma_t^2)$, *we have*

$$\|E_K(x_t, t) - \mathcal{E}_t(x_t)\| \leq \frac{\tilde{c}(x_t)\sqrt{\log(1/\delta)}}{\sqrt{K}} \tag{19}$$

*with* $c(x_t)/\tilde{c}(x_t) = 2(1 + \|\nabla \mathcal{E}_t(x_t)\|)$, *where*

$$\|S_K(x_t, t) - S_t(x_t)\| \leq \frac{c(x_t)\sqrt{\log(1/\delta)}}{\sqrt{K}} \tag{20}$$

**Proof.** We first introduce the error bound of the MC score estimator $S_K$, where $S_K = \nabla E_K$, proposed by Akhound-Sadegh et al. (2024) as follows

$$\|S_K(x_t, t) - S(x_t, t)\| \leq \frac{2C\sqrt{\log(\frac{2}{\delta})}(1 + \|\nabla \mathcal{E}_t(x_t)\|)\exp(\mathcal{E}_t(x_t))}{\sqrt{K}} \tag{21}$$

which assumes that $\exp(-\mathcal{E}(x_{0|t}^{(i)}))$ is sub-Gaussian. Let's define the following variables

$$m_t(x_t) = \exp(-\mathcal{E}_t(x_t)) \tag{22}$$

$$v_{st}(x_t) = \text{Var}_{\mathcal{N}(x_t, (\sigma_t^2 - \sigma_s^2)I)}[\exp(-\mathcal{E}(x))] \tag{23}$$

By the sub-Gaussianess assumption, it's easy to show that the constant term $C$ in Equation 21 is $C = \sqrt{2v_{0t}(x_t)}$. Notice that $E_K$ is a logarithm of an unbiased estimator . By the sub-Gaussian assumption, one can derive that $E_K$ is also sub-Gaussian. Furthermore, it's mean and variance can be derived by employing a first-order Taylor expansion:

$$\mathbb{E}E_K(x_t, t) \approx \mathcal{E}_t(x_t) + \frac{v_{0t}(x_t)}{2m_t^2(x_t)K} \tag{24}$$

$$\text{Var}[E_K(x_t, t)] \approx \frac{v_{0t}(x_t)}{m_t^2(x_t)K} \tag{25}$$

And one can obtain its concentration inequality by incorporating the sub-Gaussianess

$$\|E_K(x_t, t) - \mathbb{E}[E_K(x_t, t)]\| \leq \sqrt{2\frac{v_{0t}(x_t)}{m_{0t}^2(x_t)K}\log\frac{2}{\delta}} \tag{26}$$

By using the above Inequality 26 and the triangle inequality

$$\|E_K(x_t, t) - \mathcal{E}_t(x_t)\| \leq \|\log E_K(x_t, t) - \mathbb{E}[E_K(x_t, t)]\| + \|\mathbb{E}[E_K(x_t, t)] - \mathcal{E}_t(x_t)\| \quad (27)$$

$$= \|\log E_K(x_t, t) - \mathbb{E}[E_K(x_t, t)]\| + \frac{v_{0t}(x_t)}{2m_t^2(x_t)K} \quad (28)$$

$$\leq \sqrt{2\frac{v_{0t}(x_t)}{m_t^2(x_t)K}\log\frac{2}{\delta}} + \frac{v_{0t}(x_t)}{2m_t^2(x_t)K} \quad (29)$$

$$= \frac{C\sqrt{\log\frac{2}{\delta}}\exp(\mathcal{E}_t(x_t))}{\sqrt{K}} + O(1/K) \quad (30)$$

$$= \frac{c(x_t)}{2(1 + \|\nabla\mathcal{E}_t(x_t)\|)}\frac{\sqrt{\log\frac{1}{\delta}}}{\sqrt{K}} \quad (31)$$

Therefore, we have $c(x_t) = 2(1 + \|\nabla\mathcal{E}_t(x_t)\|)\tilde{c}(x_t)$. It demonstrates a less biased estimator, which, what's more, doesn't require a sub-Gaussianess assumption over $\|\nabla\mathcal{E}(x_{0|t}^{(i)})\|$. $\qquad\square$

## B    PROOF OF PROPOSITION 2

**Proposition 2** *If $\exp(-\mathcal{E}(x_{0|t}^{(i)}))$ is sub-Gaussian and $\|\nabla\exp(-\mathcal{E}(x_{0|t}^{(i)}))\|$ is bounded, the total variance of the MC score estimator $S_K$ is consistently larger than that of the MC energy estimator $E_K$ in low-energy regions, with*

$$\frac{\mathbf{tr}\left(Cov[S_K(x_t, t)]\right)}{Var[E_K(x_t, t)]} = 4(1 + \|\nabla\mathcal{E}_t(x_t)\|)^2 \quad (32)$$

*In high-energy regions, $C[E_K(x_t, t)] < \mathbf{tr}(Cov[S_K(x_t, t)])$ holds when the system energy $\mathcal{E}(x_t)$ is positively related to at least one element of the score $\nabla\mathcal{E}(x_t)$.*

***Proof.*** We split the proof into two parts: low-energy region and high-energy one. The proof in the low-energy region requires only the aforementioned sub-Gaussianess and bounded assumptions, while the one in the high-energy region requires an additional constraint which will be clarified later. Review that $S_K$ can be expressed as an importance-weighted estimator as follows:

$$S_K(x_t, t) = \frac{\frac{1}{K}\sum_{i=1}^K\nabla\exp(-\mathcal{E}(x_{0|t}^{(i)}))}{\frac{1}{K}\sum_{i=1}^K\exp(-\mathcal{E}(x_{0|t}^{(i)}))} \quad (33)$$

Let $\|\nabla\exp(-\mathcal{E}(x_{0|t}^{(i)}))\| \leq M$, where $M > 0$. Since a bounded variable is sub-Gaussian, this assumption resembles a sub-Gaussianess assumption of $\|\nabla\exp(-\mathcal{E}(x_{0|t}^{(i)}))\|$. Then each element of $\|\nabla\exp(-\mathcal{E}(x_{0|t}^{(i)}))\|$, i.e. $\nabla\exp(-\mathcal{E}(x_{0|t}^{(i)}))[j]$, is bounded by $M$. And therefore $\nabla\exp(-\mathcal{E}(x_{0|t}^{(i)}))[j]$ are sub-Gaussian.

**In low-energy regions.** $\exp(-\mathcal{E}(x))$ is concentrated away from 0 as $\mathcal{E}(x)$ is small. Then, there exists a constant $c$ such that $\exp(-\mathcal{E}(x_{0|t}^{(i)})) \geq c > 0$ and thus for each element $j = 1, .., d$:

$$\|S_K(x_t, t)[j]\| = \left\|\frac{\frac{1}{K}\sum_{i=1}^K\nabla\exp(-\mathcal{E}(x_{0|t}^{(i)}))[j]}{\frac{1}{K}\sum_{i=1}^K\exp(-\mathcal{E}(x_{0|t}^{(i)}))}\right\| \quad (34)$$

$$\leq \frac{\|\sum_{i=1}^K\nabla\exp(-\mathcal{E}(x_{0|t}^{(i)}))[j]\|}{Kc} \leq M/c \quad (35)$$

therefore, the $j^{th}$ element of $S_K$, i.e.$S_K[j]$, is bounded by $M/c$, suggesting it is sub-Gaussian. While Inequality 21 can be expressed as

$$\sqrt{\sum_{j=1}^d(\mathcal{S}_K(x_t, t)[j] - S_t(x_t)[j])^2} \leq \frac{2\sqrt{2v_{0t}(x_t)\log(\frac{2}{\delta})}(1 + \|\nabla\mathcal{E}_t(x_t)\|)}{m_t(x_t)\sqrt{K}} \quad (36)$$

We can roughly derive a bound elementwisely

$$|\mathcal{S}_K(x_t, t)[j] - S_t(x_t)[j]| \leq \frac{2\sqrt{2v_{0t}(x_t)\log(\frac{2}{\delta})}(1 + \|\nabla\mathcal{E}_t(x_t)\|)}{m_t(x_t)\sqrt{Kd}} \quad (37)$$

which suggests that we can approximate the variance of $S_K(x_t, t)[j]$ by leveraging its sub-Gaussianess

$$\text{Var}(S_K(x_t, t)[j]) \approx \frac{4v_{0t}(x_t)(1 + \|\nabla\mathcal{E}_t(x_t)\|)^2}{m_t^2(x_t)Kd} \quad (38)$$

Therefore, according to Equation 25 we can derive that

$$\mathbf{tr}(\text{Cov}[S_K(x_t, t)]) = \sum_{j=1}^d \text{Var}[S_K(x_t, t)[j]] \quad (39)$$

$$= \frac{4v_{0t}(x_t)(1 + \|\nabla\mathcal{E}_t(x_t)\|)^2}{m_t^2(x_t)K} \quad (40)$$

$$= 4(1 + \|\nabla\mathcal{E}_t(x_t)\|)^2\text{Var}[E_K(x_t, t)] \quad (41)$$

**In high-energy region.** we assume that there exists a direction with a large norm pointing to low energy regions, i.e. $\exists j$ such that $\mathcal{E}(x)$ are positively related to $\nabla\mathcal{E}(x)[j]$. According to Section 9.2 in Owen (2023), the asymptotic variance of a self-normalized importance sampling estimator is given by:

$$\mu = \mathbb{E}_q[f(X)] \quad (42)$$

$$\tilde{\mu}_q = \frac{\sum_{i=1}^K w_i f_i}{\sum_{i=1}^K w_i} \quad (43)$$

$$\text{Var}(\tilde{\mu}_q) \approx \frac{1}{K}\mathbb{E}_q[w(X)]^{-2}\mathbb{E}_q[w(X)^2(f(X) - \mu)^2] \quad (44)$$

By substituting $\tilde{\mu}_q = S_K(x_t, t)[j]$, $f(X) = -\nabla\mathcal{E}(X)[j]$, $w(X) = \exp(-\mathcal{E}(X))$, $q = N(x; x_t, \sigma_t^2)$, $\mathbb{E}_q[w(X)] = m_t(x_t)$ and $\mathbb{E}_q[w^2(X)] = v_{0t}(x_t) + m_t^2(x_t)$, as well as using $w(X)$ and $f(X)$ are positive related, we have:

$$\text{Var}[S_K(x_t, t)[j]] \geq \frac{1}{K}\mathbb{E}_q[w(X)]^{-2}\mathbb{E}_q[w^2(X)]\mathbb{E}_q[(f(X) - \mu)^2] \quad (45)$$

$$= \frac{v_{0t}(x_t) + m_t^2(x_t)}{m_t^2(x_t)K}\text{Var}_q[\nabla\mathcal{E}(x)[j]] \quad (46)$$

$$(47)$$

Therefore, if we further have a large variance over the system score at this region, i.e. $\text{Var}_q[\nabla\mathcal{E}(x)[j]] > 1$, then we have

$$\text{Var}[S_K(x_t, t)[j]] > \frac{v_{0t}(x_t)}{m_t^2(x_t)K} = \text{Var}[E_K(x_t, t)] \quad (48)$$

and thus $\mathbf{tr}(\text{Cov}[S_K(x_t, t)]) > \text{Var}[E_K(x_t, t)]$ holds. $\qquad\square$

## C  IDEAL BOOTSTRAP ESTIMATOR ≡ RECURSIVE ESTIMATOR

In this section, we will show that an ideal bootstrap estimator can polynomially reduce both the variance and bias w.r.t. number of MC samples $K$. Let a local bootstrapping trajectory $u \to s \to t$, with $u < s < t$. Suppose we have access to $\mathcal{E}_u$. Let $E_{\tilde{\theta}}(x_s, s)$ is learned from targetting an estimator bootstrapped from $\mathcal{E}_u$ to approximate $\mathcal{E}_s$. Suppose $E_{\tilde{\theta}}(x_s, s)$ is ideal, *i.e.* $E_{\tilde{\theta}}(x_s, s) \equiv E_K(x_s, s, u; \mathcal{E}_u)$. The bootstrap energy estimator (Eq.15) at $t$ from $s$ can be written as follows:

$$E_K(x_t, t, s; \tilde{\theta}) = -\log\frac{1}{K}\sum_{i=1}^K \exp(-E_{\tilde{\theta}}(x_{s|t}^{(i)}, s)), \quad x_{s|t}^{(i)} \sim \mathcal{N}(x; x_t, (\sigma_t^2 - \sigma_s^2)I) \quad (49)$$

By plugging $E_{\tilde{\theta}}(x_{s|t}^{(i)}, s) = E_K(x_{s|t}^{(i)}, s, u; \mathcal{E}_u)$ into the above equation, we have

$$E_K(x_t, t, s; \tilde{\theta}) = -\log \frac{1}{K} \sum_{i=1}^{K} \exp(-E_K(x_{s|t}^{(i)}, s, u; \mathcal{E}_u)) \tag{50}$$

$$= -\log \frac{1}{K} \sum_{i=1}^{K} \frac{1}{K} \sum_{j=1}^{K} \exp(-\mathcal{E}_u(x_{u|s}^{(ij)})), \quad x_{u|s}^{(ij)} \sim \mathcal{N}(x; x_s^{(i)}, (\sigma_s^2 - \sigma_u^2)I) \tag{51}$$

where $x_{u|s}^{(ij)} \sim \mathcal{N}(x; x_s^{(i)}, (\sigma_s^2 - \sigma_u^2)I)$. Notice that $x_{s|t}^{(i)} = x_t + \sqrt{\sigma_t^2 - \sigma_s^2}\epsilon_s^{(i)}$, $x_{u|s}^{(ij)} = x_s^{(i)} + \sqrt{\sigma_s^2 - \sigma_u^2}\epsilon_u^{(ij)}$, and $\epsilon_s^{(j)}$ and $\epsilon_u^{(ij)}$ are independent, we can combine these Gaussian-noise injection:

$$E_K(x_t, t, s; \tilde{\theta}) = -\log \frac{1}{K^2} \sum_{i=1}^{K} \sum_{j=1}^{K} \exp(-\mathcal{E}_u(x_{u|t}^{(ij)})) = E_{K^2}(x_t, t, u; \mathcal{E}_u) \tag{52}$$

where $x_{u|t}^{(ij)} \sim \mathcal{N}(x; x_t, (\sigma_t^2 - \sigma_u^2)I)$. Therefore, the ideal bootstrapping at $t$ from $s$ over this local trajectory ($u \to s \to t$) is equivalent to using quadratically more samples compared with estimating $t$ directly from $u$. With initial condition $\mathcal{E}_0 = \mathcal{E}$ and $\sigma_0 = 0$, we can simply show that: Given a bootstrap trajectory $\{s_i\}_{i=0}^n$ where $s_0 = 0$, $s_n = 1$. For any $t \in [s_i, s_{i+1}]$ and any $s \in [s_{i-1}, s_i]$, with $i = 0, ..., n - 1$, the **ideal** bootstrap estimator is equivalent to

$$E_K(x_t, t, s; \tilde{\theta}) \equiv E_{K^{(i+1)}}(x_t, t) \tag{53}$$

which polynomially reduces the variance and bias of the estimator. In other words, the ideal bootstrap estimator is equivalent to recursively approximating $\mathcal{E}_s$ with $E_K(x_s, s)$ utill the initial condition $\mathcal{E}$.

For simplification in Appendix D, we term the ideal bootstrap estimator with bootstrapping $n$ times as the **Sequential**$(n)$ **Estimator**, $E^{\mathrm{Seq}(n)}$, *i.e.*

$$E_K^{\mathrm{Seq}(n)}(x_t, t) := E_{K^{n+1}}(x_t, t) \tag{54}$$

## D  PROOF OF PROPOSITION 3

**Proposition 3** *Given a bootstrap trajectory $\{s_i\}_{i=0}^n$ such that $\sigma_{s_i}^2 - \sigma_{s_{i-1}}^2 \leq \kappa$, where $s_0 = 0$, $s_n = 1$ and $\kappa > 0$ is a small constant. Suppose $E_\theta$ is incrementally optimized from $t \in [0, 1]$ as follows: if $t \in [s_i, s_{i+1}]$, $E_\theta(x_t, t)$ targets an energy estimator bootstrapped from $\forall s \in [s_{i-1}, s_i]$ using Eq. 15. For $\forall 0 \leq i \leq n$ and $\forall s \in [s_{i-1}, s_i]$, the variance of the bootstrap energy estimator is given by*

$$Var[E_K(x_t, t, s; \theta)] = \frac{v_{st}(x_t)}{v_{0t}(x_t)} Var[E_K(x_t, t)] \tag{55}$$

*and the bias of $E_K(x_t, t, s; \theta)$ is given by*

$$\mathrm{Bias}[E_K(x_t, t, s; \theta)] = \frac{v_{0t}(x_t)}{2m_t^2(x_t)K^{i+1}} + \sum_{j=1}^{i} \frac{v_{0s_j}(x_t)}{2m_{s_j}^2(x_t)K^j}. \tag{56}$$

*where $m_z(x_z) = \exp(-\mathcal{E}_z(x_z))$ and $v_{yz}(x_z) = \mathrm{Var}_{\mathcal{N}(x; x_z, (\sigma_z^2 - \sigma_y^2)I)}[\exp(-\mathcal{E}_y(x))]$ for $\forall 0 \leq y < z \leq 1$.*

**Proof.** The variance of $E_K(x_t, t, s; \theta)$ can be simply derived by leveraging the variance of a sub-Gaussian random variable similar to Equation 25. While the entire proof for bias of $E_K(x_t, t, s; \theta)$ is organized as follows:

1. we first show the bias of Bootstrap(1) estimator, which is bootstrapped from the system energy

2. we then show the bias of Bootstrap($n$) estimator, which is bootstrapped from a lower level noise convolved energy recursively, by induction.

## D.1 BOOTSTRAP(1) ESTIMATOR

The Sequential estimator and Bootstrap(1) estimator are defined by:

$$E_K^{\text{Seq}(1)}(x_t, t) := -\log \frac{1}{K} \sum_{i=1}^{K} \exp(-E_K(x_{s|t}^{(i)}, s)), \quad x_{s|t}^{(i)} \sim \mathcal{N}(x; x_t, (\sigma_t^2 - \sigma_s^2)I) \quad (57)$$

$$= -\log \frac{1}{K^2} \sum_{i=1}^{K} \sum_{j=1}^{K} \exp(-\mathcal{E}(x_{0|t}^{(ij)})), \quad x_{0|t}^{(ij)} \sim \mathcal{N}(x; x_t, \sigma_t^2 I) \quad (58)$$

$$E_K^{B(1)}(x_t, t, s; \theta) := -\log \frac{1}{K} \sum_{i=1}^{K} \exp(-E_\theta(x_{s|t}^{(i)}, s)), \quad x_{s|t}^{(i)} \sim \mathcal{N}(x; x_t, (\sigma_t^2 - \sigma_s^2)I) \quad (59)$$

The mean and variance of a Sequential estimator can be derived by considering it as the MC estimator with $K^2$ samples:

$$\mathbb{E}[E_K^{\text{Seq}(1)}(x_t, t)] = \mathcal{E}_t(x_t) + \frac{v_{0t}(x_t)}{2m_{0t}^2(x_t)K^2} \quad \text{and} \quad \text{Var}(E_K^{\text{Seq}(1)}(x_t, t)) = \frac{v_{0t}(x_t)}{m_{0t}^2(x_t)K^2} \quad (60)$$

While an optimal network obtained by targeting the original MC energy estimator 9 at $s$ is [2] :

$$E_{\theta^*}(x_s, s) = \mathbb{E}[E_K(x_s, s)] = -\log m_s(x_s) + \frac{v_{0s}(x_s)}{2m_s^2(x_s)K} \quad (61)$$

Then the optimal Bootstrap(1) estimator can be expressed as:

$$E_K^{B(1)}(x_t, t, s; \theta^*) = -\log \frac{1}{K} \sum_{i=1}^{K} \exp\left(-\left(-\log m_s(x_{s|t}^{(i)}) + \frac{v_{0s}(x_{s|t}^{(i)})}{2m_s^2(x_{s|t}^{(i)})K}\right)\right) \quad (62)$$

Before linking the Bootstrap estimator and the Sequential one, we provide the following approximation which is useful. Let $a$, $b$ two random variables and $\{a_i\}_{i=1}^K$, $\{b_i\}_{i=1}^K$ are corresponding samples. Assume that $\{b_i\}_{i=1}^K$ are close to 0 and concentrated at $m_b$, while $\{a_i\}_{i=1}^K$ are concentrated at $m_a$, then

$$\log \frac{1}{K} \sum_{i=1}^{K} \exp(-(a_i + b_i)) = \log \frac{1}{K} \left\{ \sum_{i=1}^{K} \exp(-a_i) \left[ \frac{\sum_{i=1}^{K} \exp(-(a_i + b_i))}{\sum_{i=1}^{K} \exp(-a_i)} \right] \right\} \quad (63)$$

$$= \log \frac{1}{K} \sum_{i=1}^{K} \exp(-a_i) + \log \frac{\sum_{i=1}^{K} \exp(-(a_i + b_i))}{\sum_{i=1}^{K} \exp(-a_i)} \quad (64)$$

$$\approx \log \frac{1}{K} \sum_{i=1}^{K} \exp(-a_i) + \log \frac{\sum_{i=1}^{K} \exp(-a_i)(1 - b_i)}{\sum_{i=1}^{K} \exp(-a_i)} \quad (65)$$

$$= \log \frac{1}{K} \sum_{i=1}^{K} \exp(-a_i) + \log \left(1 - \frac{\sum_{i=1}^{K} \exp(-a_i)b_i}{\sum_{i=1}^{K} \exp(-a_i)}\right) \quad (66)$$

$$\approx \log \frac{1}{K} \sum_{i=1}^{K} \exp(-a_i) - \frac{\sum_{i=1}^{K} \exp(-a_i)b_i}{\sum_{i=1}^{K} \exp(-a_i)} \quad (67)$$

$$\approx \log \frac{1}{K} \sum_{i=1}^{K} \exp(-a_i) - m_b \quad (68)$$

where Approximation applies a first order Taylor expansion of $e^x \approx 1 + x$ around $x = 0$ since $\{b_i\}_{i=1}^K$ are close to 0; while Approximation uses $\log(1 + x) \approx x$ under the same assumption. Notice

---

[2]We consider minimizing the $L_2$-norm, i.e. $\theta^* = \arg\min_\theta \mathbb{E}_{x_0, t}[\|E_\theta(x_t, t) - E_K(x_t, t)\|^2]$. Since the target, $E_K$, is noisy, the optimal outputs are given by the expectation, i.e. $E_\theta^* = \mathbb{E}[E_K]$.

that when $K$ is large and $\sigma_t^2 - \sigma_s^2 \le \kappa$ is small , $\{\frac{v_{0s}(x_{s|t}^{(i)})}{2m_s^2(x_{s|t}^{(i)})K}\}_{i=1}^K$ are close to 0 and concentrated at $\frac{v_{0s}(x_t)}{2m_s^2(x_t)K}$. Therefore, by plugging them into Equation 68, Equation 62 can be approximated by

$$E_K^{B(1)}(x_t, t, s; \theta^*) \approx -\log \frac{1}{K} \sum_{i=1}^K m_s(x_{s|t}^{(i)}) + \frac{v_{0s}(x_t)}{2m_s^2(x_t)K} \tag{69}$$

When $K$ is large and $\sigma_s^2$ is small, the bias and variance of $E_K(x_{s|t}^{(i)}, s)$ are small, then we have

$$-\log \frac{1}{K} \sum_{i=1}^K m_s(x_{s|t}^{(i)}) \approx -\log \frac{1}{K} \sum_{i=1}^K E_K(x_{s|t}^{(i)}, s) = E_K^{\text{Seq}(1)}(x_t, t) \tag{70}$$

Therefore, the optimal Bootstrap estimator can be approximated as follows:

$$E_K^{B(1)}(x_t, t, s; \theta^*) \approx E_K^{\text{Seq}(1)}(x_t, t) + \frac{v_{0s}(x_t)}{2m_s^2(x_t)K} \tag{71}$$

where its mean and variance depend on those of the Sequential estimator (60):

$$\mathbb{E}[E_K^{B(1)}(x_t, t, s; \theta^*)] = \mathcal{E}_t(x_t) + \frac{v_{0t}(x_t)}{2m_t^2(x_t)K^2} + \frac{v_{0s}(x_t)}{2m_s^2(x_t)K} \tag{72}$$

$$\text{Var}[E_K^{B(1)}(x_t, t, s; \theta^*)] = \frac{v_{0t}(x_t)}{m_t^2(x_t)K^2} \tag{73}$$

## D.2 BOOTSTRAP($n$) ESTIMATOR

Given a bootstrap trajectory $\{s_i\}_{i=1}^n$ where $s_0 = 0$ and $s_n = s$, and $E_\theta$ is well learned at $[0, s]$. Let the energy network be optimal for $u \le s_n$ by learning a sequence of Bootstrap($i$) energy estimators ($i \le n$). Then the optimal value of $E_\theta(x_s, s)$ is given by $\mathbb{E}[E_K^{B(n-1)}(x_s, s)]$. We are going to show the variance of a Bootstrap($n$) estimator by induction. Suppose we have:

$$E_{\theta^*}(x_s, s) = \mathcal{E}_s(x_s) + \sum_{j=1}^n \frac{v_{0s_j}(x_s)}{2m_{s_j}^2(x_s)K^j} \tag{74}$$

$$= \mathbb{E}[E_K^{\text{Seq}(n-1)}(x_s, s)] + \sum_{j=1}^{n-1} \frac{v_{0s_j}(x_s)}{2m_{s_j}^2(x_s)K^j} \tag{75}$$

Then for any $t \in (s, 1]$, the learning target of $E_\theta(x_t, t)$ is bootstrapped from $s_n = s$,

$$E_K^{B(n)}(x_t, t) = -\log \frac{1}{K} \sum_{i=1}^K \exp(-E_{\theta^*}(x_{s|t}^{(i)}, s)), \quad x_{s|t}^{(i)} \sim \mathcal{N}(x; x_t, (\sigma_t^2 - \sigma_s^2)I) \tag{76}$$

$$= -\log \frac{1}{K} \sum_{i=1}^K \exp\left(-\mathbb{E}[E_K^{\text{Seq}(n-1)}(x_{s|t}^{(i)}, s)] - \sum_{j=1}^{n-1} \frac{v_{0s_j}(x_{s|t}^{(i)})}{2m_{s_j}^2(x_{s|t}^{(i)})K^j}\right) \tag{77}$$

Assume that $\sigma_t^2 - \sigma_s^2$ is small and $K$ is large, then we can apply Approximation 68 and have

$$E_K^{B(n)}(x_t, t) \approx -\log \frac{1}{K} \sum_{i=1}^K \exp\left(-\mathbb{E}[E_K^{\text{Seq}(n-1)}(x_{s|t}^{(i)}, s)]\right) + \sum_{j=1}^{n-1} \frac{v_{0s_j}(x_t)}{2m_{s_j}^2(x_t)K^j} \tag{78}$$

The first term in the RHS of the above equation can be further simplified as [3] :

$$-\log \frac{1}{K} \sum_{i=1}^{K} \exp\left(-\mathbb{E}[E_K^{\text{Seq}(n-1)}(x_{s|t}^{(i)}, s)]\right) \tag{79}$$

$$\approx \mathbb{E}\left[-\log \frac{1}{K} \sum_{i=1}^{K} \exp\left(-E_K^{\text{Seq}(n-1)}(x_{s|t}^{(i)}, s)\right)\right] + \sum_{i=1}^{K} \frac{\exp\left(-\mathbb{E}[E_K^{\text{Seq}(n-1)}(x_{s|t}^{(i)}, s)]\right) \text{Var}[E_K^{\text{Seq}(n-1)}(x_{s|t}^{(i)}, s)]}{2 \sum_{j=1}^{K} \exp\left(-\mathbb{E}[E_K^{\text{Seq}(n-1)}(x_{s|t}^{(i)}, s)]\right)} \tag{80}$$

$$\approx \mathbb{E}\left[-\log \frac{1}{K} \sum_{i=1}^{K} \exp\left(-E_K^{\text{Seq}(n-1)}(x_{s|t}^{(i)}, s)\right)\right] + \frac{\text{Var}[E_K^{\text{Seq}(n-1)}(x_{s|t}^{(i)}, s)]}{2} \tag{81}$$

$$= \mathbb{E}_{\epsilon^{(j)}}\left[-\log \frac{1}{K^{n+1}} \sum_{i=1}^{K} \sum_{j=1}^{K^n} \exp\left(-\mathcal{E}\left(x_{s|t}^{(i)} + \epsilon^{(j)}\right)\right)\right] + \frac{v_{0s}(x_t)}{2m_s^2(x_t)K^n} \tag{82}$$

where $\epsilon^j \overset{\text{i.i.d.}}{\sim} \mathcal{N}(0, I)$. Therefore, Eq. 78 can be simplified as

$$E_K^{B(n)}(x_t, t) = \mathbb{E}_{\epsilon^{(j)}}\left[-\log \frac{1}{K^{n+1}} \sum_{i=1}^{K} \sum_{j=1}^{K^n} \exp\left(-\mathcal{E}\left(x_{s|t}^{(i)} + \epsilon^{(j)}\right)\right)\right] + \sum_{j=1}^{n} \frac{v_{0s_j}(x_t)}{2m_{s_j}^2(x_t)K^j} \tag{83}$$

Taking the expectation over Eq. 83 yields an expectation of the Sequential($n$) estimator on its RHS, we therefore complete the proof by induction:

$$\mathbb{E}\left[E_K^{B(n)}(x_t, t)\right] = \mathbb{E}_{\epsilon^{(j)}, x_{s|t}^{(i)}}\left[-\log \frac{1}{K^{n+1}} \sum_{i=1}^{K} \sum_{j=1}^{K^n} \exp\left(-\mathcal{E}\left(x_{s|t}^{(i)} + \epsilon^{(j)}\right)\right)\right] + \sum_{j=1}^{n} \frac{v_{0s_j}(x_t)}{2m_{s_j}^2(x_t)K^j} \tag{84}$$

$$= \mathbb{E}\left[E_K^{\text{Seq}(n)}(x_t, t)\right] + \sum_{j=1}^{n} \frac{v_{0s_j}(x_t)}{2m_{s_j}^2(x_t)K^j} \tag{85}$$

$$= \mathcal{E}_t(x_t) + \frac{v_{0t}(x_t)}{2m_t^2(x_t)K^{n+1}} + \sum_{j=1}^{n} \frac{v_{0s_j}(x_t)}{2m_{s_j}^2(x_t)K^j} \tag{86}$$

which suggests that the accumulated bias of a Bootstrap($n$) estimator is given by

$$\text{Bias}[E_K^{B(n)}(x_t, t)] = \frac{v_{0t}(x_t)}{2m_t^2(x_t)K^{n+1}} + \sum_{j=1}^{n} \frac{v_{0s_j}(x_t)}{2m_{s_j}^2(x_t)K^j} \tag{87}$$

$\square$

## E    INCORPORATING SYMMETRY USING NEM

We consider applying NEM and BNEM in physical systems with symmetry constraints like $n$-body system. We prove that our MC energy estimator $E_K$ is $G$-invariant under certain conditions, given in the following Proposition.

**Proposition 4** *Let $G$ be the product group $SE(3) \times \mathbb{S}_n \hookrightarrow O(3n)$ and $p_0$ be a $G$-invariant density in $\mathbb{R}^d$. Then the Monte Carlo energy estimator of $E_K(x_t, t)$ is $G$-invariant if the sampling distribution $x_{0|t} \sim \mathcal{N}(x_{0|t}; x_t, \sigma_t^2)$ is $G$-invariant, i.e.,*

$$\mathcal{N}(x_{0|t}; g \circ x_t, \sigma_t^2) = \mathcal{N}(g^{-1}x_{0|t}; x_t, \sigma_t^2).$$

---

[3]This simplication leverages the fact that, if $X_i$ are i.i.d. bounded sub-Gaussian random variables, then $\log \sum_i \exp(-\mathbb{E}X_i)$ can be approximated by $\mathbb{E}[\log \sum_i \exp(-X_i)] - \frac{1}{2} \sum_i \frac{\exp(-\mathbb{E}X_i)\text{Var}(X_i)}{\sum_j \exp(-\mathbb{E}X_j)}$.

**Proof.** Since $p_0$ is $G$-invariant, then $\mathcal{E}$ is $G$-invariant as well. Let $g \in G$ acts on $x \in \mathbb{R}^d$ where $g \circ x = gx$. Since $x_{0|t}^{(i)} \sim \overline{\mathcal{N}}(x_{0|t}; x_t, \sigma_t^2)$ is equivalent to $g \circ x_{0|t}^{(i)} \sim \overline{\mathcal{N}}(x_{0|t}; g \circ x_t, \sigma_t^2)$. Then we have

$$E_K(g \circ x_t, t) = -\log \frac{1}{K} \sum_{i=1}^{K} \exp(-\mathcal{E}(g \circ x_{0|t}^{(i)})) \tag{88}$$

$$= -\log \frac{1}{K} \sum_{i=1}^{K} \exp(-\mathcal{E}(x_{0|t}^{(i)})) = E_K(x_t, t) \tag{89}$$

$$x_{(0|t)}^{(i)} \sim \overline{\mathcal{N}}(x_{0|t}; x_t, \sigma_t^2) \tag{90}$$

Therefore, $E_K$ is invariant to $G = \text{SE}(3) \times \mathbb{S}_n$. $\qquad\square$

Furthermore, $E_K(x_t, t, s; \phi)$ is obtained by applying a learned energy network, which is $G$-invariant, to the analogous process and therefore is $G$-invariant as well.

# F    GENERALIZING NEM

This section generalizes NEM in two ways: we first generalize the SDE setting, by considering a broader family of SDEs applied to sampling from Boltzmann distribution; then we generalize the MC energy estimator by viewing it as an importance-weighted estimator.

## F.1    NEM FOR GENERAL SDES

Diffusion models can be generalized to any SDEs as $dx_t = f(x_t, t)dt + g(t)dw_t$, where $t \geq 0$ and $w_t$ is a Brownian motion. Particularly, we consider $f(x, t) := -\alpha(t)x$, *i.e.*

$$dx_t = -\alpha(t)x_t dt + g(t)dw_t \tag{91}$$

Then the marginal of the above SDE can be analytically derived as:

$$x_t = \beta(t)x_0 + \beta(t)\sqrt{\int_0^t (g(s)\beta(s))^2 ds}\,\epsilon, \quad \beta(t) := e^{-\int_0^t \alpha(s)ds} \tag{92}$$

where $\epsilon \sim \mathcal{N}(0, I)$. For example, when $g(t) = \sqrt{\bar{\beta}(t)}$ and $\alpha(t) = \frac{1}{2}\bar{\beta}(t)$, where $\bar{\beta}(t)$ is a monotonic function (*e.g.* linear) increasing from $\bar{\beta}_{\min}$ to $\bar{\beta}_{\max}$, the above SDE resembles a Variance Preserving (VP) process (Song et al., 2020). In DMs, VP can be a favor since it constrains the magnitude of noisy data across $t$; while a VE process doesn't, and the magnitude of data can explode as the noise explodes. Therefore, we aim to discover whether any SDEs rather than VE can be better by generalizing NEM and DEM to general SDEs.

In this work, we provide a solution for general SDEs (91) rather than a VE SDE. For simplification, we exchangeably use $\beta(t)$ and $\beta_t$. Given a SDE as Equation 91 for any integrable functions $\alpha$ and $g$, we can first derive its marginal as Equation 92, which can be expressed as:

$$\beta_t^{-1} x_t = x_0 + \sqrt{\int_0^t (g(s)\beta(s))^2 ds}\,\epsilon \tag{93}$$

Therefore, by defining $y_t = \beta_t^{-1} x_t$ we have $y_0 = x_0$ and therefore:

$$y_t = y_0 + \sqrt{\int_0^t (g(s)\beta(s))^2 ds}\,\epsilon \tag{94}$$

which resembles a VE SDE with noise schedule $\tilde{\sigma}^2(t) = \int_0^t (g(s)\beta(s))^2 ds$. We can also derive this by changing variables:

$$dy_t = (\beta^{-1}(t))' x_t dt + \beta^{-1}(t) dx_t \tag{95}$$

$$= \beta^{-1}(t)\alpha(t)x_t dt + \beta^{-1}(t)(-\alpha(t)x_t dt + g(t)dw_t) \tag{96}$$

$$= \beta^{-1}(t)g(t)dw_t \tag{97}$$

which also leads to Equation 94. Let $\tilde{p}_t$ be the marginal distribution of $y_t$ and $p_t$ the marginal distribution of $x_t$, with $y_{0|t}^{(i)} \sim \mathcal{N}(y; y_t, \tilde{\sigma}_t^2 I)$ we have

$$\tilde{p}_t(y_t) \propto \int \exp(-\mathcal{E}(y))\mathcal{N}(y_t; y, \tilde{\sigma}_t^2 I)dy \tag{98}$$

$$\tilde{S}_t(y_t) = \nabla_{y_t} \log \tilde{p}_t(y_t) \approx \nabla_{y_t} \log \sum_{i=1}^{K} \exp(-\mathcal{E}(y_{0|t}^{(i)})) \tag{99}$$

$$\tilde{\mathcal{E}}_t(y_t) \approx -\log \frac{1}{K} \sum_{i=1}^{K} \exp(-\mathcal{E}(y_{0|t}^{(i)})) \tag{100}$$

Therefore, we can learn scores and energies of $y_t$ simply by following DEM and NEM for VE SDEs. Then for sampling, we can simulate the reverse SDE of $y_t$ and eventually, we have $x_0 = y_0$.

Instead, we can also learn energies and scores of $x_t$. By changing the variable, we can have

$$p_t(x_t) = \beta_t^{-1} \tilde{p}_t(\beta_t^{-1} x_t) = \beta_t^{-1} \tilde{p}_t(y_t) \tag{101}$$

$$S_t(x_t) = \beta_t^{-1} \tilde{S}_t(\beta_t^{-1} x_t) = \beta_t^{-1} \tilde{S}_t(y_t) \tag{102}$$

which provides us the energy and score estimator for $x_t$:

$$\mathcal{E}_t(x_t) \approx -\log \beta_t^{-1} \frac{1}{K} \sum_{i=1}^{K} \exp(-\mathcal{E}(x_{0|t}^{(i)})) \tag{103}$$

$$S_t(x_t) \approx \beta_t^{-1} \nabla_{x_t} \log \sum_{i=1}^{K} \exp(-\mathcal{E}(x_{0|t}^{(i)})) \tag{104}$$

$$x_{0|t}^{(i)} \sim \mathcal{N}(x; \beta_t^{-1} x_t, \tilde{\sigma}^2(t)I) \tag{105}$$

Typically, $\alpha$ is a non-negative function, resulting in $\beta(t)$ decreasing from 1 and can be close to 0 when $t$ is large. Therefore, the above equations realize that even though both the energies and scores for a general SDE can be estimated, the estimators are not reliable at large $t$ since $\beta_t^{-1}$ can be extremely large; while the SDE of $y_t$ (97) indicates that this equivalent VE SDE is scaled by $\beta_t^{-1}$, resulting that the variance of $y_t$ at large $t$ can be extremely large and requires much more MC samples for a reliable estimator. This issue can be a bottleneck of generalizing DEM, NEM, and BNEM to other SDE settings, therefore developing more reliable estimators for both scores and energies is of interest in future work.

## F.2 MC Energy Estimator as an Importance-Weighted Estimator

As for any SDEs, we can convert the modeling task to a VE process by changing variables, we stick to considering NEM with a VE process. Remember that the MC energy estimator aims to approximate the noised energy given by Eq. 6, which can be rewritten as:

$$\mathcal{E}_t(x_t) = -\log \int \exp(-\mathcal{E}(x))\mathcal{N}(x_t; x, \sigma_t^2 I)dx \tag{106}$$

$$= -\log \int \exp(-\mathcal{E}(x)) \frac{\mathcal{N}(x_t; x, \sigma_t^2 I)}{q_{0|t}(x|x_t)} q_{0|t}(x|x_t)dx \tag{107}$$

$$= -\log \mathbb{E}_{q_{0|t}(x|x_t)} \left[ \exp(-\mathcal{E}(x)) \frac{\mathcal{N}(x_t; x, \sigma_t^2 I)}{q_{0|t}(x|x_t)} \right] \tag{108}$$

The part inside the logarithm of Eq. 108 suggests an Importance Sampling technique for approximation, by using a proposal $q_{0|t}(x|x_t)$. Notice that when choosing a proposal symmetric to the perturbation kernal, $i.e.$ $q_{0|t}(x|x_t) = \mathcal{N}(x; x_t, \sigma_t^2)$, Eq. 108 resembles the MC energy estimator we discussed in Section 3.3. Therefore, this formulation allows us to develop a better estimator by carefully selecting the proposal $q_{0|t}(x|x_t)$.

However, Owen (2013) shows that to minimize the variance of the IS estimator, the proposal $q_{0|t}(x|x_t)$ should be chosen roughly proportional to $f(x)\mu_{\text{target}}(x)$, where $f(x) = \exp(-\mathcal{E}(x))$ in our case. Finding such a proposal is challenging in high-dimension space or with a multimodal $\mu_{\text{target}}$. A potential remedy can be leveraging Annealed Importance Sampling (AIS; Neal (2001)).

## G  MEMORY-EFFICIENT NEM

Differentiating the energy network to get denoising scores in (B)NEM raises additional computations compared with iDEM, which usually twices the computation of forwarding a neural network and can introduce memory overhead due to saving the computational graph. The former issue can be simply solved by reducing the number of integration steps to a half. To solve the latter memory issue, we propose a Memory-efficient NEM by revisiting the Tweedie's formula (Efron, 2011). Given a VE noising process, $dx_t = g(t)dw_t$, where $w_t$ is Brownian motion and $\sigma_t^2 := \int_{s=0}^t g(s)^2 ds$, the Tweedie's formula can be written as:

$$\nabla \log p_t(x_t) = \frac{\mathbb{E}[x_0|x_t] - x_t}{\sigma_t^2} \tag{109}$$

$$\mathbb{E}[x_0|x_t] = \int x_0 p(x_0|x_t) dx_0 \tag{110}$$

$$= \int x_0 \frac{p(x_t|x_0)p_0(x_0)}{p_t(x_t)} dx_0 \tag{111}$$

By revisiting Eq. 5, it's noticable that the noised energy, $\mathcal{E}_t$ (Eq. 6), shares the same partition function as $\mathcal{E}$, *i.e.* $\int \exp(-\mathcal{E}_t(x))dx = \int \exp(-\mathcal{E}(x))dx, \forall t \in [0, 1]$. Hence, Eq. 111 can be simplified as follows, which further suggests a MC estimator for the denoising score with no require of differentiation

$$\mathbb{E}[x_0|x_t] = \int x_0 \frac{\mathcal{N}(x_t; x_0, \sigma_t^2 I) \exp(-\mathcal{E}(x_0))}{\exp(-\mathcal{E}_t(x_t))} dx_0 \tag{112}$$

$$\approx \exp\left(-\left(\mathcal{E}(x_{0|t}) - \mathcal{E}_t(x_t)\right)\right) x_{0|t} \tag{113}$$

where $x_{0|t} \sim \mathcal{N}(x; x_t, \sigma_t^2 I)$. Given learned noised energy, we can approximate this denoiser estimator as follows:

$$D_\theta(x_t, t) := \exp\left(-\left(\mathcal{E}(x_{0|t}) - E_\theta(x_t, t)\right)\right) x_{0|t} \tag{114}$$

$$\tilde{D}_\theta(x_t, t) := \exp\left(-\left(E_\theta(x_{0|t}, 0) - E_\theta(x_t, t)\right)\right) x_{0|t} \tag{115}$$

where we can alternatively use $D_\theta$ or $\tilde{D}_\theta$ according to the accessability of clean energy $\mathcal{E}$, the relative computation between $\mathcal{E}(x)$ and $E_\theta(x, t)$, and the accuracy of $E_\theta(x, 0)$.

## H  TWEEDEM: A MIDDLE POINT BETWEEN DENOISING ENERGY MATCHING AND ENERGY FLOW MATCHING THROUGH TWEEDIE'S FORMULA

In this supplementary work, we propose TWEEDIE DEM (TweeDEM), by leveraging the Tweedie's formula (Efron, 2011) into DEM, *i.e.* $\nabla_x \log p_t(x) = \mathbb{E}_{p(x_0|x_t)}\left[\frac{x_0 - x_t}{\sigma_t^2}\right]$. Surprisingly, TweeDEM can be equivalent to the iEFM-VE proposed by Woo & Ahn (2024), which is a variant of iDEM corresponding to another family of generative model, flow matching.

We first derive an MC denoiser estimator, *i.e.* the expected clean data given a noised data $x_t$ at $t$

$$\mathbb{E}[x_0|x_t] = \int x_0 p(x_0|x_t) dx_0 \tag{116}$$

$$= \int x_0 \frac{q_t(x_t|x_0)p_0(x_0)}{p_t(x_t)} dx_0 \tag{117}$$

$$= \int x_0 \frac{\mathcal{N}(x_t; x_0, \sigma_t^2 I) \exp(-\mathcal{E}(x_0))}{\exp(-\mathcal{E}_t(x_t))} dx_0 \tag{118}$$

where the numerator can be estimated by an MC estimator $\mathbb{E}_{\mathcal{N}(x_t, \sigma_t^2 I)}[x \exp(-\mathcal{E}(x))]$ and the denominator can be estimated by another similar MC estimator $\mathbb{E}_{\mathcal{N}(x_t, \sigma_t^2 I)}[\exp(-\mathcal{E}(x))]$, suggesting

Table 2: Comparison between DEM, DDM, and TweeDEM.

| | Score estimator: $\sum_i w_i s_i$, with $w_i = Softmax(\tilde{w})[i]$ | | | |
|---|---|---|---|---|
| Sampler↓ Components→ | Weight Type | $\tilde{w}_i$ | Score Type | $s_i$ |
| DEM(Akhound-Sadegh et al., 2024) | System Energy | $\exp(-\mathcal{E}(x_{0|t}^{(i)}))$ | System Score | $-\nabla\mathcal{E}(x_{0|t}^{(i)}))$ |
| DDM(Karras et al., 2022) | Gaussian Density | $\mathcal{N}(x_{0|t}^{(i)}; x_t, \sigma_t^2 I)$ | Gaussian Score | $\nabla\log\mathcal{N}(x_{0|t}^{(i)}; x_t, \sigma_t^2 I)$ |
| TweeDEM | System Energy | $\exp(-\mathcal{E}(x_{0|t}^{(i)}))$ | Gaussian Score | $\nabla\log\mathcal{N}(x_{0|t}^{(i)}; x_t, \sigma_t^2 I)$ |

we can approximate this denoiser through self-normalized importance sampling as follows

$$D_K(x_t, t) := \sum_{i=1}^{K} \frac{\exp(-\mathcal{E}(x_{0|t}^{(i)}))}{\sum_{j=1}^{K}\exp(-\mathcal{E}(x_{0|t}^{(j)}))} x_{0|t}^{(i)} \tag{119}$$

$$= \sum_{i=1}^{K} w_i x_{0|t}^{(i)} \tag{120}$$

where $x_{0|t}^{(i)} \sim \mathcal{N}(x_t, \sigma_t^2 I)$, $w_i$ are the importance weights and $D_K(x_t, t) \approx \mathbb{E}[x_0|x_t]$. Then a new MC score estimator can be constructed by plugging the denoiser estimator $D_K$ into Tweedie's formula

$$\tilde{S}_K(x_t, t) := \sum_{i=1}^{K} w_i \frac{x_{0|t}^{(i)} - x_t}{\sigma_t^2} \tag{121}$$

where $\frac{x_{0|t}^{(i)} - x_t}{\sigma_t^2}$ resembles the vector fields $v_t(x_t)$ in Flow Matching. In another perspective, these vector fields can be seen as scores of Gaussian, *i.e.* $\nabla\log\mathcal{N}(x; x_t, \sigma_t^2 I)$, and therefore $\tilde{S}_K$ is an importance-weighted sum of Gaussian scores while $S_K$ can be expressed as an importance-weighted sum of system scores $-\nabla\mathcal{E}$. In addition, Karras et al. (2022) demonstrates that in Denoising Diffusion Models, the optimal scores are an importance-weighted sum of Gaussian scores, while these importance weights are given by the corresponding Gaussian density, *i.e.* $S_{\text{DM}}(x_t, t) = \sum_i \tilde{w}_i(x_{0|t}^{(i)} - x_t)/\sigma_t^2$ and $\tilde{w}_i \propto \mathcal{N}(x_{0|t}^{(i)}; x_t, \sigma_t^2 I)$. We summarize these three different score estimators in Table 2.

## I    EXPERIMENTAL DETAILS

### I.1    ENERGY FUNCTIONS

**GMM.** A Gaussian Mixture density in 2-dimensional space with 40 modes, which is proposed by Midgley et al. (2023). Each mode in this density is evenly weighted, with identical covariances,

$$\Sigma = \begin{pmatrix} 40 & 0 \\ 0 & 40 \end{pmatrix} \tag{122}$$

and the means $\{\mu_i\}_{i=1}^{40}$ are uniformly sampled from $[-40, 40]^2$, i.e.

$$p_{gmm}(x) = \frac{1}{40} \sum_{i=1}^{40} \mathcal{N}(x; \mu_i, \Sigma) \tag{123}$$

Then its energy is defined by the negative-log-likelihood, i.e.

$$\mathcal{E}^{GMM}(x) = -\log p_{gmm}(x) \tag{124}$$

For evaluation, we sample 1000 data from this GMM with *TORCH.RANDOM.SEED(0)* following Midgley et al. (2023); Akhound-Sadegh et al. (2024) as a test set.

**DW-4.** First introduced by Köhler et al. (2020), the DW-4 dataset describes a system with 4 particles in 2-dimensional space, resulting in a task with dimensionality $d = 8$. The energy of the system is given by the double-well potential based on pairwise Euclidean distances of the particles,

$$\mathcal{E}^{DW}(x) = \frac{1}{2\tau} \sum_{ij} a(d_{ij} - d_0) + b(d_{ij} - d_0)^2 + c(d_{ij} - d_0)^4 \tag{125}$$

where $a$, $b$, $c$ and $d_0$ are chosen design parameters of the system, $\tau$ the dimensionless temperature and $d_{ij} = \|x_i - x_j\|_2$ are Euclidean distance between two particles. Following Akhound-Sadegh et al. (2024), we set $a = 0$, $b = -4$, $c = 0.9$ $d_0 = 4$ and $\tau = 1$, and we use validation and test set from the MCMC samples in Klein et al. (2023a) as the "Ground truth" samples for evaluating.

**LJ-n**. This dataset describes a system consisting of $n$ particles in 3-dimensional space, resulting in a task with dimensionality $d = 3n$. Following Akhound-Sadegh et al. (2024), the energy of the system is given by $\mathcal{E}^{Tot}(x) = \mathcal{E}^{LJ}(x) + c\mathcal{E}^{osc}(x)$ with the Lennard-Jones potential

$$\mathcal{E}^{\text{LJ}}(x) = \frac{\epsilon}{2\tau} \sum_{ij} \left( \left(\frac{r_m}{d_{ij}}\right)^6 - \left(\frac{r_m}{d_{ij}}\right)^{12} \right) \tag{126}$$

and the harmonic potential

$$\mathcal{E}^{osc}(x) = \frac{1}{2} \sum_i \|x_i - x_{COM}\|^2 \tag{127}$$

where $d_{ij} = \|x_i - x_j\|_2$ are Euclidean distance between two particles, $r_m$, $\tau$ and $\epsilon$ are physical constants, $x_{COM}$ refers to the center of mass of the system and $c$ the oscillator scale. We use $r_m = 1$, $\tau = 1$, $\epsilon = 1$ and $c = 0.5$ the same as Akhound-Sadegh et al. (2024). We test our models in LJ-13 and LJ-55, which correspond to $d = 65$ and $d = 165$ respectively. And we use the MCMC samples given by Klein et al. (2023a) as a test set.

## I.2 Evaluation Metrics

**2-Wasserstein distance** $\mathcal{W}_2$. Given empirical samples $\mu$ from the sampler and ground truth samples $\nu$, the 2-Wasserstein distance is defined as:

$$\mathcal{W}_2(\mu, \nu) = \left(\inf_\pi \int \pi(x, y) d^2(x, y) dx dy\right)^{\frac{1}{2}} \tag{128}$$

where $\pi$ is the transport plan with marginals constrained to $\mu$ and $\nu$ respectively. Following Akhound-Sadegh et al. (2024), we use the Hungarian algorithm as implemented in the Python optimal transport package (POT) (Flamary et al., 2021) to solve this optimization for discrete samples with the Euclidean distance $d(x, y) = \|x - y\|_2$. $x - \mathcal{W}_2$ is based on the data and $\mathcal{E} - \mathcal{W}_2$ is based on the corresponding energy.

**Total Variation (TV)**. The total variation measures the dissimilarity between two probability distributions. It quantifies the maximum difference between the probabilities assigned to the same event by two distributions, thereby providing a sense of how distinguishable the distributions are. Given two distribution $P$ and $Q$, with densities $p$ and $q$, over the same sample space $\Omega$, the TV distance is defined as

$$TV(P, Q) = \frac{1}{2} \int_\Omega |p(x) - q(x)| dx \tag{129}$$

Following Akhound-Sadegh et al. (2024), for low-dimensional datasets like GMM, we use 200 bins in each dimension. For larger equivariant datasets, the total variation distance is computed over the distribution of the interatomic distances of the particles.

## I.3 Experiment Settings

We pin the number of reverse SDE integration steps for iDEM, NEM, BNEM and TweeDEM (see Appendix H) as 1000 and the number of MC samples as 1000 in most experiments, except for the ablation studies.

**GMM-40.** For the basic model $f_\theta$, we use an MLP with sinusoidal and positional embeddings which has 3 layers of size 128 as well as positional embeddings of size 128. The replay buffer is set to a maximum length of 10000.

During training, the generated data was in the range $[-1, 1]$ so to calculate the energy it was scaled appropriately by unnormalizing by a factor of 50. Baseline models are trained with a geometric

Table 3: Neural sampler performance comparison for 4 different energy functions. We measured the performance using data Wasserstein-2 distance (x-$\mathcal{W}_2$), Energy Wasserstein-2 distance ($\mathcal{E}$-$\mathcal{W}_2$), and Total Variation (TV). * indicates divergent training. Each sampler is evaluated with 3 random seeds and we report the mean ± standard-deviation for each metric.

| Energy → | GMM-40 ($d=2$) | | | DW-4 ($d=8$) | | | LJ-13 ($d=39$) | | | LJ-55 ($d=165$) | | |
|---|---|---|---|---|---|---|---|---|---|---|---|---|
| Sampler ↓ | x-$\mathcal{W}_2$↓ | $\mathcal{E}$-$\mathcal{W}_2$↓ | TV↓ | x-$\mathcal{W}_2$↓ | $\mathcal{E}$-$\mathcal{W}_2$↓ | TV↓ | x-$\mathcal{W}_2$↓ | $\mathcal{E}$-$\mathcal{W}_2$↓ | TV↓ | x-$\mathcal{W}_2$↓ | $\mathcal{E}$-$\mathcal{W}_2$↓ | TV↓ |
| DDS | $15.04_{\pm2.97}$ | $305.13_{\pm186.06}$ | $0.96_{\pm0.01}$ | $0.82_{\pm0.21}$ | $558.79_{\pm787924.86}$ | $0.38_{\pm0.14}$ | * | * | | | * | |
| PIS | $6.58_{\pm1.68}$ | $79.86_{\pm7.79}$ | $0.95_{\pm0.01}$ | * | * | * | * | * | * | * | * | * |
| FAB | $9.08_{\pm1.41}$ | $47.60_{\pm7.24}$ | $0.79_{\pm0.07}$ | $0.62_{\pm0.02}$ | $112.70_{\pm20.33}$ | $0.38_{\pm0.02}$ | * | * | * | * | * | * |
| iDEM | $8.21_{\pm5.43}$ | $60.49_{\pm70.12}$ | $0.82_{\pm0.03}$ | $0.50_{\pm0.03}$ | $2.80_{\pm1.72}$ | $0.16_{\pm0.01}$ | $0.87_{\pm0.00}$ | $77515.90_{\pm115028.07}$ | $0.06_{\pm0.01}$ | $2.06_{\pm0.04}$ | $169347.94_{\pm260160.98}$ | $0.16_{\pm0.02}$ |
| NEM (ours) | $5.28_{\pm0.89}$ | $44.56_{\pm39.56}$ | $0.91_{\pm0.02}$ | $\mathbf{0.48}_{\pm0.02}$ | $0.85_{\pm0.52}$ | $\mathbf{0.14}_{\pm0.01}$ | $\mathbf{0.87}_{\pm0.01}$ | $5.01_{\pm3.14}$ | $\mathbf{0.03}_{\pm0.00}$ | $1.90_{\pm0.01}$ | $\mathbf{118.58}_{\pm106.63}$ | $\mathbf{0.10}_{\pm0.02}$ |
| BNEM (ours) | $\mathbf{3.66}_{\pm0.30}$ | $\mathbf{1.87}_{\pm1.00}$ | $\mathbf{0.79}_{\pm0.04}$ | $0.49_{\pm0.02}$ | $\mathbf{0.38}_{\pm0.09}$ | $\mathbf{0.14}_{\pm0.01}$ | $\mathbf{0.86}_{\pm0.00}$ | $\mathbf{1.02}_{\pm0.69}$ | $\mathbf{0.03}_{\pm0.00}$ | $\mathbf{1.87}_{\pm0.05}$ | $821060.12_{\pm1397674.81}$ | $0.15_{\pm0.07}$ |

noise schedule with $\sigma_{\min} = 1e-5$, $\sigma_{\max} = 1$; NEM and BNEM are trained with a cosine noise schedule with $\sigma_{\min} = 0.001$ and $\sigma_{\max} = 1$. We use $K = 500$ samples for computing the Bootstrap energy estimator $E_K^B$. We clip the norm of $S_K$, $s_\theta$ and $\nabla E_\theta$ to 70 during training and sampling. The variance controller for BNEM is set to be $\beta = 0.2$. All models are trained with a learning rate of $5e-4$.

**DW-4.** All models use an EGNN with 3 message-passing layers and a 2-hidden layer MLP of size 128. All models are trained with a geometric noise schedule with $\sigma_{\min} = 1e-5$, $\sigma_{\max} = 3$ and a learning rate of $1e-3$ for computing $S_K$ and $E_K$. We use $K = 500$ samples for computing the Bootstrap energy estimator $E_K^B$. We clip the norm of $S_K$, $s_\theta$, and $\nabla E_\theta$ to 20 during training and sampling. The variance controller for BNEM is set to be $\beta = 0.2$.

**LJ-13.** All models use an EGNN with 5 hidden layers and hidden layer size 128. Baseline models are trained with a geometric noise schedule with $\sigma_{\min} = 0.01$ and $\sigma_{\max} = 2$; NEM and BNEM are trained with a geometric noise schedule with $\sigma_{\min} = 0.001$ and $\sigma_{\max} = 6.0$ to ensure the data well mixed to Gaussian. We use a learning rate of $1e-3$, $K = 500$ samples for $E_K^B$, and we clipped $S_K$, $s_\theta$ and $\nabla E_\theta$ to a max norm of 20 during training and sampling. The variance controller for BNEM is set to be $\beta = 0.5$.

**LJ-55.** All models use an EGNN with 5 hidden layers and hidden layer size 128. All models are trained with a geometric noise schedule with $\sigma_{\min} = 0.5$ and $\sigma_{\max} = 4$. We use a learning rate of $1e-3$, $K = 500$ samples for $E_K^B$. We clipped $S_K$ and $s_\theta$ to a max norm of 20 during training and sampling. And we clipped $\nabla E_\theta$ to a max norm of 1000 during sampling, as our model can capture better scores and therefore a small clipping norm can be harmful for sampling. The variance controller for BNEM is set to be $\beta = 0.4$.

**For all datasets.** We use clipped scores as targets for iDEM and TweeDEM training for all tasks. Meanwhile, we also clip scores during sampling in outer-loop of training, when calculating the reverse SDE integral. These settings are shown to be crucial especially when the energy landscape is non-smooth and exists extremely large energies or scores, like LJ-13 and LJ-55. In fact, targeting the clipped scores refers to learning scores of smoothed energies. While we're learning unadjusted energy for NEM and BNEM, the training can be unstable, and therefore we often tend to use a slightly larger $\sigma_{\min}$. Also, we smooth the Lennard-Jones potential through the cubic spline interpolation, according to Moore et al. (2024). Besides, we predict per-particle energies for DW-4 and LJ-n datasets, which can provide more information on the energy system. It shows that this setting can significantly stabilize training and boost performance.

# J SUPPLEMENTARY EXPERIMENTS

## J.1 MAIN RESULTS

We report a detailed version of the main table 1 in Table 3, which includes the mean and standard deviation of metrics.

Table 4: Ablation Study on applying energy smoothing based on Cubic Spline for iDEM

| Energy $\rightarrow$ | **LJ-55** ($d = 165$) | | |
|---|---|---|---|
| Sampler $\downarrow$ | $\mathbf{x}$-$\mathcal{W}_2\downarrow$ | $\mathcal{E}$-$\mathcal{W}_2\downarrow$ | **TV**$\downarrow$ |
| iDEM | $2.077 \pm 0.0238$ | $169347 \pm 2601160$ | $0.165 \pm 0.0146$ |
| iDEM (cubic spline smoothed) | $2.086 \pm 0.0703$ | $12472 \pm 8520$ | $0.142 \pm 0.0095$ |
| NEM (ours) | $\mathbf{1.898} \pm 0.0097$ | $\mathbf{118.57} \pm 106.62$ | $\mathbf{0.0991} \pm 0.0194$ |

Table 5: Neural sampler performance comparison for 3 different energy functions. The number after the sampler, *e.g.* NEM-100, represents the number of integration steps and MC samples is 100. We measured the performance using data Wasserstein-2 distance(x-$\mathcal{W}_2$), Energy Wasserstein-2 distance($\mathcal{E}$-$\mathcal{W}_2$) and Total Variation(TV).

| Energy $\rightarrow$ | **GMM-40** ($d = 2$) | | | **DW-4** ($d = 8$) | | | **LJ-13** ($d = 39$) | | |
|---|---|---|---|---|---|---|---|---|---|
| Sampler $\downarrow$ | $\mathbf{x}$-$\mathcal{W}_2\downarrow$ | $\mathcal{E}$-$\mathcal{W}_2\downarrow$ | TV$\downarrow$ | $\mathbf{x}$-$\mathcal{W}_2\downarrow$ | $\mathcal{E}$-$\mathcal{W}_2\downarrow$ | TV$\downarrow$ | $\mathbf{x}$-$\mathcal{W}_2\downarrow$ | $\mathcal{E}$-$\mathcal{W}_2\downarrow$ | TV$\downarrow$ |
| iDEM-1000 | $4.21_{\pm0.86}$ | $1.63_{\pm0.61}$ | $0.81_{\pm0.03}$ | $\mathbf{0.42}_{\pm0.02}$ | $1.89_{\pm0.56}$ | $\mathbf{0.13}_{\pm0.01}$ | $0.87_{\pm0.00}$ | $77515.90_{\pm115028.07}$ | $0.06_{\pm0.01}$ |
| iDEM-100 | $8.21_{\pm5.43}$ | $60.49_{\pm70.12}$ | $0.82_{\pm0.03}$ | $0.50_{\pm0.03}$ | $2.80_{\pm1.72}$ | $0.16_{\pm0.01}$ | $0.88_{\pm0.00}$ | $1190.59_{\pm590,\,290}$ | $0.07_{\pm0.00}$ |
| NEM-1000 | $2.73_{\pm0.55}$ | $1.68_{\pm0.98}$ | $0.81_{\pm0.00}$ | $0.46_{\pm0.02}$ | $\mathbf{0.28}_{\pm0.08}$ | $0.28_{\pm0.13}$ | $0.02_{\pm0.01}$ | $5.01_{\pm2.56}$ | $\mathbf{0.03}_{\pm0.00}$ |
| NEM-100 | $5.28_{\pm0.89}$ | $44.56_{\pm39.56}$ | $0.91_{\pm0.02}$ | $0.48_{\pm0.02}$ | $0.85_{\pm0.52}$ | $0.14_{\pm0.01}$ | $0.88_{\pm0.00}$ | $13.14_{\pm225.45}$ | $0.04_{\pm0.00}$ |
| BNEM-1000 | $\mathbf{2.55}_{\pm0.47}$ | $\mathbf{0.36}_{\pm0.12}$ | $\mathbf{0.66}_{\pm0.08}$ | $0.49_{\pm0.01}$ | $0.29_{\pm0.05}$ | $0.15_{\pm0.01}$ | $\mathbf{0.86}_{\pm0.00}$ | $\mathbf{0.62}_{\pm0.01}$ | $\mathbf{0.03}_{\pm0.00}$ |
| BNEM-100 | $3.66_{\pm0.30}$ | $1.87_{\pm1.00}$ | $0.79_{\pm0.04}$ | $0.49_{\pm0.02}$ | $0.38_{\pm0.09}$ | $0.14_{\pm0.01}$ | $0.87_{\pm0.00}$ | $5.93_{\pm3.01}$ | $\mathbf{0.03}_{\pm0.00}$ |

## J.2 COMPARING THE ROBUSTNESS OF ENERGY-MATCHING AND SCORE-MATCHING

In this section, we discussed the robustness of the energy-matching model(NEM) with the score-matching model(DEM) by analyzing the influence of the numbers of MC samples used for estimators and choice of noise schedule on the sampler's performance.

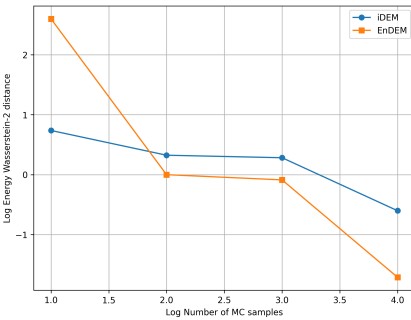

Figure 6: Comparison of the Energy Wasserstein-2 distance in DW4 benchmark between DEM and NEM across varying numbers of MC samples.

**Robustness with limited compute budget.** We first complete the robustness discussed in Section 5, by conducting experiments on a more complex benchmark - LJ-13. reports different metrics of each sampler in different settings, *i.e.* 1000 integration steps and MC samples v.s. 100 integration steps and MC samples, and different tasks.

**Robustness v.s. Number of MC samples.** As in Figure 6, NEM consistently outperforms iDEM when more than 100 MC samples are used for the estimator. Besides, NEM shows a faster decline when the number of MC samples increases. Therefore, we can conclude that the low variance of Energy-matching makes it more beneficial when we boost with more MC samples.

**Robustness v.s. Different noise schedules.** Then, we evaluate the performance differences when applying various noise schedules. The following four schedules were tested in the experiment:

- **Geometric noise schedule**: The noise level decreases geometrically in this schedule. The noise at step $t$ is given by: $\sigma_t = \sigma_0^{1-t} \cdot \sigma_1^t$ where $\sigma_0 = 0.0001$ is the initial noise level, $\sigma_1 = 1$ is the maximum noise level, and $t$ is the time step.

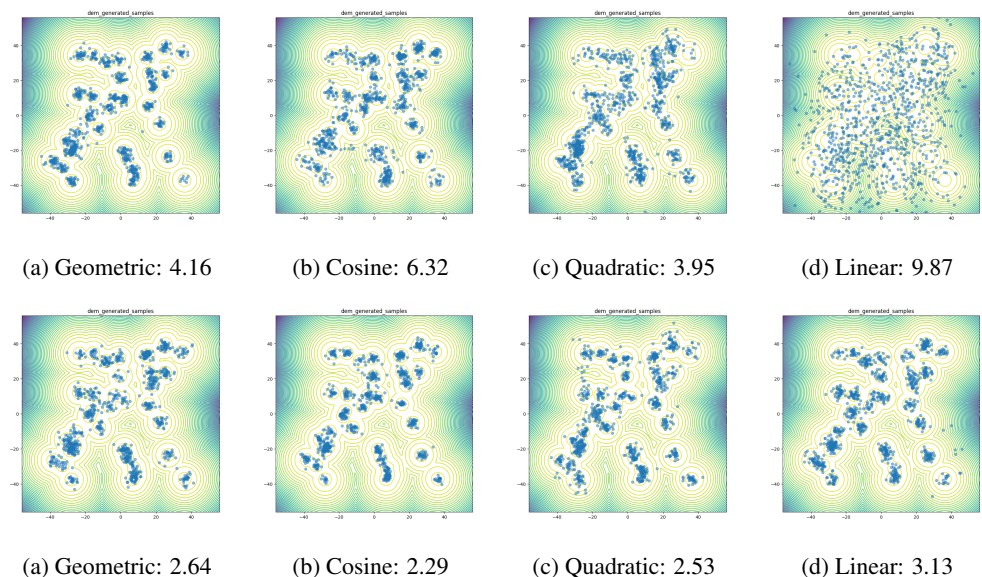

(a) Geometric: 4.16 (b) Cosine: 6.32 (c) Quadratic: 3.95 (d) Linear: 9.87

(a) Geometric: 2.64 (b) Cosine: 2.29 (c) Quadratic: 2.53 (d) Linear: 3.13

Figure 7: Comparison of different sampler (Above: iDEM; Below: NEM) when employing different noise schedules. The performances of x-$\mathcal{W}_2$ are listed.

- **Cosine noise schedule**: The noise level follows a cosine function over time, represented by: $\sigma_t = \sigma_1 \cdot \cos(\pi/2 \frac{1+\delta-t}{1+\delta})^2$, where $\delta = 0.008$ is a hyper-parameter that controls the decay rate.

- **Quadratic noise schedule**: The noise level follows a quadratic decay: $\sigma_t = \sigma_0 t^2$ where $\sigma_0$ is the initial noise level. This schedule applies a slow decay initially, followed by a more rapid reduction.

- **Linear noise schedule**: In this case, the noise decreases linearly over time, represented as: $\sigma_t = \sigma_1 t$

The experimental results are depicted in Figure 7. It is pretty obvious that for iDEM the performance varied for different noise schedules. iDEM favors noise schedules that decay more rapidly to 0 when $t$ approaches 0. When applying the linear noise schedules, the samples are a lot more noisy than other schedules. This also proves our theoretical analysis that the variance would make the score network hard to train. On the contrary, all 4 schedules are able to perform well on NEM. This illustrates that the reduced variance makes NEM more robust and requires less hyperparameter tuning.

**Robustness in terms of Outliers.** Based on Figure 3, we set the maximum energy as (GMM-40: 100; DW-4: 0; LJ-13: 0; LJ-55: $-150$). We remove outliers based on these thresholds and recomputed the $\mathcal{E}$-$\mathcal{W}_2$. We report the new values as well as percentage of outliers in Table 6, which shows that the order of performance (BNEM>NEM>iDEM) still holds in terms of better $\mathcal{E}$-$\mathcal{W}_2$ value and less percentage of outliers.

Table 6: $\mathcal{E}$-$\mathcal{W}_2$ w/o outliers (outlier%) for different models and datasets. **Bold** indicates the best value and underline indicates the second one.

| Sampler↓ Energy→ | GMM-40 ($d=2$) | DW-4 ($d=8$) | LJ-13 ($d=39$) | LJ-55 ($d=165$) |
|---|---|---|---|---|
| iDEM | 0.138 (0.0%) | 8.658 (0.02%) | 88.794 (4.353%) | 21255 (0.29%) |
| NEM (ours) | 0.069 (0.0%) | 4.715 (**0.0%**) | 5.278 (0.119%) | 98.206 (0.020%) |
| BNEM (ours) | **0.032** (0.0%) | **1.050** (**0.0%**) | **1.241** (**0.025%**) | **11.401** (**0.0%**) |

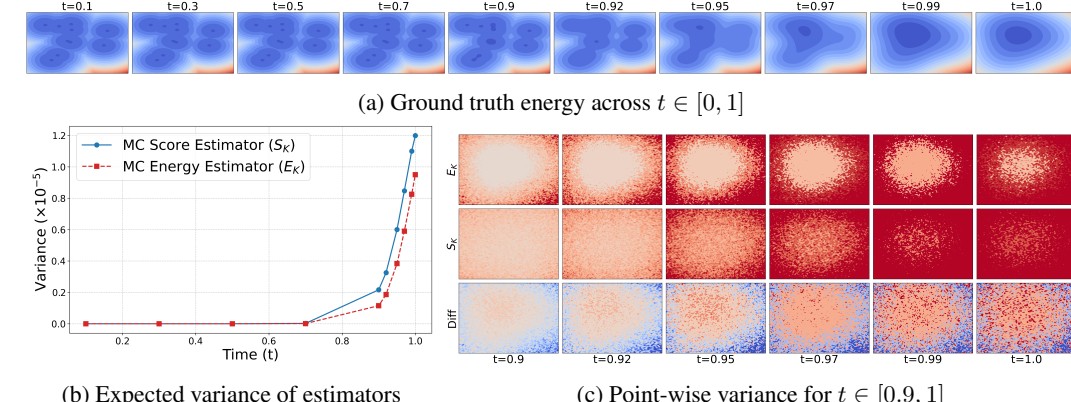

(a) Ground truth energy across $t \in [0, 1]$

(b) Expected variance of estimators

(c) Point-wise variance for $t \in [0.9, 1]$

Figure 8: (a) the ground truth energy of the target GMM from $t = 0$ to $t = 1$; (b) the estimation of expected variance of $x$ from $t = 0$ to $t = 1$, computed by a weighted sum over the variance of estimator at each location with weights equal to the marginal density $p_t$; (c) the variance of MC score estimator and MC energy estimator, and their difference (Var[score]-Var[energy]) for $t$ from 0.9 to 1, we ignore the plots from $t \in [0, 0.9]$ since the variance of both estimators are small. The colormap ranges from blue (low) to red (high), where blues are negative and reds are positive.

### J.3 Empirical Analysis of the Variance of $E_K$ and $S_K$

To justify the theoretical results for the variance of the MC energy estimator (9) and MC score estimator (7), we first empirically explore a 2D GMM. For better visualization, the GMM is set to be evenly weighted by 10 modes located in $[-1, 1]^2$ with identical variance $1/40$ for each component, resulting in the following density

$$p'_{GMM}(x) = \frac{1}{10} \sum_{i=1}^{10} \mathcal{N}\left(x; \mu_i, \frac{1}{40}I\right) \tag{130}$$

while the marginal perturbed distribution at $t$ can be analytically derived from Gaussian's property:

$$p_t(x) = (p'_{GMM} * \mathcal{N}(0, \sigma_t^2))(x_t) = \frac{1}{10} \sum_{i=1}^{10} \mathcal{N}\left(x; \mu_i, \left(\frac{1}{40} + \sigma_t^2\right)I\right) \tag{131}$$

given a VE noising process.

We empirically estimate the variance for each pair of $(x_t, t)$ by simulating 10 times the MC estimators. Besides, we estimate the expected variance over $x$ for each time $t$, i.e. $\mathbb{E}_{p_t(x_t)}[\text{Var}(E_K(x_t, t))]$ and $\mathbb{E}_{p_t(x_t)}[\text{Var}(S_K(x_t, t))]$.

Figure 8a shows that, the variance of both MC energy estimator and MC score estimator increase as time increases. In contrast, the variance of $E_K$ can be smaller than that of $S_K$ in most areas, especially when the energies are low (see Figure 8c), aligning our Proposition 2. Figure 8b shows that in expectation over true data distribution, the variance of $E_K$ is always smaller than that of $S_K$ across $t \in [0, 1]$.

### J.4 Empirical Analysis of the Bias of Bootstrapping

To show the improvement gained by bootstrapping, we deliver an empirical study on the GMM-40 energy in this section. As illustrated in Section I.3, the modes of GMM-40 are located between $[-40, 40]^2$ with small variance. Therefore, the sub-Gaussianess assumption is natural. According to Proposition A and Proposition B, the analytical bias of the MC energy estimator (Eq. 9) and Bootstrapped energy estimator (Eq. 55) can be computed by Eq. 24 and Eq. ??, respectively. We

provide these two bias terms here for reference,

$$\text{Bias}(E_K(x_t, t)) = \frac{v_{0t}(x_t)}{2m_t(x_t)^2 K} \tag{132}$$

$$\text{Bias}(E_K(x_t, t, s; \theta)) = \frac{v_{0t}(x_t)}{2m_t^2(x_t)K^{n+1}} + \sum_{j=1}^{n} \frac{v_{0s_j}(x_t)}{2m_{s_j}^2(x_t)K^j} \tag{133}$$

Given a Mixture of Gaussian with $K$ components, $p_0(x) = \sum_k \pi_k \mathcal{N}(x; \mu_k, \Sigma_k)$ and $\mathcal{E}(x) = -\log p_0(x)$, $m_t(x)$ and $v_{0t}$ can be calculated as follows:

$$m_t(x_t) = \exp(-\mathcal{E}_t(x_t)) \tag{134}$$

$$= \int \mathcal{N}(x_t; x, \sigma_t^2 I) \exp(-\mathcal{E}(x)) dx \tag{135}$$

$$= \int \mathcal{N}(x_t; x, \sigma_t^2 I) p_0(x) dx \tag{136}$$

$$= \sum_k \pi_k \int \mathcal{N}(x_t; x, \sigma_t^2 I) \mathcal{N}(x; \mu_k, \Sigma_k) dx \tag{137}$$

$$= \sum_k \pi_k \mathcal{N}(x_t; \mu_k, \sigma_t^2 I + \Sigma_k) \tag{138}$$

$$v_{0t}(x_t) = \text{Var}_{\mathcal{N}(x; x_t, \sigma_t^2 I)}(\exp(-\mathcal{E}(X))) \tag{139}$$

$$= \int \mathcal{N}(x_t; x, \sigma_t^2 I) p_0(x)^2 dx - m_t^2(x_t) \tag{140}$$

$$= \sum_{j,k} \pi_j \pi_k \int \mathcal{N}(x_t; x, \sigma_t^2 I) \mathcal{N}(x; \mu_k, \Sigma_k) \mathcal{N}(x; \mu_j, \Sigma_j) dx - m_t^2(x_t) \tag{141}$$

$$= \sum_{j,k} \pi_j \pi_k \int \mathcal{N}(x_t; x, \sigma_t^2 I) \mathcal{N}(x; \mu_{jk}, \Sigma_{jk}) C_{jk} \frac{|\Sigma_{jk}|^{1/2}}{(2\pi)^{d/2} |\Sigma_j|^{1/2} |\Sigma_k|^{1/2}} dx - m_t^2(x_t) \tag{142}$$

$$= \sum_{j,k} \pi_j \pi_k C_{jk} \frac{|\Sigma_{jk}|^{1/2}}{(2\pi)^{d/2} |\Sigma_j|^{1/2} |\Sigma_k|^{1/2}} \mathcal{N}(x_t; \mu_{jk}, \sigma_t^2 I + \Sigma_{jk}) - m_t^2(x_t) \tag{143}$$

where

$$\Sigma_{jk} = (\Sigma_j^{-1} + \Sigma_k^{-1})^{-1} \tag{144}$$

$$\mu_{jk} = \Sigma_{jk}(\Sigma_j^{-1} \mu_j + \Sigma_k^{-1} \mu_k) \tag{145}$$

$$C_{jk} = \exp\left(-\frac{1}{2}\left[\mu_j^\top \Sigma_j^{-1} \mu_j + \mu_k^\top \Sigma_k^{-1} \mu_k - (\Sigma_j^{-1} \mu_j + \Sigma_k^{-1} \mu_k)^\top \Sigma_{jk}(\Sigma_j^{-1} \mu_j + \Sigma_k^{-1} \mu_k)\right]\right) \tag{146}$$

In our GMM-40 case, the covariance for each component are identical and diagonal, i.e. $\Sigma_k \equiv \Sigma = vI$. By plugging it into the equations, we can simplify the $m_t(x_t)$ and $v_{0t}(x_t)$ terms as follows

$$m_t(x_t) = \sum_k \frac{1}{K} \mathcal{N}(x_t; \mu_k, (\sigma_t^2 + v)I) \tag{147}$$

$$v_{0t}(x_t) = \sum_{j,k} \frac{1}{K^2} \frac{\exp\left(-\frac{1}{4v}(\mu_j - \mu_k)^\top(\mu_j - \mu_k)\right)}{\sqrt{2}(2\pi v)} \mathcal{N}\left(x_t; \frac{1}{2}(\mu_j + \mu_k), (\sigma_t^2 + v/2)I\right) - m_t^2(x_t) \tag{148}$$

We computed the analytical bias terms and visualize in Figure 9. Figure 9a visualizes the bias of the both NEM and BNEM over the entire space. It shows that (1) bootstrapped energy estimator can have less bias (contributed by bias of EK and variance of training target); (2) If $E_K$ is already bias, i.e. the "red" regions in the first row of Figure 9a when $t = 0.1$, bootstrapping can not gain any improvement, which is reasonable; (3) However, if $E_K$ has low bias, i.e. the "blue" regions

Table 7: Time complexity of different neural samplers.

| Sampler↓ Phase→ | Inner-loop | Outer-loop (or Sampling) |
|---|---|---|
| iDEM | $\mathcal{O}\left(L(2K\Gamma_{\mathcal{E}}(B) + 2\Gamma_{\text{NN}}(B))\right)$ | $\mathcal{O}\left(T\Gamma_{\text{NN}}(B)\right)$ |
| NEM (ours) | $\mathcal{O}\left(L(K\Gamma_{\mathcal{E}}(B) + 2\Gamma_{\text{NN}}(B))\right)$ | $\mathcal{O}\left(2T\Gamma_{\text{NN}}(B)\right)$ |
| BNEM (ours) | $\mathcal{O}\left(L(K\Gamma_{\text{NN}}(B) + 2\Gamma_{\text{NN}}(B))\right)$ | $\mathcal{O}\left(2T\Gamma_{\text{NN}}(B)\right)$ |

Table 8: Time comparison (in seconds) of different samplers for both inner-loop and outer-loop across different energies.

| Energy → | **GMM-40** ($d = 2$) | | **DW-4** ($d = 8$) | | **LJ-13** ($d = 39$) | | **LJ-55** ($d = 165$) | |
|---|---|---|---|---|---|---|---|---|
| Sampler ↓ | **Inner-loop** | **Outer-loop** | **Inner-loop** | **Outer-loop** | **Inner-loop** | **Outer-loop** | **Inner-loop** | **Outer-loop** |
| iDEM | 1.657 | 1.159 | 6.783 | 2.421 | 21.857 | 21.994 | 36.158 | 47.477 |
| NEM (ours) | 1.658 | 2.252 | 5.217 | 8.517 | 14.646 | 52.563 | 17.171 | 114.601 |
| BNEM (ours) | 1.141 | 2.304 | 25.552 | 7.547 | 68.217 | 52.396 | 113.641 | 115.640 |

when $t = 0.1$, the Bootstrapped energy estimator can result in lower bias estimation, superioring MC energy estimator; (4) In low energy region, both MC energy estimator and Bootstrapped one result in accurate estimation. However, in a bi-level iterated training fashion, we always probably explore high energy at the beginning. Therefore, due to the less biasedness of Bootstrapped estimator at high energy regions, we're more likely to have more informative pseudo data which can further improve the model iteratively.

On the other hand, we ablate different settings of Num. of MC samples and the variance-control (VC) parameter. We visualize the results in Figure 9b. The results show that, with proper VC, bootstrapping allows us to reduce the bias with less MC samples, which is desirable in high-dimensional and more complex problems.

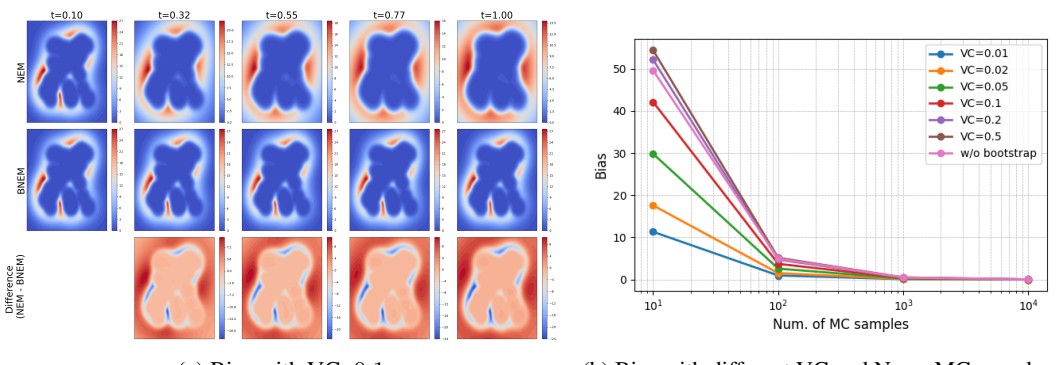

(a) Bias with VC=0.1          (b) Bias with different VC and Num. MC samples.

Figure 9: Empirical analysis on bias with bootstrapping, on GMM-40.

### J.5 COMPLEXITY ANALYSIS

To compare the time complexity between iDEM, NEM and BNEM, we let: (1) In the outer-loop of training, we have $T$ integration steps and batch size $B$; (2) In the inner-loop, we have $L$ epochs and batch size $B$. Let $\Gamma_{\text{NN}}(B)$ be the time complexity of evaluating a neural network w.r.t. $B$ data points, $\Gamma_{\mathcal{E}}(B)$ be the time complexity of evaluating the clean energy w.r.t. $B$ data points, and $K$ be the number of MC samples used. Since differentiating a function $f$ using the chain rule requires approximately twice the computation as evaluating $f$, we summurise the time complexity of iDEM, NEM, and BNEM in Table 7. It shows that in principle, in the inner-loop, NEM can be slightly faster than iDEM, while BNEM depends on the relativity between complexity of evaluating the neural network and evaluating the clean energy function.

Table 9: Comparison between iDEM, NEM, and BNEM, with similar computational budget.

| Energy $\rightarrow$ | **LJ-13** ($d = 39$) | | | **LJ-55** ($d = 165$) | | |
|---|---|---|---|---|---|---|
| Sampler $\downarrow$ | $\mathbf{x}$-$\mathcal{W}_2\downarrow$ | $\mathcal{E}$-$\mathcal{W}_2\downarrow$ | **TV**$\downarrow$ | $\mathbf{x}$-$\mathcal{W}_2\downarrow$ | $\mathcal{E}$-$\mathcal{W}_2\downarrow$ | **TV**$\downarrow$ |
| iDEM | 0.870 | 6670 | 0.0600 | 2.060 | 17651 | 0.160 |
| NEM-500 (ours) | 0.870 | 31.877 | 0.0377 | 1.896 | 11018 | 0.0955 |
| BNEM-500 (ours) | 0.866 | 2.242 | 0.0329 | 1.890 | 25277 | 0.113 |

Table 10: Performance comparison between NEM and ME-NEM.

| Energy $\rightarrow$ | **GMM-40** ($d = 2$) | | | **DW-4** ($d = 8$) | | | **LJ-13** ($d = 39$) | | |
|---|---|---|---|---|---|---|---|---|---|
| Sampler $\downarrow$ | $\mathbf{x}$-$\mathcal{W}_2\downarrow$ | $\mathcal{E}$-$\mathcal{W}_2\downarrow$ | **TV**$\downarrow$ | $\mathbf{x}$-$\mathcal{W}_2\downarrow$ | $\mathcal{E}$-$\mathcal{W}_2\downarrow$ | **TV**$\downarrow$ | $\mathbf{x}$-$\mathcal{W}_2\downarrow$ | $\mathcal{E}$-$\mathcal{W}_2\downarrow$ | **TV**$\downarrow$ |
| NEM (ours) | 1.808 | 0.846 | 0.838 | 0.479 | 2.956 | 0.14 | 0.866 | 27.362 | 0.0369 |
| ME-NEM (ours) | 2.431 | 0.107 | 0.813 | 0.514 | 1.649 | 0.164 | 0.88 | 20.161 | 0.0338 |

Table 8 reports the time usage per inner-loop and outer-loop. It shows that due to the need for differentiation, the sampling time, *i.e.* outer-loop, of BNEM/NEM is approximately twice that of iDEM. In contrast, the inner-loop time of NEM is slightly faster than that of iDEM, matching the theoretical time complexity, and the difference becomes more pronounced for more complex systems such as LJ-13 and LJ-55. For BNEM, the sampling time is comparable to NEM, but the inner-loop time depends on the relative complexity of evaluating the clean energy function versus the neural network, which can be relatively higher.

### J.6 PERFORMANCE GAIN UNDER THE SAME COMPUTATIONAL BUDGET

It's noticable that computing the scores by differentiating the energy network outputs, *i.e.* $\nabla_{x_t} E_\theta(x_t, t)$, requires twice of computation compared with iDEM which computes $s_\theta(x_t, t)$ by one neural network evaluation. In this section, we limit the computational budget during sampling of both NEM and BNEM by reducing their integration steps to half. We conduct experiment on LJ-13 and LJ-55, where we reduce the reverse SDE integration steps in both NEM and BNEM from 1000 to 500. Metrics are reported in Table 9. It shows that with similar computation, NEM and BNEM can still outperform iDEM.

### J.7 EXPERIMENTS FOR MEMORY-EFFICIENT NEM

In this section, we conduct experiments on the proposed Memory-Efficient NEM (ME-NEM). The number of integration steps and MC samples are all set to 1000. ME-NEM is proposed to reduce the memory overhead caused by differentiating the energy network in NEM, which leverages the Tweedie's formula to establish a 1-sample MC estimator for the denoising score. In principle, ME-NEM doesn't require neural network differentiation, avoiding saving the computational graph. Though it still requires evaluating the neural network twice (see Eq. 115), this only requires double memory usage of iDEM and can be computed parallelly, resulting in a similar speed of sampling with iDEM. In this section, we simply show a proof-of-concept experiment on ME-NEM, while leaving more detailed experiments as our future work.

Table 10 reports the performance of NEM and ME-NEM, showcasing that ME-NEM can achieve similar results even though it leverages another MC estimator during sampling.

### J.8 EXPERIMENTS FOR TWEEDEM

In Appendix H, we propose TweeDEM, a variant of DEM by leveraging Tweedie's formula (Efron, 2011), which theoretically links iDEM and iEFM-VE and suggests that we can simply replace the score estimator $S_K$ (7) with $\tilde{S}_K$ (121) to reconstruct a iEFM-VE. We conduct experiments for this variant with the aforementioned GMM and DW-4 potential functions.

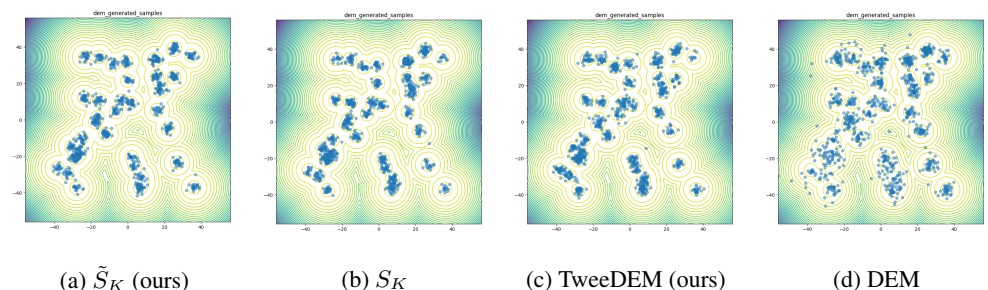

(a) $\tilde{S}_K$ (ours)      (b) $S_K$      (c) TweeDEM (ours)      (d) DEM

Figure 10: Sampled points from samplers applied to GMM-40 potentials, with the ground truth represented by contour lines. $\tilde{S}_K$ and $S_K$ represent using these ground truth estimators for reverse SDE integration.

**Setting.** We follow the ones aforementioned, but setting the steps for reverse SDE integration 1000, the number of MC samples 500 for GMM and 1000 for DW-4. We set a quadratic noise schedule ranging from 0 to 3 for TweeDEM in DW-4.

To compare the two score estimators $S_K$ and $\tilde{S}_K$ fundamentally, we first conduct experiments using these ground truth estimations for reverse SDE integration, *i.e.* samplers without learning. In addition, we consider using a neural network to approximate these estimators, *i.e.* iDEM and TweeDEM.

Table 11 reports x-$\mathcal{W}_2$, $\mathcal{E}-\mathcal{W}_2$, and TV for GMM and DW-4 potentials. Table 11 shows that when using the ground truth estimators for sampling, there's no significant evidence demonstrating the privilege between $S_K$ and $\tilde{S}_K$. However, when training a neural sampler, TweeDEM can significantly outperform iDEM (rerun), iEFM-VE, and iEFM-OT for GMM potential. While for DW4, TweeDEM outperforms iEFM-OT and iEFM-VE in terms of $x - \mathcal{W}_2$ but are not as good as our rerun iDEM.

Figure 10 visualizes the generated samples from ground truth samplers, *i.e.* $S_K$ and $\tilde{S}_K$, and neural samplers, *i.e.* TweeDEM and iDEM. It shows that the ground truth samplers can generate well mode-concentrated samples, as well as TweeDEM, while samples generated by iDEM are not concentrated on the modes and therefore result in the high value of $\mathcal{W}_2$ based metrics. Also, this phenomenon aligns with the one reported by Woo & Ahn (2024), where the iEFM-OT and iEFM-VE can generate samples more concentrated on the modes than iDEM.

Above all, simply replacing the score estimator $S_K$ with $\tilde{S}_K$ can improve generated data quality and outperform iEFM in GMM and DW-4 potentials. Though TweeDEM can outperform the previous state-of-the-art sampler iDEM on GMM, it is still not as capable as iDEM on DW-4. Except scaling up and conducting experiments on larger datasets like LJ-13, combing $S_K$ and $\tilde{S}_K$ is of interest in the future, which balances the system scores and Gaussian ones and can possibly provide more useful and less noisy training signals. In addition, we are considering implementing a denoiser network for TweeDEM as our future work, which might stabilize the training process.

Table 11: Sampler performance comparison for GMM-40 and DW-4 energy function. we measured the performance using data Wasserstein-2 distance(x-$\mathcal{W}_2$), Energy Wasserstein-2 distance($\mathcal{E}$-$\mathcal{W}_2$), and Total Variation(TV). † We compare the optimal number reported by Woo & Ahn (2024) and Akhound-Sadegh et al. (2024). . - indicates metric non-reported.

| Energy → | **GMM-40** ($d = 2$) | | | **DW-4** ($d = 8$) | | |
|---|---|---|---|---|---|---|
| Sampler ↓ | **x-$\mathcal{W}_2$**↓ | $\mathcal{E}$-$\mathcal{W}_2$↓ | **TV**↓ | **x-$\mathcal{W}_2$**↓ | $\mathcal{E}$-$\mathcal{W}_2$↓ | **TV**↓ |
| $S_K$ | 2.864 | **0.010** | **0.812** | 1.841 | **0.040** | 0.092 |
| $\tilde{S}_K$ (ours) | **2.506** | 0.124 | 0.826 | **1.835** | 0.145 | **0.087** |
| iDEM† | 3.98 | - | 0.81 | 2.09 | - | 0.09 |
| iDEM (rerun) | 6.406 | 46.90 | 0.859 | **1.862** | **0.030** | **0.093** |
| iEFM-VE† | 4.31 | - | - | 2.21 | - | - |
| iEFM-OT† | 4.21 | - | - | 2.07 | - | - |
| TweeDEM (ours) | **3.182** | 1.753 | **0.815** | 1.910 | 0.217 | 0.120 |

