# OpenReview forum: "BNEM: A Boltzmann Sampler Based on Bootstrapped Noised Energy Matching"
_ICLR.cc/2025/Conference — Submitted to ICLR 2025_

### Official Review · Reviewer_2YRA · 2024-10-27

**Soundness:** 3
**Presentation:** 3
**Contribution:** 2
**Rating:** 5
**Confidence:** 4

**Summary:**

Neural sampler of Boltzmann distributions. A variation of iDEM with better variance. The central idea of iDEM and this paper is to fit a diffusion model which generates samples which are approximately Boltzmann distributed mu(x) \propto e^(-E(x)) where E(x) is the energy function. In iDEM model is learned by matching a score network to the gradient of a monte-carlo estimator of the noised energy at a given time, t. This estimator is simulation-free but very high variance, and a bi-level training scheme is introduced to stabilize training.

The key innovation in this paper is to match the energy instead of the gradient of the energy (force/score), which they show in Propositions 1 and 2, has smaller error and variance than training against the score. A second contribution in this paper using a bootstrapping scheme to further reduce variance, at the cost of introducing some bias.

**Strengths:**

1. The paper is clearly written and addresses an important problem
2. The authors are transparent about the limitations of the work and in particular the challenges that remain for with regard to training stability etc.

**Weaknesses:**

1. Apart from the memory overhead highlighted by the authors in their limitations section, the new methods also introduce significant computational overhead, by having to use autograd to compute the derivative of the energy during sampling.
2. Experiments are performed on toy systems and not systems which reflect the actual use cases of methods like this, e.g. molecules. A simple case should be included, such as ALA2.
3. Missing comparisons to standard sampling baselines for non-normalized densities such as parallel tempering. In particular, considering the number of evaluations of E, and overall compute cost.

**Questions:**

1. To point in the weaknesses, can you give an estimate of the additional overhead with the change from score to energy parameterization? Specifically, to better understand the practical utility, an empirical comparison of the it the decrease of the variance per compute unit would need to be needed.
2. The authors speculate on the use of a Metropolis-Hastings type correction in future work. How do they envision doing this within the current BNEM framework?
3. Is it possible to correct for distributional bias of the neural sampler e.g. through importance sampling? it is with many of the base-lines you compare against. If so, what is the computational cost of this compared to the baselines?
4. Why did the authors not try a standard molecular benchmark such as ALA2? Is it related to the smoothing that needs to be performed to get the LJ-n systems to train stably? If this is the case, how do you envision moving this methodology beyond toy-systems?

---

> ### Author Response · Authors · 2024-11-23
> **Official Response to Reviewer 2YRA (Part I)**
>
> Dear reviewer 2YRA,
>
> Thank you very much for your detailed feedback and your constructive comments. We have updated our manuscript and attached answers to all of your questions and concerns below. We believe that our responses and the revised manuscript could directly address your concerns. We hope these additions encourage you to adjust your rating accordingly. If any questions or concerns arise, please do not hesitate to let us know, and we will address them properly.
>
> ## **Weakness1: Balance between performance improvement and computational cost**
>
> For your concerns about computational overheads, this section is outlined below:
>
>     (1) Time complexity Analysis
>     (2) Empirical run time comparison
>     (3) Performance comparison with similar computation
>     (4) ME-NEM: an improved method that is more memory-efficient
>
> ### (1) Time complexity Analysis
>
> We include a time complexity comparison (see https://anonymous.4open.science/r/nem_rebuttal-0D92/time_complexity.png), which theoretically shows that during training NEM can be slightly faster than iDEM in the inner-loop while it uses double computation in the outer-loop (i.e. sampling) due to the neural network differentiation. While the inner-loop training time of BNEM depends on the relative complexity of evaluating the clean energy function versus the neural network, which can be relatively higher.
>
> ### (2) Empirical run time comparison
>
> We conduct time logging for iDEM, NEM, and BNEM (see https://anonymous.4open.science/r/nem_rebuttal-0D92/run_time_comparison.png). Empirically, it shows that the sampling time, i.e. outer-loop, of BNEM/NEM is approximately twice that of iDEM, which remains within the same magnitude as iDEM. In contrast, the inner-loop time of NEM is slightly faster than that of iDEM, and the difference becomes more pronounced for more complex systems. For BNEM, the sampling time is comparable to NEM, but the inner-loop time depends on the relative complexity of evaluating the clean energy function versus the neural network, which can be relatively higher.
>
> Based on these observations, we recommend using NEM for complex target energy functions in large systems. When sufficient computational resources are available, BNEM can be employed to trade time for improved accuracy.
>
> ### (3) Performance comparison with similar computation
> We emphasize that, due to the differentiation of NN, the computational cost of both NEM and BNEM during sampling is approximately TWICE as that of iDEM. Therefore, we conduct experiments with a similar computation setting (1000 step iDEM v.s. 500 step NEM/BNEM) on LJ-13 and LJ-55. You can find the results in this link: https://anonymous.4open.science/r/nem_rebuttal-0D92/half_integration_step_comparison.png
> which shows that with similar computations NEM/BNEM can still outperform iDEM. We can also improve more by doubling the computation during sampling.
>
> ### (4) ME-NEM: an improved method that is more memory-efficient
>
> We also proposed another estimator by using Tweedie’s formula, which is able to bypass the memory issue of differentiating NN to evaluating NN twice. And the two times of NN evaluations can definitely be computed parallelly, resulting in a similar speed as iDEM but just doubling the memory usage. The proposed method can be found in (Appendix F: Memory-Efficient NEM). And experimental results are provided in https://anonymous.4open.science/r/nem_rebuttal-0D92/memory_efficient_nem.png
>
> [FYI: the experiments on LJ-55 are still running. We’ll update them once they are finished.]
>
> ## **Weakness2: Experiencing our methods on larger systems**
>
> We extended our method to the Alanine Dipeptide (ALA2) system. It is a very challenging task **without any known data** from the target ALA2 distribution. We provide preliminary results available at https://anonymous.4open.science/r/nem_rebuttal-0D92/aldp.png.
>
> iDEM exhibits mode-seeking behavior, fitting extremely sharp distributions to only a limited number of modes. In contrast, our method demonstrates slightly higher variance but achieves better mass coverage with reduced bias.
>
> Scaling up to very large systems is currently beyond the reach of existing methods, and the ALA2 problem already presents substantial challenges for current techniques. Additionally, training diffusion models becomes significantly harder when non-clean data is provided, as the noise and irregularities in the data can exacerbate the complexity of learning the underlying distributions. As the field advances, larger and more complex systems will inevitably be explored.

---

> ### Author Response · Authors · 2024-11-23
> **Officiel Response to Reviewer 2YRA (Part II)**
>
> ## **Weakness3: Missing comparison of traditional sampling methods**
>
> The primary contribution of our paper is to improve upon existing neural samplers based on diffusion models. Therefore, we didn’t include traditional methods such as parallel tempering, HMC, and simulated annealing. While it is true that these traditional sampling methods could be applied directly, these methods typically require significantly more steps (e.g., >10k) to converge to the target distribution, as is common in molecular dynamics. In contrast, although samplers based on neural networks require an additional training procedure, they are far more efficient during inference and offer an option to trade accuracy for fewer SDE integration steps.
>
> ## **Q1: Estimate of additional overhead and qualitative results for the decrease of variance**
>
> The discussion about additional overhead can be found in our answer to “**Weakness1: Balance between performance improvement and computational cost**”.
>
> For **qualitative results for the decrease of variance**, as we mentioned in the method section (Proposition 3), the variance of the training target contributes to the **error of the predicted score** (we also call it **bias**). BNEM decreases the variance of the training target and therefore reduces these biases. However, in bootstrapping, as bias can be accumulated from the previous time step, the bias can also be increased. Therefore, bootstrapping can reduce bias on one hand, but also increase it on the other hand. We empirically show such behavior on the GMM-40 task (see https://anonymous.4open.science/r/nem_rebuttal-0D92/bias_analysis.png). It shows that we can reduce bias within properly set bootstrapping parameters.
>
> ## **Q2: Envision of incorporating MH sampling**
>
> Both iDEM, NEM, and BNEM cannot generate samples with accurate weights, though they can find modes, which is not optimal in sampling settings. Therefore, improving the weights with tools like Metropolis-Hastings (MH) is necessary.
>
> To incorporate MH, we have
>
> (1) the learned marginal energy $E_\theta(x_t, t)\approx\mathcal{E}_t(x_t)$, and
>
> (2) the transitions between adjacent noise levels are Gaussian, *i.e.* 1-step Langevin dynamics $x_{t-1}|x_t\sim\mathcal{N}(x;x_t - g(t)^2\nabla E_\theta(x_t, t), g(t)^2I)$
>
> We can apply MH at each internal transition to get higher quality samples.
>
> ### Probs:
>
> (1) The model can generate samples with more accurate weights, instead of only discovering modes
>
> (2) we might be able to decrease the integration steps a lot without sacrificing the performance by doing MH.
>
> ### Cons:
>
> (1) The marginal energy is approximated by the energy model, which might not be exact. Therefore, the imperfectness of model introduces bias in the MH step, which might result in some misleadings. However, we are able to replace the predicted energy with the clean energy to provide more accurate guidance, i.e. using $\mathcal{E}(x_0)$ instead of $E_\theta(x_t, t)$ for small $t$.
>
> ## **Q3: Implement importance sampling and the corresponding computational cost**
>
> We believe that implementing importance sampling (IS) is not feasible or necessary in our case, as we do not have access to $p_\theta(x)$ for the generated samples. For standard diffusion models, $p_\theta(x)$ is obtained by integrating the probability flow ODE (PF-ODE), but this approach is often inaccurate, especially when we do not have clean sampling. In the iDEM framework, a continuous normalizing flow (CNF) is proposed to compute $p_\theta(x)$, but this method is also computationally inefficient as it requires generating lots of data and training an additional flow-based model.
> For NEM/BNEM, one potential approach could be to use $\exp(-E_\theta(x, 0))$ as an unnormalized model density. Since, in principle, $p_0$ and $p_{\theta, 0}$ share the same partition function, the importance weight could then be approximated as
> $\exp(-\mathcal{E}(x) + E_\theta(x, 0))$. However, it might yield inaccurate estimations because the ground-truth model density should be integrating the PF-ODE from t=1 to t=0. Simply using $\exp(-E_\theta(x, 0))$ as model density can introduce mismatch, therefore biased.
>
> ## **Q4: Experiencing proposed methods on Alanine Dipeptide**
>
> We updated new experiments on ALA2 system with our method, where the preliminary results (https://anonymous.4open.science/r/nem_rebuttal-0D92/aldp.png) show that NEM can be more likely to succeed in ALA2 than iDEM. The discussion can be found in our response to **Weakness2: Experiencing our methods on larger systems** above.
>
>
> ## **End of Response**
>
> Thank you again for your valuable and constructive comments. We really appreciate your efforts. We would like to kindly ask if we have addressed all of your concerns. If so, could you increase your rating accordingly? If there are any remaining concerns or if you want more discussions on the presented methods, please do not hesitate to let us know.

---

> > ### Comment · Reviewer_2YRA · 2024-11-24
> >
> > Thanks for addressing my concerns and seriously engaging with my comments. I would raise my score to a 4 for your efforts if it was possible. The paper is overall solid, my main concern with the approach remains the potential of utility down the line. The new results verified my suspicion: when calibrated compute is used the gains are fairly marginal over existing methods -- likewise, the results for ALA2, while it is unclear what is shown in the histograms, do not look promising either.  I still do not see this routinely addressing large scale problems with non-normalized densities, e.g. sampling in computational physics or chemistry, and consequently I hesitate to increase my score any further. Please let me know if you believe I have misunderstood your results and think I am being unfair.

---

> > > ### Author Response · Authors · 2024-11-24
> > >
> > > Dear reviewer 2YRA,
> > >
> > > Thank you for your response and comments. We believe that the below explanation and more details of the experiments would address your concerns.
> > >
> > > ## **Q1: Marginal performance increase with more computes**
> > >
> > > Thank you for pointing out that in the current toy tasks, we use more computations in NEM and BNEM. However, for more complex systems such as ALA2, computing molecule free energy would be much more expensive than evaluating the neural networks. Therefore, during training, the computation of the inner-loop (which involves computing the free energy when estimating the training target) would **dominate** the computation of the outer-loop (which only involves neural network evaluation and its differentiation for NEM and BNEM). Our experiment shows that, in one training iteration, the inner-loop takes around **90** seconds for both iDEM and NEM, while the outer-loop takes **5** seconds for NEM and **2** seconds for iDEM. Therefore, training NEM and BNEM is **NOT** necessary to be more expensive than training iDEM in more complex systems.
> > >
> > > ## **Q2: More details on the ALA2 experiments**
> > >
> > > We apologize for the unclear explanations for the ALA2 experiments in our previous responses. We will explain more in this section.
> > >
> > > Firstly, as we mentioned in the above answer to **Q1** the training time of one iteration between NEM and iDEM is around **95 seconds v.s. 92 seconds**, we can conclude that NEM **uses almost the same training time** as iDEM in ALA2 and even more complex systems.
> > >
> > > Secondly, we would like to give more details about the preliminary experimental results in ALA2. In our current setting, we choose to train on the internal coordinates, *i.e.* dihedral angles. The results in https://anonymous.4open.science/r/nem_rebuttal-0D92/aldp.png represent 1D histogram of each dimension (there are 19 dimensions in total, which are bond dihedrals), where blues are the ground-truth histograms and oranges are models’. These histograms show that iDEM (**left**) tends to mode-seeking or fit skewed modes, while NEM (**right**) can capture more modes.
> > >
> > > We agree with you that existing diffusion-based sampling methods are not yet applicable to relatively larger systems. When visualizing the two key dihedral angles in ALA2 (see https://anonymous.4open.science/r/nem_rebuttal-0D92/dihedral.png), we observe that both diffusion-based models successfully capture the primary modes of the target distribution, despite utilizing only a three-layer MLP for the energy network.
> > > We believe the primary bottleneck in applying our method lies in the model architecture rather than the diffusion process itself. This is evident when considering the degrees of freedom: ALA2 models 19 degrees of freedom (bond dihedrals), whereas LJ55 has 162. With respect to the number of modes, ALA2 exhibits fewer modes than ManyWells, a system we have previously worked on within our additional experiment.
> > >
> > > For instance, in the FAB paper[1], the authors employed sophisticated techniques to construct flows on a manifold ranging from $0$ to $2\pi$. When we ran our experiments on ALA2, we found if we are using a noising schedule where the maximum noise is close to $\pi$, the model would be extremely unstable because it cannot make the same output for $2\pi + x$ and $x$.  A solution exists for data-driven diffusion, such as torsion diffusion[2], which could be implemented here. However, we consider this an avenue for future exploration.
> > >
> > > Thanks again for your comments and suggestions for future directions to apply our methods.
> > >
> > > [1] Laurence Illing Midgley et al, Flow Annealed Importance Sampling Bootstrap, arxiv.org/abs/2208.01893
> > >
> > > [2] Bowen Jing et al, Torsional Diffusion for Molecular Conformer Generation
> > > , arxiv.org/abs/2206.01729
> > >
> > >
> > > ## End of Response
> > > Again, thank you for replying and your constructive comments. We sincerely appreciate that. We would like to kindly ask if we have addressed your concerns on scaling up to larger and more complex systems as well as the ALA2 experiment results. If there are any remaining concerns, please do not hesitate to let us know.

---

> > > > ### Comment · Reviewer_2YRA · 2024-11-25
> > > >
> > > > Thanks for providing additional context for my first concern and thanks for providing an additional plot of the main torsion angles of ala2. What potential energy model used for this. The baseline does not look converged at all, it is only sampling two out of 6 modes of the typical ala2 surface (in implicit/explicit solvent) or 3 in vacuum.

---

> > > > > ### Author Response · Authors · 2024-11-26
> > > > > **Official Response to Reviewer 2YRA (for ALA2)**
> > > > >
> > > > > Dear reviewer 2YRA,
> > > > >
> > > > > Thank you very much for the question as well as pointing out the bugs there. We hope our response address your concerns.
> > > > >
> > > > > We are currently using the **implicit environment** for the potential model. In our earlier visualizations of key dihedral angles, only 3-4 modes were observed because the distribution was based on 1,000 generated conformers. To address this, we increased the number of conformers to 10,000 and resolved a few bugs in the plotting code. Additionally, we incorporated positional encoding into the MLP input and expanded the network to a 4-layer MLP. The updated results, available at https://anonymous.4open.science/r/nem_rebuttal-0D92/aldp.png, clearly demonstrate that both methods can explore multiple modes. Moving forward, we aim to investigate more complex model architectures to better handle diffusion on the $[0,2\pi]^d$ manifold in the future.

---

> > > > > > ### Comment · Reviewer_2YRA · 2024-11-27
> > > > > >
> > > > > > Thanks for drawing more samples. As I expected it seems like neither iDEM nor NEM capture the target density well in this space. Naturally, it would be better than a naive proposal density and consequently, one could see it favorably in this light. While, I remain skeptical about the overall utility of the approach, I acknowledge the efforts and transparency of the authors and have decided to raise my score.

---

### Official Review · Reviewer_zWMt · 2024-11-03

**Soundness:** 2
**Presentation:** 3
**Contribution:** 2
**Rating:** 3
**Confidence:** 4

**Summary:**

In this work, the authors propose Noised Energy Matching (NEM) and Bootstrap NEM (BNEM), a diffusion-based neural sampler based on previous related work (iDEM). In particular, the authors propose to parametrise the energy with a network, instead of the score, to reduce the variance of the estimator. They also propose to bootstrap the energy estimator at high noise levels from learned estimators at lower noise levels to further reduce the variance.

**Strengths:**

The paper is generally well-written and easy to follow. The authors provide enough background to make the paper easy to understand. The topic of diffusion-based samplers is also an important and interesting topic with applications in many domains. Additionally,  improving the MC estimator of the score when we have access to the energy and not data is, in my opinion, very well-motivated.

**Weaknesses:**

**Novelty and Motivation**:
Although the NEM algorithm is a simple reparametrization of the estimator in iDEM, the paper analyzes the implications of an energy or score parametrization and  shows that this simple change can lead to a better estimator. The more novel algorithm is BNEM, however, the paper does not provide enough experimental results to support the benefit of this method (see my notes under Experiments).

- Theoretical Limitations: The work makes several claims regarding the variance of the estimators which aren’t precise. For example, the claim that “the variances of EK and SK explode at high noise levels as a result of the VE noising.” This isn’t correct as the variance is a function of multiple factors, including the importance sampling proposal chosen for the estimator, as well as the energy landscape itself.
Additionally, the proof of proposition 2 considers low-energy regions and claims that $exp{(\mathcal{E}(x_{0|t}^{(i)})} \geq c$. However, it isn’t clear to me how these regions are defined as the energy is evaluated at the noised samples.

**Experiments**:
- The work contains several experiments on the same datasets and benchmarks as the previously related paper (iDEM), showing better performance of the method, especially when the number of MC samples and integration steps are limited. However, I find that additional experiments, to show the benefit of lowering the variance at the cost of increasing the bias would strengthen the claims of the paper much more. For example, the performance gain from BNEM, at the cost of increasing the bias isn’t clear in my opinion. Generally, neural networks are fairly good at training in high-variance settings (e.g. diffusion models). On the other hand, for complex energy distributions and in high dimensional settings, the bias could result in significant problems. In general, I think the authors should provide more analysis of the trade-off of bias and variance when introducing the BNEM estimator.
- The paper also claims that the proposed approach is more robust compared to iDEM as it doesn’t rely on clipping the score. However, for the LJ potentials cubic spline interpolation was used, as the energy is very sharp. It isn’t clear to me how much of an advantage this provides if smoothing is still necessary and how applicable this is to other complex and realistic settings.
- The results for iDEM on LJ-13 and LJ-55 tasks are significantly different from what iDEM paper reported. The authors also indicate divergent training for DDS, PIS and FAB on those tasks, while iDEM reported competitive values for both FAB and PIS on LJ-13 and for FAB on LJ-55. Could the author elaborate on how and why they were unable to reproduce the results?
- Some metrics such as NLL and ESS which are commonly reported in relevant works are not reported.
- In Table 3, the standard deviations over the seeds isn’t reported, and the authors report the best value, making the results misleading. For example, they indicate that the mean E-W2 of LJ-55 over the 3 seeds is in fact even higher than the one they report for iDEM. This in fact confirms my concern that improving the variance of the estimator with bootstrapping at the cost of increasing the performance doesn’t necessarily lead to increased performance.

**Questions:**

See above.

---

> ### Author Response · Authors · 2024-11-24
> **Official Response to Reviewer zWMt (Part I)**
>
> Dear reviewer zWMt,
>
> Thank you very much for pointing out your concerns. We sincerely apologize for replying late because of computational resource shortage for running experiments to address your concerns.
> We much appreciate your detailed feedback and the constructive comments. We hope that we carefully addressed your concerns and made significant updates to the manuscript based on your comments. We believe that our responses and the revised submission should address your concerns and demonstrate the strengths and contributions of our work. If so, we wish you could adjust your rating accordingly. If not, please let us know so that we can address any remaining concerns.
>
>
> ## **Q1: The definition of low-energy region in Proposition 2 and clarification on variance explosion**
>
> The “low energy regions” is defined as the regions that $\mathcal{E}(x_t) < c$, where $x_t\in\mathcal{X}$ is in the support of the target distribution and $c$ is some small constant. Therefore, the surrounding points $x_{0|t}^{(i)}$ of $x_t$ can be assumed also in the low energy region of the energy function. The estimated score in this region is bounded , so that we could derive $tr(Cov(S_K(x_t, t))) > Var(E_K(x_t, t))$
>
> Thank you for stressing out that the variance of the estimator is of multiple factors including the importance sampling proposal. It is true that when the sampling proposal is proportional to the energy, as if we used clean samples to get $x_t$, the variance only comes from the energy landscape. However, when we prove Proposition 2, we **already taken the proposal $x_{0|t}^{(i)}\sim\mathcal{N}(x_t, \sigma_t^2I)$ into account** for comparing $S_K$ with $E_K$.
>
> In addition to theoretical results, the variance explosion at large time $t$ is also true from experimental results. As in Figure 9, the variance of  $S_K$ and $E_K$ are visualized for the GMM example. It is obvious that for $t$ close to 1, both estimators have high variance compared to when $t$ is close to 0. Beyond this toy example, we also observe much lower losses for NEM and BNEM than iDEM, which suggests much lower variance of training targets (ideally, the optimal loss would be the variance of your target when using a $L_2$ loss).
>
> ## **Q2: Clarification on the bias and variance trade-off**
>
> Thank you for providing valuable comments on the bias-variance trade-off. We would like to explain more intuitions about the this in the following answer. Please do not hesitate to reach out for more discussions if you have any concerns. This answer would be outlined as follows:
>
>     (1) Why is high variance of training target favorable in standard diffusion?
>     (2) Why is high variance of training target NOT favorable in iDEM, NEM, and BNEM?
>     (3) Why bootstrapping can improve performance?
>     (4) How to improve performance by bootstrapping?
>     (5)Answer to “for complex energy distributions and in high dimensional settings, the bias could result in significant problems”
>     (6) Empirical study
>
> ### **(1) Why is high variance of training target favorable in standard diffusion?**
>
> In data-driven diffusion, it is true that neural networks perform well when training on high-variance input. But that’s because normal diffusion **is designed to fit the posterior mean**. Generally, when targeting a neural network $f_\theta(x)$ to noisy targets $\hat{y}(x)$, the optimality would be obtained as the posterior mean of the noisy target, i.e. $f_{\theta^*}(x)=\mathbb{E}[\hat{y}(x)|x]$. Therefore, standard diffusion models can be fitted by optimizing a relatively simple objective, e.g. noise prediction. In such case, the optimal neural network outputs $f_\theta(x_t, t)=\mathbb{E}[\epsilon_t|x_t]$ and we therefore can simply undo the Gaussian noise injection via $\hat{x}_0=\frac{x_t - \sigma_t f\theta(x_t, t)}{\alpha_t}$
>
> In particular, the optimal marginal scores at time $t$ are: $s_t(x_t)=\int p(x_0|x_t)\nabla\log\mathcal{N}(x_t;x_0, \sigma_t^2I)dx_0$ (DSI), then the optimal $s_\theta$ obtained by
>
> $\min_{\theta}  \mathbb{E}|| s_\theta(x_t, t)- \nabla_{x_t} \log \mathcal{N}(x_t;x_0, \sigma_t^2I)||^2$, with $x_0\sim p_0$ and $x_t|x_0\sim\mathcal{N}(x_0, \sigma_t^2I)$
>
>  is given by the above posterior mean (DSI).
>
> ### **(2) Why is high variance of training target NOT favorable in iDEM, NEM, and BNEM?**
>
> But in our case, the high variance of training target only means informative training signals and the well-fitted posterior means are not informative. In fact, learning the noisy score estimator can result in an incorrect score at $t$ when the noise scale surpasses the data scale.

---

> > ### Author Response · Authors · 2024-11-24
> > **Official Response to Reviewer zWMt (Part II)**
> >
> > ### **(3) Why bootstrapping can improve performance?**
> >
> > It’s clear that, due to the high variance of the training target, the learned scores can be much biased to the ground truth one. Here, we name the difference between the predicted score and the ground truth one as “**error**” (instead of “bias”) for clarity.
> >
> > Intuitively, reducing the variance of the training target can reduce the “**error**”. To achieve this, we employ a bootstrapping technique, which reduces the variance of training targets at time $t$ by using learned noised energy at slightly smaller time $s$. However, the learned energy at $s$ has “error”, because it is also trained on noisy targets, and therefore would introduce “bias” to the energy estimator for training at time $t$. So, we conclude that in BNEM, the “**error**” of our predicted score can be influenced by two ways:
> >
> >     (1) Variance: It can be reduced by the smaller variance of training target
> >     (2) Bias: It can be increased because the training target has accumulated error due to bootstrapping
> >
> > Therefore, BNEM trades accumulated small “error” (bias) for direct large “error” (variance). A proper way to bootstrap can reduce the “error”, while an improper one can even increase it. A more formal derivation is in our Proposition 3.
> >
> > ### **(4) How to improve performance by bootstrapping?**
> >
> > To ensure we can earn improvement by bootstrapping, one can choose K (the number of MC samples) to satisfy that the accumulated bias is yet smaller than the original bias, i.e.
> > $\frac{v_{0t}(x_t)}{2m_t^2(x_t)K^{i+1}} + \sum_{j=1}^i\frac{v_{0s_j}(x_t)}{2m_{sj}^2(x_t)K^j} < \frac{v_{0t}(x_t)}{2m_t^2(x_t)K’}$. Typically we can use less MC samples, i.e. $K<K’$, on LHS, which would not break the inequality. In the below **Empirical study** part, we include an experiment on the GMM-40 to show how to tune the bootstrap parameter to gain improvement.
> >
> > ### **(5) Answer to “for complex energy distributions and in high dimensional settings, the bias could result in significant problems”**
> >
> > Thank you for pointing out that the “bias”, which is the “**error**” we mentioned above, can be problematic in higher dimensional and more complex problems. We agree with that. However, following the argument above, the scores predicted by both iDEM, NEM, and BNEM are biased (i.e. they have “**error**”). While the “**error**” of NEM is smaller as its training target has less variance, and BNEM can further reduce the “**error**” by employing proper bootstrapping. Therefore, for complex problems, NEM and BNEM can be more favorable in terms of less “**error**”.
> >
> > ### **(6) Empirical study**
> >
> > We have visualized the bias in the GMM-40 experiment (see https://anonymous.4open.science/r/nem_rebuttal-0D92/bias_analysis.png). Supported by the theoretical understanding outlined above, these results empirically demonstrate that bootstrapping can help us learn better noised energies (i.e. less “error”).
> >
> > ## **Q3: Justification on the usage of Cubic Spline**
> >
> > Using cubic spline interpolation to smooth the LJ potential is a simple and natural approach, as introduced by [1]. In our appendix, we have included results for both iDEM and NEM using cubic spline smoothing (see https://anonymous.4open.science/r/nem_rebuttal-0D92/ablation-on-smooth.png). These experimental results demonstrate that our method outperforms the baseline when sampling from the same smoothed energy landscape. Additionally, we observed that iDEM still requires score clipping, even when the target energy is smoothed, to prevent the estimator from exploding. And that’s very likely because the training target of iDEM is very noisy, resulting in large errors in the learned scores and therefore leading to instability in sampling without careful score clipping.
> >
> > Besides, it is noticeable that the **LJ-potential is commonly used** in molecules’ free energy calculations, and therefore this smoothing technique can be applied widely in molecular tasks. We've already implemented another smoothing technique for the LJ-potential from [2] in OpenMM, one of the most popular MD frameworks. And we'll further implement the cubic spline interpolation in our future work.
> >
> > [1] Moore, J. Harry, Daniel J. Cole, and Gabor Csanyi. "Computing hydration free energies of small molecules with first principles accuracy." arXiv preprint arXiv:2405.18171(2024).
> >
> > [2] T. C. Beutler, A. E. Mark, R. C. Van Schaik, P. R. Gerber, and W. F. Van Gunsteren, Avoiding singularities and numerical instabilities in free energy calculations based on molecular simulations, Chemical Physics Let- ters 222, 529 (1994).

---

> ### Author Response · Authors · 2024-11-24
> **Official Response to Reviewer zWMt (Part III)**
>
> ## **Q4: Elaboration on the reproduction of results**
>
> It is true that our visualized performance is different from the baseline model. Our experiment is running in a **more challenging setting**.
>
> For the GMM experiment, we reduced both the number of integration steps and the number of Monte Carlo (MC) samples from 1000 to 100. If we adopt the sampling setting in the iDEM paper there will not be notable performance differences observed in visualizations, because it’s a very simple task if we have enough computational budget.
> For the LJ-55 experiment, we found that training iDEM is highly sensitive to random seed selection. Consequently, reproducing iDEM’s results without the specific random seed provided is nearly impossible. To ensure fairness, we reran their code and selected the best-performing seed for all models when generating plots.
> For other baseline methods, like FAB, DDS and PIS, we are not sure what is the exact setting that the authors of the iDEM paper conduct their experiment as they do not provide codes or random seeds, even the architecture of neural networks for them. Therefore, we have implemented all these methods and aligned them by applying the **same architecture** (3 layer MLP for GMM, EGNN for DW4, LJ13 and LJ55) with the **same number of parameters**. We have ensured a fair comparison between baseline methods in our paper.
>
> ## **Q5: Missing metrics of NLL and ESS**
>
> NLL and ESS metrics, used for evaluation in the iDEM paper, are missing in our experiments because they rely on an additional flow-matching model trained on sampled data points to compute the likelihood. While reusing the code from the public iDEM repository, we discovered that the ESS implementation contained mathematical errors. Moreover, we observed that both ESS and NLL on the validation set exhibit high variance depending on the choice of checkpoints, often overshadowing the differences between sampling methods.
>
> Also, we believe that the energy Wasserstein-2 (E-W2) contains some information about the NLL and ESS. Given
>
> 1. samples $x_i \sim p_\theta$ ($i=1,...,n$) generated by the neural sampler and
> 2. samples $y_i\sim \mu_\mathrm{target}$ ($i=1,...,n$) from the ground-truth distribution,
>
> the E-W2 is computed between $[\mathcal{E}(x_1), ..., \mathcal{E}(x_n)]$ and $[\mathcal{E}(y_1), ..., \mathcal{E}(y_n)]$, *i.e.* the log-target-density between ground-truth samples and generated samples.
>
> 1. When E-W2 is large, it indicates a fundamental mismatch between $p_\theta$ and $\mu_\mathrm{target}$. Therefore, it implies poor NLL and ESS.
>
> 2. When E-W2 is small, the samples $y_j\sim p_\theta$ have similar log-target-density as the samples $x_j\sim \mu_\mathrm{target}$. In other words, our model $p_θ$ is generating samples that have similar "typicality" under $\mu_\mathrm{target}$ as real samples from $\mu_\mathrm{target}$, implying a good NLL and potentially good ESS
>
> For these reasons, we decided to exclude these metrics from our evaluation.
>
> ## **Q6: Clarification on missing random seeds reviation**
>
> We apologize for making confusion when presenting our main table. According to your suggestion, we would like to present the experimental values with their means and standard-deviations instead of just the best value. On the other hand, we realize that our previous implementation can introduce instability when training BNEM. We’ve improved our codes and rerun their experiments. The new table (see https://anonymous.4open.science/r/nem_rebuttal-0D92/main-table-new.png) is updated in our revised manuscript.
>
> The results show that BNEM can outperform NEM in most cases. But it’s noticeable that for LJ-55, the $\mathcal{E}$-$\mathcal{W}_2$ metric of BNEM is worse than that of NEM in terms of mean and std. However, we also notice that the best trained BNEM can achieve the best value (**45.61** (BNEM) v.s. 84.17 (NEM)). And that’s because bootstrapping highly depends on whether we learn the “base” energy (the one without bootstrapping) well. It turns out that we still need a better variance indicator for evaluating NEM on small $t$ to determine whether the energy network is ready for bootstrapping. Therefore, we will keep improving the training stability of BNEM in our future work.
>
> According to our discussion in the above **Q2** and the experiment results, we conclude that bootstrapping can **theoretically** reduce the error of predicted scores, while **practically** achieve better performance by improving NEM.

---

> ### Author Response · Authors · 2024-11-24
> **Official Response to Reviewer zWMt (Part IV)**
>
> ## **Additional: we would like to point out one more benefit of NEM/BNEM**
> Generating high-quality samples from the target distribution is a fundamental sub-task in sampling. Beyond this, it is often essential for the generated samples to be assigned correct weights under the target distribution. However, several studies have noted that score-based sampling methods can be problematic in this context (e.g., [1][2]). While [1] shows that Metropolis-Hastings can solve this problem. Notably, iDEM, NEM, and BNEM cannot achieve this (see experiments on DW-4 and additional experiments on MW32 https://anonymous.4open.science/r/nem_rebuttal-0D92/mw32_prelim.png ). However, NEM and BNEM enable the estimation of unnormalized marginal densities, i.e., noised energies. This allows more techniques to be employed, such as Metropolis-Hastings inside the sampling trajectory, which leverage the learned energy to yield better weight estimations.
>
> Reference
> [1] Chen, Wenlin, et al. "Diffusive Gibbs Sampling." arXiv preprint arXiv:2402.03008 (2024).
> [2] Shi, Zhekun, et al. "Diffusion-PINN Sampler." arXiv preprint arXiv:2410.15336 (2024).
>
> ## **End of Response**
>
> We really appreciate your efforts for these reviews and all the comments. We would like to kindly ask if we have addressed all of your concerns. If so, could you increase your rating accordingly? If there are any remaining concerns on experiments or theoretical analysis, please let us know so that we can address them promptly.

---

> > ### Comment · Reviewer_zWMt · 2024-11-25
> > **Response to the Authors' Rebuttal**
> >
> > Thank you for your reply and engaging with my comments. I will maintain my score for the following reasons:
> > 1. The paper argues that energy-based training results in a smaller error in the regression target. However, during sampling, the authors still need to take the gradient and use the score. It is unclear to me whether the lower L2 error of the energy translates to the lower L2 error in the energy and consequently a better performance. Can the authors provide theoretical bounds in the error of the score when doing energy-based training?
> > 2. The authors’ response for why a high variance target is different in sampling vs data-based training was not convincing. In both cases when trained for long enough the network will regress to the mean of the target. I also think the authors are confusing bias and error in their point (3) when they state: "It’s clear that, due to the high variance of the training target, the learned scores can be much biased to the ground truth one." and in the revision when they state " In general, the high variance of training target introduces high bias of the learned value." The estimator has an inherent bias which is further increased due to bootstrapping. The authors call this a “small error (bias)” without providing any evidence for why they find this error is small.
> > 3. Additionally, I do not find the experimental results convincing that bootstrapping is beneficial in complex energy landscapes, as evident by the worse performance of BNEM compared to NEM in the E-W2 metric for LJ55.  I think the results could be strengthened if the authors could provide a setting where NEM suffers and where BNEM clearly helps with the learning.
> > 4. Regarding the reported performance of the baseline models, I am unfortunately not convinced of the results the authors report. For example, for FAB, [1] also reports energy-based training results for LJ13 and [2] states that they used that architecture for their equivariant tasks. [2] also seems to be providing for PIS: https://github.com/jarridrb/DEM/blob/main/dem/models/pis_module.py. Could the authors also clarify how their setup was different in LJ13 for iDEM as they were unable to reproduce the results there as well?
> >
> > [1] Midgley, L. I., Stimper, V., Simm, G. N., Scholkopf, B., and Hernandez-Lobato, J. M. Flow annealed importance sampling bootstrap. International Conference on Learning Representations (ICLR), 2023b.
> >
> > [2] Tara Akhound-Sadegh, Jarrid Rector-Brooks, Avishek Joey Bose, Sarthak Mittal, Pablo Lemos, Cheng-Hao Liu, Marcin Sendera, Siamak Ravanbakhsh, Gauthier Gidel, Yoshua Bengio, Nikolay Malkin, and Alexander Tong. Iterated denoising energy matching for sampling from boltzmann densities, 2024.

---

> > > ### Author Response · Authors · 2024-11-25
> > > **Official Response to Reviewer zWMt (Part I)**
> > >
> > > Dear reviewer zWMt,
> > >
> > > Thank you for your response and comments. We believe that the below explanations would address your concerns.
> > >
> > > ## **Q1: Theoretical Bound for scores obtained by energy-based model**
> > >
> > > It’s a good point that although the $l_2$-error of energy is much lower than that of score, the learned energy need to be differentiated when sampling. Unfortunately, we didn’t find a clean way to theoretically derive a bound for that, which is claimed in our manuscript. However, all experiment results convince that scores obtained by differentiating the learned energies are **better** than learning the scores directly. On the other hand, the learned energy allows us to combine more techniques like Metropolis-Hastings beyond denoising diffusion sampling. Therefore we conclude that at the current stage, energy models are better than score models practically and have more potential to further gain improvement. Deriving a theoretical bound showing that $\text{error}(\nabla E_\theta) < \text{error}(s_\theta)$ would be one of our main future works.
> > >
> > > ## **Q2.1: Training with high variance targets**
> > >
> > > We would like to explain this question in a clearer way. Firstly, for any diffusion models, the goal is to learn the marginal score, *i.e.*
> > >
> > > $$
> > > s_t(x_t)=\nabla_{x_t}\log \int p_0(x_0)\mathcal{N}(x_t|x_0)dx_0
> > > $$
> > >
> > > $$\min_\theta \mathbb{E}||s_\theta(x_t, t) - s_t(x_t)||^2$$
> > >
> > > ### In data-based diffusion models
> > >
> > > The above objective can be further simplified by using Denoising-Score-Identity ($s_t(x_t) = \int p(x_0|x_t)\nabla_{x_t}\log\mathcal{N}(x_t|x_0)dx_0$) and replacing the posterior term $p(x_0|x_t)$ with the joint distribution $p_0(x_0)p(x_t|x_0)$. Since we have data $x_0\sim p_0$, we can simply draw $x_0$ and its noisy version $x_t$, then optimize a simple l2-loss $s_\theta(x_t, t) - \nabla_{x_t}\log\mathcal{N}(x_t|x_0)$. In such case, we **DO NOT** need to worry about the variance of target, since $\nabla_{x_t}\log\mathcal{N}(x_t|x_0)$ is NOT what we want. Instead, we want $s_t(x_t) = \int p(x_0|x_t)\nabla_{x_t}\log\mathcal{N}(x_t|x_0)dx_0$, which is **implicitly optimized** in data-based diffusions.
> > >
> > > ### In our setting
> > >
> > > We **explicitly** target $s_t(x_t)$, since no data is available. In this case, we need to **estimate** $s_t(x_t)$ using the MC estimator as our training target (remember that, there is **NO** estimation in data-based diffusions). Therefore, the variance plays an important role here, which can have a great impact on model training. Training with a smaller variance object can always result in a more stable training process and better results. Our experimental results highly convince this.

---

> ### Author Response · Authors · 2024-11-25
> **Official Response to Reviewer zWMt (Part II)**
>
> ## **Q2.2: Bias and Error**
>
> We apologize for making any confusion. We would like to explain why **BNEM can improve performance in terms of less bias** in a more intuitive way.
>
> Firstly, we should notice that the “true” target of our energy model is the noise-energy $\mathcal{E}_t(x)$. However, we now only can get a **biased** estimator $E_K(x, t)$. Therefore, it would introduce the first mismatch, we named it **estimation bias**. And this **estimation bias** can be increased along time $t$ (see Proposition 1 and its proof in Appendix A).
>
> Secondly, we tend to train an energy model to fit those energy estimators. The variance of the training target can introduce the second mismatch that the learned energy is NOT perfect. And typically in NEM, when $t$ increases the variance of training target increases, which results in larger mismatches. We name this mismatch between learned energy $E_\theta(x, t)$ and the original MC estimator $E_K(x, t)$ **learning bias**, which is positively correlated to the variance of the training target.
>
> Therefore, in NEM, the **true bias** between our learned energy $E_\theta(x, t)$ and its “true” target $\mathcal{E}_t(x)$ is contributed by both **estimation bias** and **learning bias**.
>
> On the other hand, BNEM bootstraps the learned energy at a slightly smaller time $s$ ($s<t$), resulting in smaller **learning bias** and slight larger **estimation bias**.
>
> In Proposition 3, we theoretically compare the **estimation bias** between NEM and BNEM at time $t$, where the **estimation bias at t** of BNEM is contributed by all **estimation bias at s** and **learning bias at s** with some rescaling factors. Proposition 3 shows that, their **estimation bias at t** can be derived as:
>
> NEM: $\frac{v_{0t}(x)}{2m_t(x)^2K_1}$
>
> BNEM: $\frac{v_{0t}(x)}{2m_t^2(x)K_2^{i+1}} + \sum_{j=1}^i\frac{v_{0s_j}(x)}{2m_{sj}^2(x)K_2^j}$
>
> For example, consider we learn noised energy at $s$ by NEM, and learn noised energy at $t$ by bootstrapping the learned energy at $s$, and we use the same $K_1=K_2=K$. Then the **estimation bias** between NEM and BNEM at $(x, t)$ become
>
> $\text{Bias}^{\text{NEM}}(x, t)=\frac{v_{0t}(x)}{2m_t(x)^2K}$
>
> $\text{Bias}^{\text{BNEM}}(x, t; s)=\frac{v_{0t}(x)}{2m_t^2(x_t)K^{2}} +\frac{v_{0s}(x)}{2m_{s}^2(x)K}$
>
>
> It shows that $\text{Bias}^{\text{BNEM}}(x, t; s)=\frac{1}{K}\text{Bias}^{\text{NEM}}(x, t) + \text{Bias}^{\text{NEM}}(x, s)$. Typically, $\text{Bias}^{\text{NEM}}(x, s)$ is much smaller and $K$ can reduce $\text{Bias}^{\text{NEM}}(x, t)$ much, Therefore, we can have $\text{Bias}^{\text{BNEM}}(x, t; s)<\text{Bias}^{\text{NEM}}(x, t)$.
>
> Our additional experiments also convince that BNEM can reduce such **estimation bias** to gain improvement. For **learning bias**, since the variance of BNEM’s target is always smaller than NEM’s (you can intuitively think that we’re sampling in a smaller neighbourhood region when estimating the bootstrapped energy estimation in BNEM), the **learning bias** of BNEM can be smaller. Therefore, we can also gain improvement in terms of reducing **learning bias**.

---

> ### Author Response · Authors · 2024-11-25
> **Official Response to Reviewer zWMt (Part III)**
>
> ## **Q3: Worse E-W2 in BNEM and negative tasks for NEM**
>
> Thank you for pointing out the the previous E-W2 of BNEM is worse than NEM. We realize there’re few implementation issues a few days ago, so we rerun the experiments on LJ-55. Our rerun **1.0 version**, the one we shown in our first response, still has some issues and we were still waiting for the rest of experiments at that time due to shortage of computational resources in this week. Fortunately, the remaining experiments are finished now, and we would like to provide you the latest results (**2.0 version**) on LJ-55: https://anonymous.4open.science/r/nem_rebuttal-0D92/main-table-new.png. It shows that **BNEM** achieves similar mean E-W2 (**119.46** v.s. 118.58), while having less variance (**77.92** v.s. 106.63). We believe that it can convince the improvement gained by BNEM.
>
> We found that NEM can consistently outperform iDEM in the providing tasks, which are considered in [1]. We therefore cannot provide any negative example that NEM fails at the current stage. But we will conduct more experiments on larger systems like ALA2 in the future. Nevertheless, we believe that the improvement gained by NEM also convince our motivation and theoretical results, which suggests learning less variated targets and has more potential to improve performance *e.g.* bootstrapping or Metropolis-Hastings (future work).
>
>
> [1] Tara Akhound-Sadegh, Jarrid Rector-Brooks, Avishek Joey Bose, Sarthak Mittal, Pablo Lemos, Cheng-Hao Liu, Marcin Sendera, Siamak Ravanbakhsh, Gauthier Gidel, Yoshua Bengio, Nikolay Malkin, and Alexander Tong. Iterated denoising energy matching for sampling from boltzmann densities, 2024.
>
>
> ## **Q4: Different architectures for baselines and Inreproducibility of iDEM in LJ13**
>
> For the baselines, FAB, PIS, and DIS, we change their model architecture to be the same as iDEM and ours, to ensure all models have the same number of parameters for fair comparison.
>
> For LJ-13 and LJ-55, it’s true that our results cannot reproduce the ones provided in iDEM [1]. We didn’t change the setting of iDEM, rerun their experiments on both LJ-13 and LJ-55, and reported the results. Especially for LJ-55, we found that training iDEM is highly sensitive to random seeds. We also found that few people **raised issues** in iDEM's repo (https://github.com/jarridrb/DEM) about the reproducibility of iDEM on LJ tasks, which is **hard to reproduce** and can have a very **unstable training process**. We believe that, to the best of our knowledge, reproducing the results in [1] is a common challenge.
>
> [1] Tara Akhound-Sadegh, Jarrid Rector-Brooks, Avishek Joey Bose, Sarthak Mittal, Pablo Lemos, Cheng-Hao Liu, Marcin Sendera, Siamak Ravanbakhsh, Gauthier Gidel, Yoshua Bengio, Nikolay Malkin, and Alexander Tong. Iterated denoising energy matching for sampling from boltzmann densities, 2024.
>
>
> ## **End of Response**
>
> Once again, thank you for your response and constructive comments. We sincerely appreciate your feedback. We kindly ask if our explanations, intuitions, and theories were conveyed clearly and whether we have adequately addressed your concerns. If you have any remaining questions or uncertainties, please do not hesitate to let us know.

---

> ### Author Response · Authors · 2024-11-27
> **Official Response to Reviewer zWMt (additional explanation on bias of bootstrapping)**
>
> Dear reviewer zWMt,
>
> Thank you very much for the discussion! We would like to introduce a more intuitive way to show **why bootstrapping can be beneficial**. We hope this additional content can address your concerns.
>
> ## **Additional Explanation on Bias of Bootstrapping**
>
> Suppose for $0\leq s<t\leq 1$, we learn $E_\theta(x_s, s)$ by NEM, *i.e.* targeting $E_K(x_s, s)$ and learn $E_\theta(x_t, t)$ by bootstrapping from the learned values at $s$.
>
> Let’s forget about the randomness of $E_K(x_s, s)$ first. Then the optimal network predicts $E_{\tilde\theta}(x, s)=E_K(x, s)$ for $\forall x$. In such case, the bootstrapping estimator becomes:
>
> $$E_K^{B(n)}(x_t, t, s; \tilde \theta) = -\log\frac{1}{K}\sum_{i=1}^K\exp\left(-E_K(x_{s|t}^{(i)}, s)\right)=-\log\frac{1}{K}\sum_{i=1}^K\frac{1}{K}\sum_{j=1}^K\exp(-\mathcal{E}(x_{0|t}^{(ij)}, t))$$
>
> where $x_{s|t}^{(i)}\sim\mathcal{N}(x;x_t, (\sigma_t^2-\sigma_2)I)$ and $x_{0|t}^{(ij)}\sim\mathcal{N}(x;x_{s|t}^{(i)}, \sigma_s^2I)$. Therefore, we can have
>
> $$E_K^{B(n)}(x_t, t, s; \tilde \theta)=E_{K^2}(x_t, t)$$
>
> which means that **ideally** the bootstrap(1) estimator (with bootstrapping once corresponding to a bootstrapping chain $0\rightarrow s\rightarrow t$) is equivalent to using **quadratically** more MC samples. By induction, if we keep bootstrapping for several times, *e.g.* via a chain $0\rightarrow s_1\rightarrow s_2\rightarrow...\rightarrow s_n=s\rightarrow s_{n+1}=t$, then the bootstrap(n) estimator at $t$ can reduce the variance and bias of $E_K$ **polynomially** w.r.t. $K$.
>
> However, since $E_K$ is a random variable, the optimal neural network can only fit its mean. While $E_K$ is biased, its mean equals to the true noised energy $\mathcal{E}_t$ plus a bias-term, which is also half of its variance, *i.e.* $\mathbb{E}[E_K(x_t, t)]=\mathcal{E}_t(x_t)+\mathrm{Bias}[E_K(x_t, t)]$ and $\mathrm{Bias}[E_K(x_t, t)]=\mathrm{Var}[E_K(x_t, t)]/2$.
>
> Therefore, bootstrapping can polynomially reduce the original variance and bias of $E_K$ but involves extra bias inherited from the last noise level, *i.e.*
>
> For $t\in[s_1, s_2]$ and $s\in[0, s_1]$, $\mathrm{Bias}[E_K(x_t, t, s;\theta)]=\frac{v_{0t}(x_t)}{2m_t^2(x_t)K}=\mathrm{Bias}[E_K(x_t, t)]$
>
> For $t\in[s_2, s_3]$ and $s\in[s_1, s_2]$, $\mathrm{Bias}[E_K(x_t, t, s;\theta)]=\frac{v_{0t}(x_t)}{2m_t^2(x_t)K^{2}}+\frac{v_{0s}(x_t)}{2m_{s}^2(x_t)K}
> $
>
> For $t\in[s_3, s_4]$ and $s\in[s_2, s_3]$, $\mathrm{Bias}[E_K(x_t, t, s;\theta)]=\frac{v_{0t}(x_t)}{2m_t^2(x)K^{3}}+\frac{v_{0s}(x_t)}{2m_{s}^2(x_t)K^2}+\frac{v_{0s_1}(x_t)}{2m_{s_1}^2(x_t)K}
> $
>
> We can control the bootstrapping parameters as well as $K$ such that $\mathrm{Bias}[E_K(x_t, t, s;\theta)]<\mathrm{Bias}[E_K(x_t, t)]$. Clearer mathematical discussions can be found in the revised 3.4 section and the referenced contents in our appendix.
>
> ## **End of Response**
>
> Again, thank you so much for discussing this with us. If you have any remaining concerns or questions, please do not hesitate to let us know!

---

> ### Comment · Reviewer_zWMt · 2024-11-27
>
> Thank you for your response. I have carefully read your explanations and will maintain my score for the following reasons:
> 1. I believe that not providing a bound on the error of the score when learning about energy is a significant limitation of the work. In fact, the theoretical discussions the authors provide do not apply to what we are interested in, which is the score used for sampling. As far as I can tell, the current theoretical results cannot be used to bound the variance of the score. This is very concerning and limits the theoretical contributions of the paper significantly.
> 2. I think the authors have a misunderstanding regarding the effect of the variance in the data-based training of diffusion models. Reducing the variance in the estimated score is also important in diffusion models and can lead to an improvement in their performance. This is why researchers are interested in flow-matching models or optimal transport and many works have shown that reducing the variance of the score/flow estimator improves performance. Therefore, claiming that it is not a concern in data-based training is not accurate [1, 2, 3].
> 3. I understand the authors' point regarding the learning vs. estimation bias. Firstly, the term "learning bias" is misleading and they should refer to this as a learning error since the term bias refers to an irreducible error. Conversely, this "learning bias" will, in theory, go to 0 when trained for long enough. However, the estimation bias is irreducible and causes significant problems, especially in high-dimensional settings.
> 4. Can the authors clarify how they used the same architecture for FAB when it is a normalizing flow and hence requires an invertible architecture? Additionally, besides the results of iDEM, I am concerned about how the authors failed to produce results for PIS on LJ13.
>
> [1] Daegyu Kim, Jooyoung Choi, Chaehun Shin, Uiwon Hwang, and Sungroh Yoon. Improving diffusion-based generative models via approximated Optimal Transport, 2024.
>
> [2] Tong, A., Malkin, N., Huguet, G., Zhang, Y., Rector-Brooks, J., Fatras, K., Wolf, G., and Bengio, Y. Improving and generalizing flow-based generative models with minibatch optimal transport, 2023.
>
> [3] Yiheng Li, Heyang Jiang, Akio Kodaira, Masayoshi Tomizuka, and Kurt Keutzer. Immiscible Diffusion: Accelerating Diffusion Training with Noise Assignment, 2024.

---

> > ### Author Response · Authors · 2024-12-01
> > **Official Response to Reviewer zWMt (Part I)**
> >
> > Dear Reviewer zWMt,
> >
> > Thank you very much for the discussion and your constructive comments. We hope our response could address your concerns!
> >
> > ## **Q1: Bound on Learned Scores in NEM**
> > We fully agree with the reviewer that the error for the learned scores (w.r.t. the ground-truth scores) provided by NEM is of practical interest. Additionally, this is also necessary for iDEM, which is not provided in [1]. In fact, [1] only offers an error bound for the score estimator, whereas the error for an optimally learned score network pertains to the **expectation** of the score estimator. We appreciate the reviewer for highlighting this, and we have since figured out how to quantify this error term. To outline, we characterise two kinds of “error”:
> >
> > 1. Bias of Learned Scores (**irreducible**): due to the bias of training targets $E_K$ (for NEM) and $S_K$ for iDEM
> >
> > 2. Error of Learned Scores (**reducible**): due to the imperfectness of neural network
> >
> > ### **(1) Bias of Learned Scores**
> >
> > To derive the bias for learned scores given by NEM and iDEM, we leverage the sub-Gaussian assumption and derive that:
> >
> > $$\nabla_{x_t}E_{\theta*}(x_t, t) = (1+\frac{1}{K})S_t(x_t) +\frac{1}{K}\int\exp(-2\mathcal{E}(x)+2\mathcal{E}_t(x_t))(\nabla\mathcal{E}_t(x_t)-\nabla\mathcal{E}(x))\mathcal{N}(x_t;x, \sigma_t^2I)dx $$
> >
> > Therefore, the bias of learned scores is given by:
> >
> > $$||\nabla_{x_t}E_{\theta*}(x_t, t)-S_t(x_t) ||= ||\frac{1}{K}S_t(x_t) -\frac{1}{K}\int\exp(-2\mathcal{E}(x)+2\mathcal{E}_t(x_t))(\nabla\mathcal{E}_t(x_t)-\nabla\mathcal{E}(x))\mathcal{N}(x_t;x, \sigma_t^2I)dx||$$
> >
> > which can become very accurate for small $t$.
> >
> > In fact, the scores given by the **optimal** NEM (i.e. with 0 learning error) are equivalent to the ones given by the **optimal** iDEM, i.e. $\nabla_{x_t}E_{\theta^*}(x_t, t)=S_{\phi^*}(x_t, t)$. Therefore, the previous error is also for the learned scores in iDEM, which is not provided in [1]. The proof is provided in https://anonymous.4open.science/api/repo/nem_rebuttal-0D92/file/error_score.pdf?v=fe4c67c7.
> >
> > We also derive the **bias of learned score for BNEM** in C.2 in https://anonymous.4open.science/api/repo/nem_rebuttal-0D92/file/error_score.pdf?v=fe4c67c7
> >
> > ### **(2) Error of Learned Scores**
> >
> > Due to the imperfectness of neural network, where training time is not infinitely long and the network architecture is not infinitely flexible, both $E_\theta$ and $s_\theta$ are slightly different from their optimalities $E_{\theta^*}(x_t, t)=\mathbb{E}[E_K(x_t, t)]$ and $s_{\theta^*}(x_t, t)=\mathbb{E}[S_K(x_t, t)]$, *i.e.* $E_\theta(x_t, t)=\mathbb{E}[E_K(x_t, t)]+e_\mathcal{E}(x_t, t)$ and $s_\theta(x_t, t)=\mathbb{E}[E_K(x_t, t)]+e_S(x_t, t)$. And in diffusion sampling, we are interested in the scores given by the learned networks and their errors, which are $\nabla_{x_t}E_\theta(x_t, t)=\nabla_{x_t}\mathbb{E}[E_K(x_t, t)]+\nabla_{x_t}e_\mathcal{E}(x_t, t)$ and $s_\theta(x_t, t)=\mathbb{E}[E_K(x_t, t)]+e_S(x_t, t)$.
> >
> > As a consequence of the small variance of $E_K$, regressing $E_K$ is practically **easier** than regressing $S_K$. We assume that $e_\mathcal{E}(x_t, t)$ is **$L$-Lipchitz**, which results that the differentiated learning error in NEM can be bounded by $L$, *i.e.* $||\nabla_{x_t}e_\mathcal{E}(x_t, t)||\leq L$. Empirical experiments, where NEM surpasses iDEM in all tasks, show that the Lipchitz constant of the learning error in NEM can be small, resulting in smaller error in scores and better performance.
> >
> > [1] Tara Akhound-Sadegh, Jarrid Rector-Brooks, Avishek Joey Bose, Sarthak Mittal, Pablo Lemos, Cheng-Hao Liu, Marcin Sendera, Siamak Ravanbakhsh, Gauthier Gidel, Yoshua Bengio, Nikolay Malkin, and Alexander Tong. Iterated denoising energy matching for sampling from boltzmann densities, 2024.
> >
> > ## **Q2: Effect of Variance**
> > We agree that the variance does affect the training process and model performance. And that’s absolutely why we are claiming that NEM is better than iDEM. In optimal case, where we assume training infinitely long and use infinitely flexible networks, the score learned by NEM (differentiating the learned energy) is **equivalent** to the one learned by iDEM. However, the variance of the energy estimator (NEM) is smaller than that of the score estimator (iDEM). This leads regressing $E_K$, *i.e.* NEM, to an easier task with smaller learning error compared to iDEM. A detailed discussion about this was shown in the above answer to **Q1**. In practice, the smaller variance of BNEM's training target can also benefit for **reducing the learning error**.

---

> > > ### Author Response · Authors · 2024-12-01
> > > **Official Response to Reviewer zWMt (Part II)**
> > >
> > > ## **Q3: problems about bias**
> > > We thank the reviewer for pointing out that using “bias” is somehow unclear and we also appreciate for pointing out that the learning error (which we meant learning bias) is reducible and estimation bias is irreducible. In fact, our **proposition 3** shows that BNEM can have smaller **estimation bias** compared to NEM, once within proper bootstrapping parameter setting.
> > >
> > > In addition, the smaller variance of BNEM's training target can also benefit for **reducing the learning error** as we mentioned in **Q2**. Therefore, with proper bootstrapping setting, BNEM can have **smaller estimation bias** as well as **smaller learning error**, and the experiments show that BNEM achieves the best performance empirically.
> > >
> > > ## **Q4: clarifying settings in baseline models**
> > > Regarding the reproduction of the results for LJ13 using PIS, we attempted to use the model defined in this link. However, the authors of the iDEM paper did not provide the specific experiment configurations they used. Furthermore, since PIS is not a simulation-free method, the model configurations they provided use fewer parameters compared to iDEM. We believe that using the same model architecture is more critical than the number of integration steps. Therefore, we applied the same model architecture and used 100 integration steps during training. However, neither in this setting nor with 1000 integration steps, using the smaller model from the iDEM repository, did we achieve a converged model. If you have a valid configuration that successfully reproduces the LJ13 results, we would be more than happy to give it a try!
> > >
> > > ## **End of Response**
> > >
> > > Again, thank you very much for pointing out your concerns. We sincerely appreciate your comments and hope our response could address your concerns. Please do not hesitate to let us know if you have any remaining concerns or questions.

---

> > > > ### Author Response · Authors · 2024-12-03
> > > > **Reminder for reviewer comment deadline**
> > > >
> > > > Dear Reviewer zWMt,
> > > >
> > > > Thank you for your valuable comments and insightful discussions. We would like to kindly remind you that the deadline for posting messages to the authors is approaching (December 2nd, AOE).
> > > >
> > > > If you have any further comments or concerns regarding our previous response, we would greatly appreciate hearing from you before the deadline.
> > > >
> > > > Thank you once again for your time and thoughtful feedback.

---

### Official Review · Reviewer_ajpL · 2024-11-04

**Soundness:** 3
**Presentation:** 3
**Contribution:** 3
**Rating:** 8
**Confidence:** 4

**Summary:**

The authors consider a setting in which we want to draw samples from a Boltzmann distribution, for which we have the ground truth energy function, but cannot normalize into a probability distribution (for the usual reasons).  A very recent proposal (Akhound-Sadegh et al., 2024, "iDEM") for this problem is to model the "score function" (in Hyvarinen's sense) with a neural network, as in a diffusion model; but, lacking data samples, to target not the empirical score (via Tweedie's formula) but rather a Monte Carlo estimate written in terms of the energy function.  The MC samples are drawn in an outer loop by simulating the reverse diffusion process, using the current score-estimating network.

The present MS proposes to alter iDEM slightly, by modeling the *energy* rather than the score--again by targeting a MC estimate, in this case of the energy itself rather than its gradient.  They prove that (under certain conditions--I did not read the proof in detail) the error of iDEM's score estimator is strictly larger than the error of their own energy estimator; and (empirically) than the error in the score function that is produced by differentiating their energy estimator (and which matters because it will be used for sampling).  To reduce variance in this estimator further still, the authors further propose a "bootstrapping" technique in which the target energy is not constructed from the known data energy, but instead estimated from a model energy at a smaller noise level.

Empirically, the authors show that their approach yields superior samples to iDEM as well as other recent proposals, particularly with the "bootstrap" improvement.

**Strengths:**

The MS attacks an important problem and improves upon the state of the art.  The solution is original (to the best of this reviewer's knowledge) and provides theoretical as well as empirical reasons to prefer their proposal over alternatives.

**Weaknesses:**

(1) On the GMM-40 task, the manuscript's version of iDEM (Fig. 2) is visibly inferior to the one in the original paper (Akhound-Sadegh et al, Fig. 3), which is much closer to ground truth.  Something similar is true of the energy histogram for LJ-55 (Fig. 3 in this MS, Fig. 4 in op. cit.).  Can the authors explain these discrepancies?

(2) For DW-4, LJ-13, and LJ-55, the improvement in Wasserstein-2 distance provided by NEM and BNEM over iDEM appears marginal (Table 1).  Considering the additional computational cost of evaluating the energy gradient, this may not justify the proposed approach.

(3) The reported W2-E for iDEM is high, potentially due to outlier samples as pointed out in the manuscript. It would be nice to re-evaluate W2-E removing outliers for comparison with NEM and BNEM.

(4) Learning energy instead of score may improve estimation, but sampling requires costly repeated gradient evaluations as steps increase.  A table comparing sampling times for score-based (iDEM) and energy-based (NEM) training would be helpful.

(5) The descriptions of the problem setting and proposed solution could be much improved; this reviewer, at least, found the exposition in Akhound-Sadegh et al. (2024) much clearer.

(6) The Fisher divergence (Eq. 8) is a well understood quantity, with connections to KL divergence---e.g., in the context of diffusion models, it arises naturally under the standard ELBO loss.  Can the authors provide any similar theoretical appeal for the squared error in energy, Eq. 10?



MINOR
Diffusion models were introduced by Sohl-Dickstein et al. in "Deep unsupervised learning using nonequilibrium thermodynamics" (ICML, 2015) and this paper should be referenced when they are introduced in the MS.

**Questions:**

See "Weaknesses" above.

---

> ### Author Response · Authors · 2024-11-22
> **Official Response to Reviewer ajpL (Part I)**
>
> Dear reviewer ajpL,
>
> Thank you very much for your recognition for our methods, your detailed feedback, and your constructive comments. We believe that our responses and the revised manuscript could directly address your concerns. We hope these additions encourage you to adjust your rating accordingly. If any questions or concerns arise, please do not hesitate to let us know, and we will address them properly. Before starting, thank you for pointing out that we missed to reference the paper by Sohl-Dickstein et al , which is already referenced now.
>
> **Q1: Explanation of the performance discrepancies compared to the iDEM paper**
>
> It is true that our visualized performance is different from the baseline model. Our experiment is running in a more challenging setting.
> For the GMM experiment, we reduced both the number of integration steps and the number of Monte Carlo (MC) samples from 1000 to 100. If we adopt the same setting in the iDEM paper there will not be notable performance differences observed in visualizations.
> For the LJ-55 experiment, we found that training iDEM is highly sensitive to random seed selection. Consequently, reproducing iDEM’s results without the specific random seed provided is nearly impossible. To ensure fairness, we reran their code and selected the best-performing seed for all models when generating plots.
>
> **Q2: Balance between performance improvement and computational cost**
>
> We emphasize that, due to the differentiation of NN, the computational cost of both NEM and BNEM during sampling is approximately TWICE as that of iDEM. Therefore, we conduct experiments with a similar computation setting (1000 step iDEM v.s. 500 step NEM/BNEM) on LJ-13 and LJ-55. You can find the results in this link: https://anonymous.4open.science/r/nem_rebuttal-0D92/half_integration_step_comparison.png
> which shows that with similar computations NEM/BNEM can still outperform iDEM. We can also improve more by doubling the computation during sampling.
>
> Besides, (Fig. 5, see: https://anonymous.4open.science/r/nem_rebuttal-0D92/lj13-100.png) in our manuscript shows that NEM and BNEM can even achieve better results by reducing their computations in sampling to 1/10, which means they are 5 times faster than iDEM.
>
> **Q3: Experimental results of removing outlier**
>
> Based on (Fig 4.), we set the maximum energy as (GMM: 100; DW4: 0; LJ13: 0; LJ55: -150). We removed outliers based on this threshold and recomputed the E-W2. We report the new E-W2 as well as the percentage of outliers. The results can be found in https://anonymous.4open.science/r/nem_rebuttal-0D92/remove_outliers.png, which shows that
> the order of performance (BNEM>NEM>iDEM) still holds in terms of better E-W2 value and less
> percentage of outliers. It also indicates that without the influence of outliers, samples generated by NEM and BNEM are much more similar to the ground-truth ones.
>
> **Q4: Comparison of training time and sampling time**
>
> We include a time complexity comparison (see https://anonymous.4open.science/r/nem_rebuttal-0D92/time_complexity.png), which theoretically shows that during training NEM can be slightly faster than iDEM in the inner-loop while it uses double computation in the outer-loop (i.e. sampling) due to the neural network differentiation. The inner-loop training time of BNEM depends on the relative complexity of evaluating the clean energy function versus the neural network, which can be relatively higher.
>
> We conduct time logging for iDEM, NEM, and BNEM (see https://anonymous.4open.science/r/nem_rebuttal-0D92/run_time_comparison.png). Empirically, it shows that the sampling time, i.e. outer-loop, of BNEM/NEM is approximately twice that of iDEM, which remains within the same magnitude as iDEM. In contrast, the inner-loop time of NEM is slightly faster than that of iDEM, and the difference becomes more pronounced for more complex systems. For BNEM, the sampling time is comparable to NEM, but the inner-loop time depends on the relative complexity of evaluating the clean energy function versus the neural network, which can be relatively higher.
>
> Based on these observations, we recommend using NEM for complex target energy functions in large systems. When sufficient computational resources are available, BNEM can be employed to trade time for improved accuracy.
>
> We also proposed another estimator by using Tweedie’s formula, which are able to bypass the memory issue of differentiating NN to evaluating NN twice. And the two times of NN evaluations can be definitely computed parallelly, resulting in a similar speed as iDEM but just doubling the memory usage. The proposed method can be found in (Appendix F: Memory-Efficient NEM). Experimental results are provided in https://anonymous.4open.science/r/nem_rebuttal-0D92/memory_efficient_nem.png
>
> [FYI: the experiments on LJ-55 are still running. We’ll update them once they are finished.]

---

> ### Author Response · Authors · 2024-11-22
> **Official Response to Reviewer ajpL (Part II)**
>
> **Q5: Improvement of method presentation**
>
> Thank you for your suggestion on clarifying our manuscript. We have refined our method section to enhance clarity and readability. Specifically, we have added a more comprehensive introduction to the overall framework before diving into the detailed analysis of the variance and bias of the proposed Monte Carlo estimator for energy, as well as the bootstrapping technique. Revised contents are colored as blues
>
> **Q6: Theoretical appeal from KL div /Fisher div for Eq. 10**
>
> In fact, (Eq.10) doesn’t relate to any f-Divergence, but simply a $L_2$ error of the energy. But there’re still theoretical advantages behind optimizing this $L_2$ error.
>
> A previous work [1] shows that for any two $\mathbb{R}^d$ GMM with same modes but different weights, the Fisher divergence between them can be arbitrary small but the KL divergence between them remains large when two modes are sufficiently separate (See Example 1 and Theorem 6).
>
> In addition, their Figure 1 shows that  (1) In left, the Fisher divergence can NOT distinguish two GMM with same modes but different weights, while KL divergence can and $L_2$ error can be even better.(2) In right shows that the perturbed score can not tell the difference of weights until the every end of the forward process. On the other hand, it is noticeable that the perturbed log-density distinguishes the weights well throughout the forward process (Figure 1, middle).
>
> [1] Shi, Zhekun, et al. "Diffusion-PINN Sampler." arXiv preprint arXiv:2410.15336 (2024).
>
> **End of Response**
>
> Thank you again for your reviews and detailed comments. We hope that our responses have effectively addressed all of your concerns. If so, could you increase your score accordingly? If there are any issues or further clarifications needed, please let us know. We will make every effort to address them promptly.

---

> > ### Comment · Reviewer_ajpL · 2024-11-27
> >
> > Thanks.  I'm still concerned that, on a fixed computational budget, the proposed methods are not obviously better than iDEM.  I understand the authors' argument that, for larger systems, the inner loop dominates (but is this true for BNEM as well as NEM?).  In any case I retain my score.

---

> > > ### Author Response · Authors · 2024-11-27
> > > **Official Response to Reviewer ajpL**
> > >
> > > Dear reviewer ajpL,
> > >
> > > Thank you very much for discussing this. We hope our response could address your remaining concerns.
> > >
> > > In fact, we do not think comparing NEM and iDEM with different integration steps is fair, because diffusion models can be highly affected by the sampling steps and perform much worse without any advanced sde/ode solver. However, it’s impressive that our methods still achieve decent performances in this challenging setting. We could still further increase the integration steps during training or sampling for iDEM *e.g. 2000 steps*, but compared to NEM/BNEM the improvement we gain is limited.
> > >
> > > On the other hand, for BNEM in large systems, the inner loop computation should be decreased accordingly, because we employ bootstrapping for $t\geq s_0$. Therefore, since evaluating the clean energy is much more expensive than evaluating the NN in this case, we can bypass a large part of the computation in the inner-loop and therefore could speed up a bit. But the exact experiments for BNEM on large system, *e.g.* ALA2, would be our future work, and we’re still working on training NEM on ALA2 at the current stage.
> > >
> > > Again, thank you very much for your recognition of our work. We also sincerely appreciate your constructive comments and the helpful discussions. Please let us know if you have any remaining concerns or questions!

---

### Official Review · Reviewer_MWcQ · 2024-11-04

**Soundness:** 4
**Presentation:** 3
**Contribution:** 3
**Rating:** 8
**Confidence:** 4

**Summary:**

The paper presents a neural sampling method for known energy functions. The importance of this problem is well-established, and the distinction between this and methods learned from sample data is sound. The NEM method improves on previous neural samplers by using an energy-based training objective, using a lower noise target, and introducing a bootstrapping method. The results show improvements over other listed neural samplers.

**Strengths:**

The paper describes the method very clearly. In particular, figure 1 is quite concise. Moreover, it shows proof that the improvements suggested are indeed improvements. Section 3 is well written, and the score vs. energy section contribution is valuable.

The results are strong and show improvement over other methods for the toy problems.

**Weaknesses:**

While the paper clearly shows improvement over previous neural sampling methods, it still makes little or no progress on the fundamental challenge of scaling to larger systems. The authors acknowledge this, but it ultimately make this an incremental improvement, not a transformational one.

**Questions:**

Why is the distribution of interatomic distances for NEM and BNEM less sharply peaked than the ground truth for  LJ-55?

Have the method been tried on any larger, more complicated systems? Even if the results are not impressive, it would be useful for the field to know when this method breaks down.

---

> ### Author Response · Authors · 2024-11-23
> **Official Response to Reviewer MWcQ**
>
> Dear reviewer MWcQ,
>
> We would like to start by thanking you for highlighting our strengths, including the concise figure 1, the theories, and the results. We hope that our detailed responses and the updates in the revised manuscript reaffirm your satisfaction with the paper. We also hope these additions encourage you to increase your rating accordingly. If any questions or concerns arise, please do not hesitate to let us know, and we will address them properly.
>
> **Weakness1: Paper makes little or no progress on the fundamental challenge of scaling to large systems**
>
> Thank you for pointing this out. Firstly, we would like to say that NEM and BNEM are transformational rather than incremental. Here’s the reason: Generating high-quality samples from the target distribution is a fundamental sub-task in sampling. Beyond this, it is often essential for the generated samples to be assigned correct weights under the target distribution. However, several studies have noted that score-based sampling methods can be problematic in this context (e.g., [1][2]), where modes tend to be covered but are associated with inaccurate weights. While [1] can generate samples with correct weights once applying Metropolis-Hastings. Notably, iDEM, NEM, and BNEM cannot achieve this (see experiments on DW-4 and additional experiments on MW32 https://anonymous.4open.science/r/nem_rebuttal-0D92/mw32_prelim.png). However, NEM and BNEM enable the estimation of unnormalized marginal densities, i.e., exponential negative noised energies. This allows more techniques to be employed, such as Metropolis-Hastings, which leverage the learned nosied energies to potentially yield better weight estimations.
>
> Secondly, we admit that we are testing models on toy problems. However, generating samples **without any data** is a very difficult problem, and only FAB [3] can achieve this on Alanine Dipeptide (ALA2) now. Our preliminary results show that NEM can be more likely to succeed in ALA2 than iDEM. Here’s the link for our preliminary results: https://anonymous.4open.science/r/nem_rebuttal-0D92/aldp.png, and we also include more discussions in the below Q2 answer.
>
> [1] Chen, Wenlin, et al. "Diffusive Gibbs Sampling." arXiv preprint arXiv:2402.03008 (2024).
>
> [2] Shi, Zhekun, et al. "Diffusion-PINN Sampler." arXiv preprint arXiv:2410.15336 (2024).
>
> [3] Midgley, Laurence Illing, et al. "Flow annealed importance sampling bootstrap." arXiv preprint arXiv:2208.01893 (2022).
>
> **Q1: Why interatomic distances are less sharply peaked than ground truth**
>
> Thank you for highlighting this observation. The less sharply peaked distance distribution in our results may be influenced by the gradient clipping in iDEM and the energy smoothing employed in our work. Modeling extremely steep energy landscapes—where the energy approaches infinity as two particles come close in space—remains a challenge for model numeric stability.
> However, when both iDEM and BNEM sample from the same smoothed target energy function, our model consistently outperforms score-based methods. This demonstrates the robustness and effectiveness of our approach under comparable conditions.
>
> **Q2: Experiencing our methods on larger systems**
>
> We scale up our method to the ALA2 system, where preliminary results are provided in https://anonymous.4open.science/r/nem_rebuttal-0D92/aldp.png.
> DEM tends to exhibit mode-seeking behavior, fitting extremely sharp distributions to only a limited number of modes. In contrast, our method, while demonstrating slightly higher variance, shows a mass-covering behavior with reduced bias.
> Scaling up to very large systems remains beyond the capabilities of existing methods, and the ALA2 problem already poses significant challenges for current techniques. As the field progresses, larger and more complex systems will undoubtedly be considered. However, at this stage, it is premature to address such systems comprehensively.
>
> **End of Response**
>
> Thank you again for your valuable reviews. We hope that our responses have effectively addressed all of your concerns. Please feel free to comment if you have other concerns or suggestions. We hope you could increase the rating accordingly if we have addressed your concerns successfully.

---

> > ### Comment · Reviewer_MWcQ · 2024-11-25
> >
> > I agree that that the problem of sampling without data is an incredibly hard one. I applaud the authors for making an attempt on sampling alanine dipeptide. Ultimately, "transformational" is a subjective category, and I'm happy to disagree on this point. I will keep my current 8 rating as I believe it reflects the strong quality of the work.

---

### Meta-Review · Area_Chair_EdEc · 2024-12-21

**Metareview:**

This paper proposes a method to learn a Boltzmann sampler by learning the noise energy function to eventually get an estimation of the noise score function to sample efficiently according to the Boltzmann distribution. They provide theory bounding the variance of the energy function and provide experiments on a toy Mixture of Gaussians datasets, LJ-13, and LJ-55.

The concerns of the reviewers were the following:
1. The theory bounds the variance of the stochastic target of the energy function. However, since the quantity of interest is the score, It is not clear that having a lower variance in the score target and then differentiate the estimate is better than havnig a larger variance directly estimating the gradients. Hence the claim
> (NEM) has lower variance ... than previous related works.

is somewhat misleading since $\nabla E_K = S_K$ (the gradient of the estimator proposed in this work is exactly the one proposed in iDEM so I would argue that we cannot really compare the variances of this work and the previous related works)

2. Secondly, two reviewers had some concerns about the experiments. In particular, I agree that the authors should be able to reproduce the baselines to reasonable performance or at least take some time to explain why they think they could not reproduce it after having tried a significant amount of time.

3. The NEM methods introduce a computational overhead by requiring the computation of the gradient of the energy function during sampling. Even though the computational overhead is reasonable, it remains a disadvantage of the method.

Overall, even though I recommend rejection, I believe that a future iteration of this paper with a more substantial experimental section (in particular, with respect to the baselines) would be a significant contribution to the community.

**Additional Comments On Reviewer Discussion:**

Reviewer zWMt is the one who had the best experience and engaged the most in the discussions. I gave the other reviewers a last chance to champion the paper, which they did not do so I ended-up weighting more the opinion of Reviewer zWMt.

---

### Decision · Program_Chairs · 2025-01-22

Reject